# The Fabaceae in Northeastern Mexico (Subfamilies Caesalpinioideae (Excluding Tribe Mimoseae), Cercidoideae, and Detarioideae)

**DOI:** 10.3390/plants13172477

**Published:** 2024-09-04

**Authors:** Eduardo Estrada-Castillón, José Ángel Villarreal-Quintanilla, Gerardo Cuéllar-Rodríguez, Leticia Torres-Colín, Juan Antonio Encina-Domínguez, Jaime Sánchez-Salas, Gisela Muro-Pérez, Diego Axayácatl González-Cuéllar, Oralia Magaly Galván-García, Luis Gerardo Rubio-Pequeño, Arturo Mora-Olivo

**Affiliations:** 1Facultad de Ciencias Forestales, Universidad Autónoma de Nuevo León, Linares 67700, Mexico; aeduardoestradac@prodigy.net.mx (E.E.-C.); luis.cuellarr@uanl.mx (G.C.-R.); diegogonzalezcu@gmail.com (D.A.G.-C.); 2Departamento de Botánica, Universidad Autónoma Agraria Antonio Narro, Saltillo 25315, Mexico; javillarreal00@hotmail.com (J.Á.V.-Q.); jaencinad@gmail.com (J.A.E.-D.); 3Departamento de Botánica, Instituto de Biología, Universidad Nacional Autónomoa de Mexico, A.P. 70-233, Ciudad de México 04510, Mexico; lety@ib.unam.mx; 4Facultad de Ciencias Biológicas, Universidad Juárez del Estado de Durango, Gómez Palacio 35010, Mexico; j.sanchez@ujed.mx (J.S.-S.); giselamuro@ujed.mx (G.M.-P.); 5Escuela Libre de Ciencias Politicas y Administración (ELCPAPO), Xalapa 91120, Mexico; magalydemora@hotmail.com; 6Instituto de Ecología Aplicada, Universidad Autónoma de Tamaulipas, Ciudad Victoria 87019, Mexico; luisgerardoes@live.com.mx

**Keywords:** Leguminosae, subfamilies Caesalpinioideae, Cercidoideae, and Detarioideae, taxonomy, diversity, northeastern Mexico

## Abstract

As part of the Fabaceae project of northeastern Mexico and based on field work, collection of botanical samples over the past 37 years, and reviewing botanical materials in national and international herbaria, the diversity of legumes of the subfamilies Caesalpinioideae (excluding tribe Mimoseae), Cercidoideae, and Detarioideae in northeastern Mexico has been recorded. New nomenclatural changes in tribes and genera of the subfamily Caesalpinioideae found in the new scientific bibliography are included. The subfamily Caesalpinioideae (excluding the tribe Mimoseae) includes five tribes: tribe Caesalpinieae, with eight genera (*Caesalpinia*, *Coulteria*, *Denisophytum*, *Erythrostemon*, *Guilandina*, *Hoffmannseggia*, *Haematoxylum*, and *Pomaria*) and 21 species; tribe Cassieae with three genera (*Cassia*, *Chamaecrita*, and *Senna*) and 28 species; tribe Ceratonieae with one genus (*Ceratonia*) and 1 species; tribe Gleditsieae with one genus (*Gleditsia*) and 1 species. The subfamily Cercidoideae includes two genera (*Bauhinia* and *Cercis*) and eight species, and the subfamily Detarioideae includes only one genus and one species (*Tamarindus indicus*). The total flora of these three subfamilies comprises 18 genera and 63 species, including 56 native species and 7 exotic ones: *Bauhinia variegata*, *Cassia fistula*, *Ceratonia siliqua*, *Delonix regia*, *Erythrostemon gilliesii*, *Senna alata*, and *Tamarindus indicus*. Endemism includes a total of 22 species and nine infraspecific categories.

## 1. Introduction

The Fabaceae rank third in species diversification worldwide, comprising 770 genera and 19,581 species [1]. This family has had radical changes in its classification [2,3,4,5]. This has allowed us to recognize natural and homogeneous monophyletic groups of taxa and increase the number of subfamilies from three to six: Caesalpinioideae, Cercidoideae, Detarioideae, Dialioideae, Duparquetioideae, and Papilionoideae [1]. Based on the new concept of subfamilies, these are formed as follows. The subfamily Caesalpinioideae is composed of 11 tribes, 163 genera, and 4680 species [5]. The subfamily Cercidoideae is composed of 12 genera and 335 species [1]; the subfamily Detarioideae includes 84 genera and 760 species [1]; the subfamily Dialioideae includes 17 genera and 85 species [1]; and the subfamily Papilionoideae includes 503 genera and almost 14,000 species [1]. The subfamily Caesalpinioideae is the one that has had the greatest number of taxonomic modifications [4,5,6,7,8]. The most significant taxonomic changes within the current subfamily Caesalpinioideae are the inclusion of the subfamily Mimosoideae as a tribe within this [4,5], the subtribes Dialiinae and Duparquetiinae belonging to the tribe Cassieae [8] are now recognized as two different subfamilies, Dailioideae and Duparquetioideae, respectively [1].

The subfamily Cercidoideae is distinguished by its simple, entire or bilobed, or bifoliolate leaves. The subfamily Detarioideae is recognized by its extra-floral glands located on the edges of the leaflets abaxially. The inflorescences of the subfamily Caesalpinioideae have spiral anthotaxy, commonly with paripinnate leaves, and the seeds usually have open or closed pleurogram on each of the two sides.

The Fabaceae is one of the most economically important groups of plants in the world. This importance is due to the multiple uses that humanity has given them over time, such as carbon sequestration, reduction in greenhouse gas emissions, biodiversity conservation, diversification of livestock diets, flowers for pollinators, and shade for livestock [9,10,11,12]. At least 150 species of food and forage legumes have been recorded [13]. Among the most important species for human consumption, *Vigna*, *Phaseolus*, *Cicer*, *Pisum*, and *Lens* stand out [14]. Among the forage species most recognized for their economic importance at a global level are several species of *Trifolium*, *Medicago*, and *Lotus* [12,15]. Many legume species are capable of fixing atmospheric nitrogen in association with bacteria (*Rhizobium* and other genera), increasing its availability in the soil [16] and ensuring nutrient cycling for better plant growth for agricultural activities [17]. Legumes are also used for timber, firewood, and ornamental purposes [18]; species of the genera *Peltophorum*, *Dalbergia*, and *Sesbania* are used as timber; and species of the genera *Bauhinia*, *Delonix*, and *Cassia* are frequently used as ornamental [19]. Some species of legumes produce phytosterols, steroids found in plants similar to cholesterol and are part of the components of cell membranes [20]; *Tamarindus indica* seed oils are rich in phytosterols [21], including phytosterols in the diet can reduce blood cholesterol, decreasing the risk of cardiovascular diseases [20]. Many species of legumes are a source of food for humans, such as the leaves of *Acacia auriculiformis* and *A. concinna*, flowers of *Cassia fistula*, seeds of *Acacia concinna* and *Senegalia catechu*, and the pulp of *Vachellia nilotica* [18]. From a legume species conservation perspective, pollinators and legumes are interdependent; the legumes provide key forage and depend on insect pollination for reproduction and maintenance of genetic diversity [22]. Therefore, increasing legumes in intensive agricultural systems could be key to mitigating the decline of pollinators [23]. Legumes are present in practically all plant communities in Mexico [24], and a large number of uses of these species have been recorded from the nutritional, environmental, manufacturing of materials, fuel, medicinal purposes, and fodder point of view [25]. Among the most important uses are timber (*Neltuma*, *Ebenopsis*, *Havardia*, *Senegalia*, and *Vachellia*), food (*Phaseolus*, *Lens*, *Tamarindus*, and *Cicer*), medicinal purposes (*Eysenhardtia* and *Indigofera*), forage (*Medicago*, *Vachellia*, *Neltuma* and *Leucaena*), artisanal (*Ebenopsis*), green manure (*Leucaena*), fuel (*Neltuma*, *Vachellia*, *Senegalia*, *Havardia* and *Ebenopsis*), and charcoal (*Neltuma* and *Vachellia*) [26,27,28,29]. Several species, such as *Enteroloboium cyclocarpum*, *Neltuma juliflora*, *Lonchocarpus castilloi*, and *Dalbergia granadillo*, are used as timber, furniture, bars for train tracks, wooden floors, and guitars [30]. *Olneya tesota* is considered a high ecological value species [31], also used for medicinal purposes, food, artisanal, and hand tools [32,33]. *Parkinsonia praecox* is widely used as food and for medicinal purposes [34]; it is used for forage, fuel, and medicinal purposes [35]. Species of *Canavalia*, *Crotalaria*, *Glycine*, and *Lupinus* are used as a source of healthy products and human food [36,37,38,39]. *Lablab*, *Lathyrus*, and *Medicago* are widely used as a source of forage, green manure, industrial, medicinal purposes, ornamental, nitrogen fixer, and food [40,41,42]. The genus *Phaseolus* has represented one of the most important food legumes for human consumption [43]. *Erythrina* species provide high-quality protein and alkaloids [38]. *Tamarindus* is frequently used as a condiment [7] and to make flavored and sweet drinks [28].

The diversity of Fabaceae in Mexico consists of approximately 1893 species; at least 40% of them are endemic [25]. In northeastern Mexico, 56 genera and 134 species have been reported for Coahuila [10], while 67 and 234, respectively, have been reported for Nuevo León [17]. Twenty-two genera and 55 species of legumes of melliferous importance have been recorded for Tamaulipas [44]. Studies of subfamilies Caesalpinioideae, Cercidoideae, and Deatarioideae in northeastern Mexico have recorded the diversity of *Senna*, *Chamaecrista* and *Cassia* [8,45], *Caesalpinia* and *Erythrostemon* [46], *Bauhinia* [45,47], *Cercis* [45,48], *Delonix* [45], *Gleditsia* [49], *Haematoxylum* [45], *Hoffmannseggia* [45,50], *Parkinsonia* [45], *Pomaria* [51], *Tamarindus* [45], *Conzattia*, *Coulteria*, *Denisophytum*, and *Guilandina* [52,53]. 

Based on the nomenclatural changes of the new classification of Fabaceae [1,4,5] and given the lack of a complete study of the species richness of these three subfamilies in northeastern Mexico, the aims of this study are to provide a study of the diversity of this family and three of the subfamilies, Caesalpinioideae (excluding tribe Mimoseae, see [11]), Cercidoideae, and Detarioideae in northeastern Mexico, adding information concerning their new recently published nomenclature [1,4,5], ecology, uses, and distribution of all taxa.

## 2. Results

### 2.1. Diversity

The subfamily Caesalpinioideae is composed of six tribes: (1) the tribe Mimoseae (no covered in this manuscript see [11]); (2) the tribe Cassieae, which includes three genera (*Cassia*, *Chamaecrista*, and *Senna*) in northeastern Mexico (Figure 1) and 28 species; (3) the tribe Caesalpinieae, which includes eight genera (*Caesalpinia*, *Coulteria*, *Denisophytum*, *Erythrostemon*, *Guilandina*, *Haematoxylum*, *Hoffmannseggia*, and *Pomaria*) (Figure 1 and Figure 2) and 21 species; (4) the tribe Ceratonieae, which includes one genus and 1 species, *Ceratonia siliqua* (Figure 2); (5) the tribe Gleditsieae, which includes one genus and 1 species, *Gleditsia triacanthos* (Figure 3); (6) the tribe Schizolobieae, which is composed by two genera (*Delonix* and *Parkinsonia*) (Figure 3) and 3 species. The subfamily Cercidoideae includes two genera (*Bauhinia* and *Cercis*) (Figure 3) and eight species. The subfamily Detarioideae includes one genus and one species (*Tamarindus indicus*) (Figure 3). The total comprises 19 genera and 64 species, including 57 native and 7 exotic ones (*Bauhinia variegata*, *Cassia fistula*, *Ceratonia siliqua*, *Delonix regia*, *Erythrostemon gilliesii*, *Senna alata*, and *Tamarindus indicus*). With the exception of *Bauhinia* (seven species), the genera with the highest number of species belong to the tribe Caesalpinieae: *Senna* (21), *Pomaria* (6), *Chamaecrista* (6), *Erythrostemon* (5), and *Hoffmannseggia* (5). The most common growth forms of the species are herbaceous (31), shrubs (25), and trees (15). 

Table 1 shows the number of species per genus of each of the three subfamilies studied in northeastern Mexico, and it is compared with the diversity of species existing in Mexico and in the world, according to [1,4,5,52,53,54,55,56,57,58,59,60,61].

### 2.2. Endemism

Table 2 shows the subfamilies, genera, and species of the family Fabaceae and subfamilies Caesalpinioideae and Cercidoideae with endemism in Mexico.

### 2.3. Taxonomic Treatment

Subfamilies of Fabaceae studied (excluding morphological characters of the tribe Mimoseae (subfamily Caesalpinioideae)) and subfamily Papilionoideae occurring in northeastern Mexico
1A.Leaves bipinnateSubfamily **Caesalpinioideae**1B.Leaves pinnate22A.Leaves simples, unifoliolate, entire or bilobed, or compound and bifoliolate (if so, then shrubs) Subfamily **Cercidoideae**2B.Leaves bifoliolate (if so, then herbaceous) or pinnate33A.Fertile stamens 3, its filaments united; trees cultivated Subfamily **Detarioideae**
3B.Fertile stamens 7–10, its filaments freeSubfamily **Caesalpinioideae**

**Subfamily Caesalpinioideae** DC., Prodr. [A. P. de Candolle] 2: 473. 1825. Caesalpiniaceae R. Br., in M. Flinders, Voy. Terra Austral. 2: 551. 1814.

**Type:** *Caesalpinia* L., Sp. Pl. 1: 380 1753.

Trees, shrubs, subshrubs, or herbaceous, unarmed or armed with prickles or spines. Stipules absent or present. Leaves pinnate or bipinnate (paripinnate or imparipinnate). Flowers in racemes or panicles, rarely solitary, usually bisexual, less commonly unisexual and sterile flowers, actinomorphic or zygomorphic. Sepals 3–8, free or united, valvate or imbricate, rarely absent. Petals 3–8, free or fused, rarely absent, aestivation imbricate, sometimes valvate, with the adaxial petal the innermost. Stamens 3–10 free, homomorphic or heteromorphic. Staminodes present or absent present. Filaments are free, anthers longitudinally dehiscent or through terminal pores. One ovary with many ovulate. Fruit is a pod, one to many seeds, chartaceous or thick and woody, samariform, lomentiform, craspediform, dehiscent inherently or explosively or indehiscent. 

The subfamily Caesalpinioideae groups 163 genera and almost 4680 species [4], distributed in tropical, subtropical, and xeric regions of the world, and some species reach temperate areas.

Tribes of the subfamily Caesalpinioideae, excluding morphological characters of the tribe Mimoseae (see [11]), which occur in Northeastern Mexico.
1A.Petals absent; flowers unisexual; fertile stamen 5**Ceratonieae**1B.Petals present; flowers bisexual; fertile stamens 7–1022A.Shrubs or trees armed with straight thorns, the thorns branched **Gleditsieae**2B.Herbaceous, shrubs or trees unarmed or armed, if armed, the thorns or modified stipules single, never branched**3**3A.Leaves pinnate or bifoliolate**Cassieae**3B.Leaves bipinnate; if pinnate, the plants are armed with thorns (*Haematoxylum*)**4**4A.Lower lobe of the calyx is different from the rest of the lobes, modified into a hood shape in the bud stage**Caesalpinieae**4B.Lower lobe of the calyx is similar to the rest of the lobes, not modified into a hood shape at the bud stage**Schizolobieae**

**Tribe Caesalpinieae** Rchb., Fl. Germ. Excurs. 2(2): 544. 1832. 

**Type**: *Caesalpinia* L., Sp. Pl. 1: 380 1753.

Herbaceous shrubs or trees, unarmed or armed with prickles or thorns, are simple. Leaves pinnate or bipinnate, ending in a single pinnae or in a pair of pinnae. One pinnae has several pairs per leaf. Flowers are yellow, orange, red, scarlet-red, white, pink or green, bisexual or unisexual. Calyx is imbricate, the sepals are pubescent or glandular pectinate, and the lower lobe is different from the rest of the lobes, modified into a cap-shaped in the bud stage. Ten stamens, where the filaments are usually pubescent and glandular basally. Fruit flattened, membranous, papery, chartaceous to woody, indehiscent or dehiscent, the valves curl up when opening, pubescent and/or with glandular, plumose, dendritic or palmate trichomes and sessile black to orange glands, unarmed or provided with spines. 

The tribe contains 27 genera and 223 species [4]. 

The genus *Ceratonia*, previously included in Cassiaeae [8] and Caesalpinieae [52,53,54], is now included in the tribe Ceratonieae [4].
1A.Leaves pinnate***Haematoxylum***1B.Leaves bipinnate22A.Leaves ending in a pinna32B.Leaves ending in a pair of pinnae53A.Sepals persistent in the fruits; herbaceous***Hoffmannseggia***3B.Sepals caducous in the fruit; subshrubs, shrubs, or trees44A.Androecium and gynoecium basally cupped on the lower cuculate sepal, lower lateral sepals forming a platform at right angles to the abaxial cuculate sepal; pods with simple trichomes, glandular-punctate trichomes, and plumose, dendritic, and/or stellate trichomes; adaxial surface of leaflets with sessile black or orange glandular spots***Pomaria***4B.Androecium and gynoecium not cupped basally, flexed at lower cuculate sepal, lateral sepals not forming a platform; pods glabrous or with simple and/or glandular-dotted trichomes, the latter sometimes also dendritic or feathery; adaxial surface of leaflets without sessile black or orange glandular spots***Erythrostemon***5A.Unarmed plants; fruit slender, flattened, elliptical to oblong-elliptical, membranous or papery, indehiscent; margin of the base of the sepals glandular-pectinate; unisexual flowers; leaflets without glands***Coulteria***5B.Plants armed with thorns; fruit oblong-elliptic, elastically dehiscent, broadly elliptic, its valves curling up when opening, unarmed or armed with spines; base margins of sepals entire; bisexual flowers; leaflets without glands or with red glands 66A.Creeping shrubs; fruit broadly elliptical or circular, armed with spines***Guilandina***6B.Shrubs or erect trees; fruit elliptical-oblong, without prickles77A.Flowers yellow; pinnae 1 pair per leaf; leaflets 2–3 pairs per pinna***Denisophytum***7B.Flowers orange, red, or yellow; pinnae 8–18 pairs per leaf; leaflets 7–24 pairs per pinna ***Caesalpinia***

***Caesalpinia*** L. Sp. Pl. 1: 380. 1753. *Poinciana* L., Sp. Pl. 1: 380. 1753. *Caesalpinia* sect. *Brasilettia* DC., Prodr. [A.P. de Candolle] 2: 481. 1825. *Brasilettia* (DC.) Kuntze, Revis. Gen. Pl. 1: 164. 1891.

**Type:** *Caesalpinia brasiliensis* L. Sp. Pl. 1: 380. 1753.

Shrubs or trees, armed with curved prickles scattered on stems, leaves, leaf rachis, and leaflets or in pairs (base of leaves). Leaves bipinnate, paripinnate. Flowers in terminal or axillary racemes or panicles, zygomorphic, bisexual, yellow, orange, or red. Calyx 5-merous, caducous, the sepals free, the lower one cucullate, in the form of a cap. Petals 5, free, longer than sepals. Stamens 10, free, the filaments pubescent. Fruit linear, oblong to elliptic, coriaceous, compressed, explosively dehiscent, the valves twisted after dehiscence. Seeds laterally compressed.

Of the approximately 20 species of *Caesalpinia* recorded for northeastern Mexico, 19 are currently included within other genera. 

Nine species of *Caesalpinia* are currently recognized [5], distributed in neotropical regions, inhabiting tropical, subtropical woods, and coastal areas. Only one species is recorded in northeastern Mexico.

***Caesalpinia pulcherrima*** (L.) Sw., Observ. Bot. (Swartz) 166. 1791. Basionym: *Poinciana pulcherrima* L., Sp. Pl. p. 380. 1753.

**Type:** Habitat in Indiis. RCN: 2988. (Lectotype: designated by Roti-Michezzi in Webbia 13: 214. 1957): [icon] Crista Pavonis in Breyn, Exot. Pl. Cent.: 61, t. 22. 1678).

**Distinguishing features:** Shrub or tree, 3–7 m tall, armed with straight and sharp prickles, rarely unarmed. Leaves bipinnate. Pinnae 8–18 pairs per leaf. Leaflets 7–24 pairs per pinna. Flowers axillary or terminal, in racemes or panicles, red or yellow. Sepals are much shorter than the petals. Stamens 10, free, protruding of petals, filaments red. Fruit 6–16 cm long, linear-oblong, dark brown, or black when mature.

**Representative examined material:** Coahuila: 12-IX-1991, *M.A. Carranza s.n*. (ANSM). Nuevo León: 23-VII-2002, *E. Estrada 15007* (CFNL); 15-XI-2009, *E. Estrada s.n.* (CFNL). Tamaulipas: 8-VII-1986, *C.G. Romo 59* (UAT); 29-IX-1996, *C. Ramos 135* (UAT); 22-IX-1999, *A. Mora-Olivo 7595* (UAT); 17-VIII-1985, *M. Martínez 804* (UAT).

**Comments:** Apparently native from northwestern Mexico (Sonora) [5]. Used as ornamental in patios and private and public gardens. In northeastern Mexico, it is commonly known as *tabachin*.

***Coulteria*** Kunth, Nov. Gen. Sp. 6 ed. fol. 258. 1824. *Guaymasia* Britton & Rose, N. Amer. Fl. 23(5): 322. 1930.

**Type**: *Coulteria mollis* Kunth, Nov. Gen. Sp. 6: 330. 1824.

Trees or shrubs, unarmed, dioecious, young parts bronze velvety pubescent. Leaves bipinnate. Flowers axillary or in terminal racemes, zygomorphic, unisexual. Calyx gamosepalous, 5-lobed, the lower sepal cucullate, glandular pectinate. Staminate flowers with a rudimentary sterile ovary. Female flowers with ovary and style well developed, anthers lacking pollen. Fruit pendulous, flattened, oblong to elliptic, rarely suborbicular, chartaceous, papery or membranaceous, indehiscent or tardily so along one suture, sometimes persisting to the next season. Seeds are transversally arranged.

A genus of 7–8 species [5,55,56] in subtropical deciduous woods, seasonal tropical forests, and scrublands in semiarid areas in Mexico, Central America, and north of South America. 

***Coulteria pringlei*** (Britton & Rose) J.L. Contreras, S. Sotuyo & G.P. Lewis. Phytotaxa 291(1): 39. 2016. Basionym: *Brasilettia pringlei* Britton & Rose, in N. Amer. Fl. 23(5): 322 1930. *Caesalpinia pringlei* (Britton & Rose) Standl., in Trop. Woods 34: 40. 1933.

**Type**: Mexico, San Luis Potosi, Las Palmas, 8-VII-1896, *C. G. Pringle 6356* (Lectotype: designated by Sotuyo et al. in Phytotaxa 291(1): 39. 2017).

**Distinguishing features:** Shrub or tree up to 7 m tall. Pinnae 1–3 pairs per leaf. Leaflets 6–12 pairs per pinna, alternate along the rachis, prominently veined and yellowish abaxially. Flowers in axillary racemes, unisexual, yellow, banner strongly recurved, with tick texture. Lower sepal cucullate, larger than the others, marginally eroso-glandular. Fruit 5–8 cm long, compressed, oblong to elliptical, indehiscent, light-brown. 

**Representative examined material:** Tamaulipas: 25-I-1985, *L. Hernández 1389* (UAT, MEXU); 15-VIII-1930, *H.H. Bartlett 10964* (NY3196585!).

**Comments:** Endemic to Mexico. Recorded only in the state of Tamaulipas, in piedmont scrub, 300 (Sierra de San Carlos)–800 m altitude (Municipality of Abasolo), outside the area, in San Luis Potosi, Guanajuato, Querétaro, Guerrero, Sinaloa, Oaxaca, and Hidalgo.

***Denisophytum*** R. Vig., Notul. Syst. (Paris) 13(4): 349. 1948.

**Type.** *Denisophytum madagascariense* R. Vig. Notul. Syst. 13: 349. 1949.

Shrubs or trees up to 5 m tall, armed with curved or straight prickles. Leaves bipinnate. Stipels spinose arises in the insertion of pinnae on the leaf rachis and occasionally arises at the base of leaflets. Pinnae 1–6 pairs per leaf. Leaflets 2–11 pairs per pinna. Flowers in axillary or terminal racemes, yellow, bisexual, zygomorphic. Calyx 5-lobed, the lower one marginally entire, cucullate. Petals free, the banner with red tones in the center. Stamens 10, free, hairy. Fruit flattened, coriaceous, explosively dehiscent, valves twisted after dehiscence. 

Genus of 8 species of North and South America, Madagascar, and Africa [5], almost similar to *Caesalpinia*; the only striking difference between them is the color of flowers, which is more variable in *Caesalpinia* (orange, red, green, and white).

***Denisophytum sessilifolium*** (S. Watson) E. Gagnon & G.P. Lewis, PhytoKeys 1(1): 47. 2016. Basionym: *Caesalpinia sessilifolia* S. Watson, Proc. Amer. Acad. Arts and Sci. 21: 450. 1886*. Poinciana sessilifolia* (S. Watson) Rose, in Contrib. U. S. Nat. Herb. 13(9): 303. *1911.*


**Type:** Mexico, Mexico, Bolson de Mapimi, 10-V-1847, *Gregg s.n.* (Original material: NY!); Mexico, Coahuila, on hills and mesas about Jumulco, V-1885, *C.G. Pringle 202* (Original material: MO 125071!).

**Distinguishing features:** Shrub, 1–2 m tall, glabrous, armed with a pair of straight or curved prickles at the insertion of the petiole or immediately below. Pinnae 1 pair per leaf. Leaflets 2–3 pairs per pinna. Flowers axillary or in terminal racemes. Corolla yellow. Stamens 10, free, filaments hairy. Calyx of 5 free sepals. Fruit 2–3 × 1–1.5 cm, oblong to rhomboid-trapezoid, dehiscent, reddish-brown. 

**Representative examined material:** Coahuila: 18-VII-2007, *J. A. Alba 110* (CFNL); 18-VII-2007, *J. A. Alba 148* (CFNL); 25-VIII-1988, *J.A. Villarreal 4386* (ANSM). 

**Comments:** Endemic to north of Mexico, recorded only in the state of Coahuila in arid shrublands. Outside the study area, in Durango and Chihuahua.

***Erythrostemon*** Klotzsch, in Link, Klotzsch & Otto, Icon. Pl. Rar. Horti. Berol. 2: 97, t. 39. 1844. *Poincianella* Britton & Rose, N. Amer. Fl. 23(5): 327. 1930. *Schrammia* Britton & Rose, N. Amer. Fl. 23(5): 317. 1930.

**Type:** *Erythrostemon gilliesii* (Hook.) Klotzsch, Icon. Pl. Rar. [Link, Klotzsch & Otto] 1: 97, t. 39. 1844.

Subshrubs, shrubs, or trees, with or without stipitate glands. Leaves bipinnate. Leaf margin without or with black sessile glands. Flowers in distal racemes, bisexual, zygomorphic, bright yellow, whitish yellow, or pink. Calyx dialysepal, sepals 5, not cupped in the lower cucullate sepal. Corolla of 5 free reflexed petals. Stamens 10, free, the filaments hairy. Pod oblong to elliptic, coriaceous, with or without stipitate glands. 

An American genus of 31 species, including those within the former genus *Poincianella* [5], distributed from the south USA, through Mexico and Central America to South America.
1A.Herbaceous perennial or subshrubs; many stems rising from the base; terminal pinna much larger than lateral pinnae; leaflets abaxially with dark glands, or only one at apex***E. caudatus***1B.Shrubs; stems woody; terminal pinnae equal in size to lateral pinnae or only slightly longer; leaflets abaxially without dark glands 22A.Flowers laterally compressed***E. exostemma*** var. ***tampicoanus***2B.Flowers not laterally compressed33A.Petals 2.2–3.2 cm long; stamen filaments 7–12 cm long; exotic species***E. gilliesii***3B.Petals 12 mm long or shorter; stamen filaments 13 mm long or shorter; native species 44A.Perennial herbaceous or subshrub, 80 cm tall or less, rhizomatous, colonial habit; leaflets obovate–elliptic; inflorescence without glands, except at the base of the calyx and apex of the pedicels (glandular–stipitate); stamens 9–10 mm long; fruits orbicular, 2.5 cm long; seeds 1–2***E. phyllantoides***4B.Shrubs or trees, 1 m tall or more, never rhizomatous; leaflets obovate–elliptic, oblong-elliptic to orbicular; inflorescence axis and pedicels without glands; stamens 10 mm long or longer; fruit oblong, 4–8 cm long; seeds 3–6 ***E. mexicanus***

***Erythrostemon caudatus*** (A. Gray) E. Gagnon & G. P. Lewis, Phytokeys 71. 120. 2016. Basionym. *Hoffmannseggia caudata* A. Gray, Boston J. Nat. Hist. 6: 179. 1850. *Caesalpinia caudata* (A. Gray) E. M. Fisher, Bot. Gaz. 18: 123. 1893. *Schrammia caudata* (A. Gray) Britton & Rose, N. Amer. Flora 23(5): 317. 1930.

**Type:** USA, Texas, western Texas to El Paso, New Mexico, V-1849, *Wright 146* (Holotype: GH, Isotype K81717!). 

**Distinguishing features:** Herbaceous, perennial, up to 1 m tall. Pinnae 2–4 pairs per leaf, plus one terminal pinnae longer than the lateral ones. Leaflets 8–20 pairs in the terminal pinnae, with dark glands abaxially or the glands absent but the leaflet apex is gland-tipped. Flowers in axillary or terminal racemes, yellow. Calyx lobes fimbriate, the lower sepal cucullate. Stamens 10, hairy basally. Fruit 2.5–4.5 cm long, flattened, explosively dehiscent, pubescent, and with sessile and stalked glands. 

**Representative examined material:** Coahuila: 15-IV-1999, *A. Mora-Olivo 7519* (UAT). Nuevo León: 17-IV-1939, *T.C. Frye 2380* (IND-49057!); 26-XI-1966, *Ripley & Barneby 14783a* (NY!); 17-III-1962, *J.G. Rivas* (TEX260203!); 24-III-1062, *M. Dominguez M. 8206* (TEX260204!).

**Comments:** Endemic to northeastern Mexico and south of Texas. In Tamaulipan thornscrub and mezquite woods, 80–300 m. In northeastern Mexico, it is recorded only in the states of Nuevo León and Coahuila.

***E. exostemma*** subsp. ***tampicoanus*** (Britton & Rose) E. Gagnon & G. P. Lewis. PhytoKeys 71: 122. 2016. Basionym. *Poincianella tampicoana* Britton & Rose, N. Amer. Fl. 23(5): 330. 1930. *Caesalpinia tampicoana* (Britton & Rose) Standl., Publ. Field Mus. Nat. Hist., Bot. Ser. 11(5): 159. 1936. *Caesalpinia exostemma* var. *tampicoana* (Britton & Rose) G. P. Lewis), Roy. Bot. Gard. 72. 1998. 

**Type**. Mexico, Veracruz, vicinity of Pueblo Viejo, 2 km S of Tampico, 1/2-VI-1910. *Palmer 556* (Holotype: US4615!).

**Distinguishing features:** Shrub or tree, up to 10 m tall, with simple and stellate or plumose hairs mixed with stipitate glands. Pinnae 2–5 pairs per leaf, plus one terminal. Leaflets 3–6 pairs per pinna, their margin revolute and glandular. Flowers in terminal or lateral racemes, yellow, laterally compressed. Calyx gibbous, pink-salmon to orange. The lateral petals with stipitate glands. Stamen filaments bicolored, hairy white basally, red in the distal half. Fruit 6–7.8 cm, explosively dehiscent. 

**Representative examined material:** Tamaulipas: 3-I-1939, *LeSueur 188* (F!); 3-III-1961, *King 4015* (NY3570607!); 15/I-1/VI-1910, *E. Palmer 556* (NY4615!, in Pueblo Viejo, Veracruz, town adjacent to Tampico, separated only by the Pánuco river).

**Comments:** Endemic to Mexico. Recorded only in the south of the state of Tamaulipas, in deciduous woodlands, 30–100 m. Outside the area, in Veracruz.

***E. gilliesii*** (Hook.) Klotzsch, Ic. Pl. Rar. Horti. Berol. 2 (3): 97, t. 39. 1844. Basionym. *Poinciana gilliesii* Wall. ex Hook., Bot. Misc. 1: 129. 1830. *Caesalpinia gilliesii* (Hook.) D. Dietr., Synop. Pl. 2: 1495. 1840.

**Type:** Argentina, near Rio Quatro and Rio Quinto, and in La Punta de San Luis, *Gillies s.n.* (Holotype: K).

**Distinguishing features:** Shurb, 1–3 m tall. Pinnae 7–15 pairs plus one terminal per leaf. Leaflets 7–12 pairs per pinna. Flowers in terminal racemes, with abundant stipitate glands, yellow with light red tones. Petals obovate, 2.2–3.2 cm long. Stamen filaments red, 7–11 cm long. Fruit 5–7 cm long, flattened, oblong, elastically dehiscent, with pedicellate glands. 

**Representative examined material:** Coahuila: 8-VI-1984, *A. Rodríguez 123* (ANSM); 9-V-1987, *A. Rodríguez 3697 (*ANSM). Nuevo León: 15-X-1987, *E. Estrada 362* (ANSM). 

**Comments: N**ative of South America (Argentina or Uruguay) [46]. Widely used in northeastern Mexico as ornamental. Sometimes grows in disturbed places.

***E. mexicanus*** (A. Gray) E. Gagnon & G. P. Lewis, PhytoKeys 71: 124. 2016. Basionym*. Caesalpinia mexicana* A. Gray, Proc. Amer. Acad. Arts 5: 157. 1861. *Poinciana mexicana* (A. Gray) Rose, Contr. U.S. Natl. Herb. 13: 303. 1911. *Poincianella mexicana* (A. Gray) Britton & Rose, N. Amer. Fl. 23(5): 330. 1930.

**Type:** Mexico, Nuevo León, near Monterrey, 11-II-1847, *Gregg s.n. (*Lectotype*:* designated by McVaugh, 1987*).*

**Distinguishing features:** Shrub or tree. Leaves bipinnate. Pinnae 2–3 pairs per leaf plus 1 terminal. Leaflets 3–5 pairs per pinna, obovate–elliptic to orbicular. Pedicels without glands. Flowers in axillary and/or terminal racemes, yellow, 12 mm long or shorter. Stamens 10, free, 10–13 mm long. Fruit 4–7 cm long, oblong. Seeds 3–6. 

**Representative examined material:** Coahuila**:** 12-IX-1991, *M.A. Carranza s.n.* (ANSM)**.** Nuevo León: Agualeguas: 2-IV-2002, *E. Estrada 13400* (CFNL); 21-III-2002, *E. Estrada 13316* (CFNL); 5-VII-2001, *C. Yen y E. Estrada 12860* (CFNL); 5-VII-2001, *C. Yen y E. Estrada 12846* (CFNL). Tamaulipas: 10-XI-2006, *J. Encina 1674*, *I Ramirez*, *F.J. Diaz* (CFNL); 9-III-1985, *J. Jimenez 002* (CFNL); 21-VIII-1991, *E Estrada 2093*, *J. Fairey*, *C. Schoenfeld* (CFNL); 23-IV-2009, *E. Estrada 20774* (CFNL); 6-VI-1987, *A. Mora-Olivo 7246* (UAT); 4-IX-1985, *M. Martínez 849* (UAT); 21-II-1998, *M. Galván* 690 (UAT).

**Comments:** From southeastern Texas, through northeastern Mexico to Querétaro, Hidalgo, and Veracruz. In Tamaulipan thornscrub, piedmont scrub, and sometimes in desert scrublands, 290–1500 m. Known as “*potro*” or “*yerba del potro*”. Used as ornamental.

***E. phyllanthoides*** (Standl.) E. Gagnon & G. P. Lewis, 71: 12–126. 2016. Basionym: *Caesalpinia phyllanthoides* Standl., Contr. U.S. Natl. Herb. 23: 425. 1922. *Poincianella phyllanthoides* (Standl.) Britton & Rose, N. Amer. Fl. 23(5): 332. 1930.

**Type**: Mexico, Tamaulipas, Hacienda Buena Vista, 18-VI-1919, *Wooton s.n.* (Holotype: US2606!).

**Distinguishing features:** Herbaceous to subshrub, rhizomatous, up to 80 cm tall, forming colonies. Pinnae 3–4 pairs per leaf. Leaflets 2–3 pairs per pinna, obovate–elliptic. Flowers yellow in axillary or distal racemes, with stipitate glands at the base of the calyx and apex of the pedicels. Stamens 9–10 mm long. Fruit orbicular, 2.5 cm long.

**Representative examined material:** Tamaulipas: 26-IV-1960, *Crutchfield & Johnston 5343* (MEXU); 8-II-1960, *Crutchfield & Johnston 5071* (MEXU); 18-VI-1914, *E.O. Norton?* (NYBG). 

**Comments:** endemic to southeastern Texas and Tamaulipas, in Tamaulipan thornscrub, 100–300 m.

***Guilandina*** L., Sp. Pl.: 381. 1753. *Bonduc* Mill., Gard. Dict. Abr., ed. 4: 28. 1754. *Caesalpinia* subgenus *Guilandina* (L.) Gillis & Proctor, J. Arnold Arbor. 55(3): 426 1974.

**Type:** *Guilandina bonduc* L., Sp. Pl. 1: 381. 1753.

Monoecious climbers, lianas, or trailing shrubs armed with prickles, commonly forming dense colonies. Stems and branches are armed with curved spines and a pair of spines below the leaf base. Leaves bipinnate, with a pair of prickles in the insertion of pinnae and leaflets. Flowers in supra or distal racemes, yellow, unisexual; female ones staminate but sterile, and the male ones have no functional pistil. Sepals valvate, the lower one cucullate. Petals 5, similar or longer than sepals. Stamens 10, free, basally hairy. Fruit oblong to elliptic, swollen, frequently armed with spines. Seeds 1–4, subspherical to obovoid, orange, gray to brown, shiny.

Genus not fully studied is its taxonomy, still with an indeterminate number of species, up to 20 [4], with pantropical distribution, the species are widely distributed in Asia, Indo–China, Japan, South Africa, Central America, the Caribbean region, Madagascar, and Australia.

Only one species was recorded in northeastern Mexico, *Guilandina bonduc.*

***Guilandina bonduc*** L. Sp. Pl. 1: 381. 1753.

**Type:** (Lectotype: designated by Dandy & Exell, J. Bot. 76: 177, 1938: Voucher Herb. Hermann 3: 35. No. 156 (BM)). 

**Distinguishing features:** Subshrubs, creeping, up to 6 m tall. Stems and leaves armed with prickles. Leaves bipinnate. Pinnae 4–8 pairs per leaf. Leaflets 4–8 pairs per pinna. Flowers yellow to green-yellow. Calyx pubescent. Fruit 5–10 × 4–6 cm, ovate, compressed but little swollen, spinescent, brown, tardily dehiscent. Usually 2 seeds, up to 2 cm diameter, gray and shiny.

**Representative examined material:** Tamaulipas: 14-IV-2023, *E Estrada 26080*, *C. Yen* (CFNL); 30-IX-1986, *D. Baro 900* (UAT); 18-VII-2018, *Castillo Campos J. Pale 26710* (MEXU).

**Comments:** Recorded only in the state of Tamaulipas, in coastal areas, forming dense tickets. In Mexico along the Pacific and Gulf of Mexico coasts, also in Puebla, and San Luis Potosí.

***Haematoxylum*** L., Sp. Pl. 1: 384. 1753. *Cymbosepalum* Baker, Bull. Misc. Inform. Kew 1895(100–101): 103. 1895. 

**Type:** *Haematoxylum campechianum* L., Sp. Pl. 1: 384. 1753.

Shrubs or trees up to 13 m tall. Branches armed, the spines straight and triangular. Leaves pinnate (in northeastern Mexico). Leaflets 2–6 pairs per leaf, obovate, when bipinnate (very rarely so) leaves, pinnae 1–3 pairs per leaf plus one terminal pinnae. Leaflets 2–6 pairs per pinna. Flowers in axillary or distal racemes or panicles, yellow, light-yellow to white, bisexual. Calyx 5-merous, the lower sepal cucullate and slightly covering the other 4 sepals in bud. Corolla of 5 petals. Stamens 10, free, the filaments hairy in the lower half. Fruit flattened, chartaceous to membranaceous, dehiscing along the middle of the valves or near the margin, never along the sutures. Seeds 1–3, flattened. 

Genus of five species, four in the tropics of America and one endemic to Namibia [59], in semi-deciduous woods, dry tropical scrublands, thorn scrub, and riparian areas. 

***Haematoxylum brasiletto*** H. Karst., Fl. Columb. (H. Karst.) 2(1): 27, t. 114. 1862. Basionym. *Haematoxylum boreale* S. Watson. Proc. Amer. Acad. 21: 426. 1886.

**Type:** South America, Habitat regiones calidas et aridas montanas ad pedes Andium Columbiae septentrionalis, ab incolis “Brasil” vel “Brasiletto” dicta Fl. Columb. (H. Karst.) 2(1): 27, t. 114. 1862. 

**Distinguishing features:** Shrub or tree, up to 10 m tall, armed with dark, straight prickles. Leaves pinnate. Leaflets 3–4 pairs per leaf, obovate. Calyx red. Fruit 2–6 cm long, oblong, flattened, opening along the middle of the valves or parallel to the margin, acute at both ends. 

**Representative examined material:** Tamaulipas: 1-V-1969, *F. González M. 2303* (MEXU).

**Comments:** Rare in northeastern Mexico, only one collection of the central-south in the state of Tamaulipas, near Cruillas town, with tall sub thorn scrub, 500 m. It is the only member of Caesalpinieae with pinnate leaves.

***Hoffmannseggia*** Cav., Icon 4: 63. 1798. *Larrea* Ort., Nov. Pl. Descr. Dec.: 15, t. 2. 1797.

**Type:** *Hoffmannseggia glauca* (Ortega) Eifert. Sida 5: 43. 1972.

Herbaceous perennials or subshrubs, frequently with orange glands. Leaves bipinnate, imparipinnate. Flowers in terminal racemes, with or without pedicellate glands, yellow, reddish-yellow, or salmon color. Calyx with a short tube, pubescent, glandular, or both, sepals persistent. Petals 5, with glandular hairs at the base. Stamens 10, subequal, free, filaments glandular or villous. Fruit lunate, arched, rectangular, straight, or oblong, with simple and/or glandular capitate and subsessile trichomes. 

Genus of 22 species [50] with amphitropical distribution; 11 species recorded in Mexico and 11 in South America [50].
1A.Subshrubs, woody at least basally; flowers arising on the branches; fruit ovate–angled, lunate, or arched; dehiscent; its valves rarely curling longitudinally after dehiscence***H. drummondii***1B.Perennial herbaceous; flowers arising on peduncles from the base of the plant; fruit lunate, arched, orbicular to unequally ovate–angled in profile, indehiscent or dehiscent with the valves curling longitudinally after dehiscence 22A.Sepals and pedicels strigose or tomentose, without obvious glandular trichomes32B.Sepals and pedicels with evident glandular multicellular trichomes43A.Fruit oblong, rectangular, straight, apically acute***H. watsonii***3B.Fruit strongly arcuate, apically rounded***H. drepanocarpa***4A.Petals with evident glandular trichomes in the base; legume slightly arched, indehiscent, sparsely tomentose***H. glauca***4B.Petals without glandular trichomes in the base, or only the banner with few glandular trichomes; legume rectangular or arched, with abundant glandular trichomes 55A.Legume rectangular, reticulate-veined, indehiscent; leaflets usually strigous or villous***H. humilis***5B.Legume arched, dark reticulate-veined, dehiscent; leaflets generally glabrous***H. oxycarpa*** var. ***oxycarpa***

***Hoffmannseggia drepanocarpa*** A, Gray, Pl. Wright. 1; 58, 1852. Basionym: *Caesalpinia drepanocarpa* (A. Gray) Fisher, Bot. Gaz, 18: 122. 1893. *Larrea drepanocarpa* (A. Gray) Britton & Rose, N. Am. Fl. 23(5); 312. 1930.

**Type**: (Lectotype designated by Simpson, B.B. 1999, A revision of *Hoffmannseggia* (Fabaceae) in North America. *Lundellia* 2: 20: Voucher: United States, Texas, El Paso, V-1881, *C. Wright 1027* (BC: MO-176084; A 1846469). 

**Distinguishing features:** Herbaceous, perennial. Pinnae 4–9 pairs plus one terminal per leaf. Leaflets 6–9 pairs per pinna. Flowers in terminal racemes, yellow, commonly tinged with red, turbinate in lateral view. Calyx lobes strigose. Banner with a small tuft of trichomes in the claw adaxially. Fruit 2–4 × 0.5–0.8 cm, falcate or forming a complete circle, parallel margined, apically rounded, pulverulent.

**Representative examined material:** Coahuila: 1-IX-1938, *I.M. Johnston 7280* (GH).

***H. drummondii*** Torr. & A. Gray, Fl. N. Amer. 1(3): 393. 1840. Basionym: *Caesalpinia drummondii* (Torr. & A. Gray) Fisher, Bot. Gaz. 18: 123. 1893. *Larrea drummondii* (Torr. & A. Gray) Britton, N. Amer. Fl. 23(5): 311. 1930. *Hoffmannseggia texensis* Fisher, Contr. U.S. Natl. Herb. 1: 147. 1892. *Caesalpinia texensis* (Fisher) Fisher, Bot. Gaz. 18: 123. 1893. *Larrea texensis* (Torr. & A. Gray) Britton, N. Amer. Fl. 23(5): 311. 1930. 

**Type:** USA, New Mexico, Piedernales, *T.S. Drummond s.n.* (Holotype: NY431810!). 

**Distinguishing features: S**ubshrub, woody basally, young parts, and leaf rachis with multicellular glandular trichomes. Pinnae 1 pair, plus one terminal per leaf. Leaflets 3–6 pairs per pinna. Flowers in lateral or terminal racemes arise in the branches. Calyx with few multicellular glandular trichomes. Fruit lunate or arched, broadest in the middle, subglabrous, the valves coiled outward after dehiscence. 

**Representative examined material:** Tamaulipas: 10-XII-1959, *M.C. Johnston 4934* (MEXU); 12-XII-1959, *M.C. Johnston 4987*, *J. Crutchfield* (TEX). 

**Comments**: Endemic to northeastern Mexico and southeastern Texas, in scrublands, 200–600 m. 

***H. humilis*** (M. Martens & Galeotti) Hemsley, Biol. Centr.-Amer. Bot. 1: 326. 1880. Basionym: *Pomaria humilis* M. Martens & Galeotti; Bull. Acad. Roy. Sci. Bruxelles 10(2): 303. 1843. *Larrea humlis* (M. Martens & Galeotti) Britton, N. Amer. Fl. 23(5): 316. 1930. *H. gladiata* Benth. in A. Gray, Pl. wright. 1: 57. 1852. *Caesalpinia gladiata* (Benth.) Fisher, Bot. Gaz. 18: 122. 1893. *Larrea gladiata* (Benth.) Britton; N. Amer. Fl. 23(5): 314. 1930. *Hoffmannseggia platycarpa* Benth., in A. Gray, Pl. Wright. 1: 57. 1852. *Larrea platycarpa* (Benth.) Britton, N. Amer. Fl. 23(5): 314. 1930. Larrea potosina Britton, N. Amer. Fl. 23(5): 313. 1930. *Larrea pueblana* Britton, N. Amer. Fl. 23(5): 313. 1930. *Hoffmannseggia pueblana* (Britton) Britton, Publ. Field. Mus. Nat. Hist. Bot. Ser. 11(5): 160. 1936. *Larrea villosa* Britton, N. Amerc. Fl. 23(5): 313. 1930.

**Type**: México, Puebla, *Galeotti 3228* (Holotype: not found, Isotype: K264532!). 

**Distinguishing features:** Herbaceous perennial. Stems up to 18 cm long. Pinnae 3–13 pairs per leaf, plus one terminal. Leaflets 4–7 pairs per pinna, strigose or villous. Sepals with glandular multicellular trichomes. Fruit 1–3 × 0.5 cm, rectangular with parallel margins, slightly arched, indehiscent, reddish, glandular-pubescent. 

**Representative examined material:** Nuevo León: 15-III-1981, *G.B. Hinton* et al. *18131* (TEX). Tamaulipas: 20-V-1971, *M.C. Johnston* et al., *11147* (TEX).

**Comments:** Endemic to Mexico, from Nuevo León, through Hidalgo, Oaxaca, San Luis Potosi, Querétaro, Sna Luis Potosí, and Veracruz. In desert scrub and chaparral communities, 2400 m.

***H. glauca*** (Ortega) Eifert, Sida 5(1): 43. 1972. Basionym: *Caesalpinia chicamana* Killip & J.F. Macbr., Publ. Field Mus. Nat. Hist., Bot. Ser. 13(3): 191.1943. *Caesalpinia falcaria* Fisher Bot. Gaz. 18: 122. 1893. *Hoffmannseggia falcaria* Cav., Icon. Descr. Pl. 4: 63. Tab 392. 1798. *Larrea glauca* Ort., Nov. Pl. Descr. Dec. 2: 15–16, pl. 2. 1797. *Poinciana hirsuta* Sessé & Moc., Naturaleza (Mexico City), ser. 2, 1(App.): 66. 1888. *Hoffmannseggia alpina* Gillies ex Hook. & Arn., Bot. Misc. 3: 209. 1833. *Hoffmannseggia chinensis* Miers ex Hook. & Arn., Bot. Misc. 3: 209. 1833. *Hoffmannseggia demissa* Benth., Smithsonian Contr. Knowl. 3(5): 56. 1852. *Hoffmannseggia densiflora* Benth., in A. Gray. Pl. wright. 1: 55. 1852. *Hoffmannseggia falcaria* var. *capitata* Fisher, Contr. U.S. Natl. Herb. 1: 145. 1892. *Hoffmannseggia falcaria* var. *demissa* (A.Gray) Fisher, Contr. U.S. Natl. Herb. 1: 145. 1892. *Hoffmannseggia falcaria* var. *pringlei* Fisher, Contr. U.S. Natl. Herb. 1: 145. 1892. *Hoffmannseggia falcaria* var. *rusbyi* Fisher, Contr. U.S. Natl. Herb. 1: 145. 1892. *Hoffmannseggia falcaria* Cav. var. *stricta* (Bent. In A. Gray) Fisher, Contr. U.S: Natl. Herb. 1: 144. 1892. *Hoffmannseggia stricta* Benth., in A. Gray, Pl. wright 1: 56. 1852. *Hoffmannseggia stricta* var. *demissa* A. Gray, Smithsonian Contr. Knowl. 3(5): 56. 1852. *Larrea densiflora* (Benth.) Britton. N. Amer. Fl. 23(5): 311. 1930. *Caesalpinia falcaria* var. *capitata* (Fisher) Fisher, Bot. Gaz. 18: 122. 1893. *Caesalpinia falcaria* var. *densiflora* (Benth. ex A.Gray) Fisher Bot. Gaz. 18: 122. 1893. *Caesalpinia falcaria* var. *pringlei* (Fisher) Fisher, Bot. Gaz. 18: 122. 1893. *Caesalpinia falcaria* var. *rusbyi* (Fisher) Fisher, Bot. Gaz. 18: 122. 1893. *Caesalpinia falcaria* var. *stricta* (Benth.) Fisher, Bot. Gaz. 18: 122. 1893. *Caesapinia glauca* (Ortega) O. Kuntze, Revis. gen. pl. 3(2): 52. 1898. *Caesalpinia glauca* var. *glandulosissima* Kuntze, Revis. Gen. Pl. 3(2): 53. 1898. *Caesalpinia glauca* var. *pauciglandulosa* Kuntze, Revis. Gen. Pl. 3(2): 53. 1898. 

**Type**: (Lectotype: designated by Fisher, 1892: Revision of the North American species of *Hoffmansseggia*. Contr. U.S. Natl. Herb. 1: 145. Voucher: USA, Western Texas to El Paso, New Mexico [Valley of the Pecos], 14-VIII-1849, *C. Wright 148* (GH00254175!). 

**Distinguishing features:** Perennial, herbaceous, sub-scapose or caulescent. Pinnae 3–13 pairs per leaf, plus one terminal one. Leaflets 4–13 pairs per pinna. Flowers yellow. Sepals free, glandular-pubescent. Petals with trichomes, basally. Fruit 2–4 × 0.5–0.8 cm, falcate or rectangular, flattened, indehiscent, tomentose, with few glandular trichomes. 

**Representative examined material:** Coahuila: Coahuila: 1-IX-2007, J.I. *Calzada 24965* (ANSM), 7-VIII-1973. *J. Henrickson 12093b* (ANSM). Nuevo León: 5-VIII-1993, *E. Estrada 2452* (CFNL); 1-VII-1999, *E. Estrada 10963* (CFNL); 31-V-2003, 21-VI-2003, *E. Estrada 15772* (CFNL); 7-IV-1999, *E. Estrada 10143* (CFNL). Tamaulipas 24-V-1959, *D. Seigler*, *E. Rodríguez 1286* (TEX); 7-XII-1993, *A. Mora-Olivo 5033* (UAT).

**Comments:** In the southern USA, frequent in the north of Mexico and extending to Puebla. In desert scrublands, halophytic grasslands, and Tamaulipan thorn scrub, 280–2200 m. In South America, it is distributed in Perú, Chile, and Argentina [50].

***H. oxycarpa*** Benth. in A. Gray var. ***oxycarpa***, Pl. Wright 1: 55. 1852. Basionym: *Caesalpinia oxycarpa* (Benth.) Fisher, Bot. Gaz. 18: 122. 1893. *Larrea oxycarpa* (Benth.) Britton, N. Amer. Fl. 23(5): 312. 1930.

**Type**: (Lectotype: designated by Simpson, B.B. 1999, A revision of *Hoffmannseggia* (Fabaceae) in North America. *Lundellia* 2: 38: Voucher: Mexico, Monterrey, non date, *L.A. Edwards*, *J.H. Eaton 12* (GH62317!).

**Distinguishing features:** Herbaceous perennial. Stems densely villous and glandular. Pinnae 1–4 pairs of pinnae plus one terminal per leaf. Leaflets 4–9 pairs per pinna. Flowers in racemes, yellow. Sepals villous and with multicellular glandular trichomes. The banner is without or with few glandular trichomes at the base on the external side, and lateral petals are without glandular trichomes. Fruit 1–3.6 × 0.5–0.7 cm, falcate, arched, dehiscent, pubescent and with abundant capitate glandular trichomes. 

**Representative examined material:** Coahuila: Coahuila: 18-IV-1992, *J.A. Villarreal 1368a* (ANSM), 11-IX-1991, *M.A. Carranza s.n.* (ANSM). Nuevo León: 15-IV-2001, *C. Yen y E. Estrada 12932* (MEXU); 14-VII-2002, *C. Yen y E. Estrada 14900* (MEXU); 7-IV-1980, *E. Bridges*, *L. Woodruff 13132* (TEX-LL); 7-IV-1962, *M. Domínguez*, *Wm. McCart 8226* (TEX-LL). Tamaulipas: 7-IV-1962, *M. Dominguez*, *M.W. McCart 8226* (TEX). 

**Comments:** Endemic to northeastern Mexico and southern Texas, 360–2000 m, Tamaulipan thornscrub, desert scrublands, halophytic grasslands, rosteophyllous scrub. Frequent in calcareous soils.

***H. watsonii*** (Fisher) Rose, Contr. U.S. Natl. Herb. 10(3): 98. 1906. Basionym: *Caesalpinia watsonii* Fisher, Bot. Gaz. 18(4): 122. 1893. *Larrea watsonii* (Fisher) Britton, N. Amer. Fl. 23(5): 312 1930. *Hoffmannseggia gracilis* S. Watson, Proc. Amer. Acad. Arts 17: 347. 1882. 

**Type:** (Lectotype**:** designated by Fisher, 1882: Revision of the North American species of *Hoffmansseggia*. Contr. U.S. Natl. Herb. 1: 146. Voucher: Mexico, Coahuila, Sierra Madre, south of Saltillo, II-1880, *E. Palmer 275* (GH103088!).

**Distinguishing features:** Herbaceous perennial, branched from the base, prostrate or decumbent, without glands or with few scattered sessile glands. Pinnae 3–9 pairs of pinnae per leaf, plus one terminal. Leaflets 5–7 pairs per pinna. Flowers in terminal racemes, red-orange or red-salmon. Sepals with abundant short trichomes on the back. Stamen filaments with retrorse dendritic trichomes almost the entire length. Fruit 1.5–3 × 0.6 cm linear-oblong, flattened, straight, apically acute or rounded, sometimes with few glandular trichomes. 

**Representative examined material:** Coahuila: 7-VII-1995, *G. Hinton 25354* (ANSM), 5-VI-1990, *J.A. Villarreal 5686* (ANSM)**.** Nuevo León: 21-VI-1988, *E. Estrada 1550* (CFNL); 30-VI-2012, *E. Estrada 22355* (CFNL); 8-VII-2003, *E. Estrada 15802* (CFNL); 4-VIII-1999, *E. Estrada 10499* (CFNL); 27-VI-1989, *E. Estrada 1616* (CFNL). 

8-VII-2003, *C. Yen y E. Estrada 15802* (CFNL); 27-VII-1989, *E. Estrada C. 1616* (CFNL; TEX-LL); 1-VII-1999, *C. Yen y E. Estrada 10980* (CFNL).

**Comments:** Endemic to Mexico. Recorded only in Coahuila and Nuevo León, in gypsophilous and halophytic grasslands, Tamaulipan thornscrub, desert scrublands, *Pinus* and *Juniperus* forest, 370–2220 m. Outside the area in San Luis Potosí, Querétaro, Guanajuato and Hidalgo.

***Pomaria*** Cav., Icon. 5(1): 1–2, pl. 402. 1799. *Melanosticta* DC., Prodr. 2: 484. 1825. *Cladotrichium* Vogel, Linnaea 11: 401. 1837.

**Type:** *Pomaria glandulosa* Cav., Icon. [Cavanilles] 5: 2, t. 402. 1799.

Herbaceous, subshrubs or shrubs, pubescent, stems and leaves with orange or black sessile glands and dendroid (multicellular projections consisting of a vertical axis with plumose projections radiating off the main axis or at the tip) or palmate (similar to the dendroid ones but projections radiating mainly at the tip). Pinnae 1–7 pairs per leaf, plus one terminal. Leaflets 2–10 pairs per pinna, with sessile glands abaxially. Flowers in axillary or terminal racemes, yellow-light or bright-yellow. Sepals of different sizes, the lower one the largest, containing inside the stamens and pistil. The innermost petal (banner), erect and upper, with trichomes and stipitate-glandular trichomes dorsally, the 2 adjacent lateral petals are unguiculate, yellow, and dyed red basally. Ten stamens. Fruit oblong, falcate lunate to trapezoidal, with fluffy to stellate pubescence and glands, dehiscent, the valves coiling spirally after dehiscence. 

Genus with 16 species: nine in North America [4,59], seven in northern Mexico, four in South America, and three in southern Africa [4]. The next key follows [59] with few modifications.
1A.Fruit arched, its margins parallel, or moon-shaped, basally broadened, or the fruits ovate with multicellular projections consisting of a vertical axis with trichomes radiating mainly at the tip 21B.Fruit trapezoid or lanceolar-oblong in profile, with abundant cylindrical setose projections, 1–2 mm long or with multicellular projections with a short axis with a few radial, apical trichomes of different sizes 42A.Fruit arched, its margins more or less parallel; valves densely covered with trichomes, glandular-punctate***P. canescens***2B.Fruit lunate, with its dorsal margin slightly curved and the lower margin strongly inclined outwards; valves with multicellular projections ending in a variable number of palmate trichomes 33A.Stems angularly branched, slightly pilose; fruits 18–21 mm long, their valves and margins with few multicellular projections, each one presenting few terminal trichomes***P. fruticosa***3B.Stems straight or basally decumbent, hirsute, with short, coiled pilose trichomes; fruits 20–25 mm long with scattered glandular-punctate trichomes and numerous multicellular projections with abundant star-shaped apical trichomes***P. jamesii***4A.Sepals with long glandular-punctate trichomes turning black on drying, mixed with smaller peltate red or orange trichomes; fruit margins with multicellular projections up to 1 mm long***P. melanosticta***4B.Sepals with long glandular-punctate trichomes, turning black when dry, without the presence of other types of trichomes; fruit margins without multicellular projections or only with few projections 0.5–2 mm long 55A.Fruits with multicellular projections 0.5–2 mm long***P. wootonii***5B.Fruit with few short multicellular projections, 0.5 mm long***P. austrotexana***

***Pomaria austrotexana*** B.B. Simpson. Lundellia 1: 51. 1998.

**Type:** USA, Texas, Jim Hogg Co., Farm Rd 649, 11 mi N of Guerra, 23-III-1962, *S. Alvarez 7765* (Holotype: LL31188!).

**Distinguishing features:** Stems dense, pubescent, and glandular-punctate with scattered glands. Pinnae 2–3 pairs plus one terminal per leaf. Leaflets 3–5 pairs per pinna. Flowers in terminal racemes. Sepals with long glandular-punctate trichomes, the lower one the longest. The banner is dyed red with glands dorsally, and the lateral petals are glabrous or almost so. Stamens with pubescent filaments in the lower half. Fruit 2–3 × 1 cm, trapezoid or lanceolar-oblong in profile, with cylindrical projections, 0.5 mm long. 

**Representative examined material:** Tamaulipas: 25-IX-1981, *P.A. Fryxell 3724* (MEXU).

**Comments:** Endemic of southern Texas and northeastern Mexico. Rare in Mexico, only recorded at the northern end of Tamaulipas. *Pomaria wootonii* is very similar to *P. austrotexana*, but the first one has longer (0.5–2 mm) dendroid or palmate trichomes on the pods. 

***P. canescens*** (Fisher) B.B. Simpson, Lundellia 1: 56. 1998. Basionym: *Caesalpinia canescens* (Fisher) Fisher, Bot. Gaz. 18: 123. 1893. *Hoffmannseggia canescens* Fisher, Contr. U.S. Natl. Herb. 1(5): 149. 1892. *Larrea canescens* (Fisher) Britton, N. Amer. Fl. 23(5): 316. 1930.

**Type:** Mexico, Coahuila, near Saltillo, 1/15- IV-1880, *E. Palmer 269* (Holotype: US2573!).

**Distinguishing features:** Subshrub up to 1 m tall, canescent to tomentose, the trichomes retrorse, and glandular. Pinnae 2–4 pairs plus one terminal per leaf. Leaflets 4–7 pairs per pinna. Flowers in terminal racemes, yellow, occasionally dyed red. Sepals of different sizes. Fruit 2–2.6 × 0.5 cm, arched, its margins parallel, glandular-punctate, and pubescent with dendroid trichomes apically branched. 

**Representative examined material**: Nuevo León: 21-VI-2003, *C. Yen y E. Estrada 15780* (CFNL), 21-VI-2003, *C. Yen y E. Estrada 15796* (CFNL); 22-III-2003, *C. Yen y E. Estrada 15347* (CFNL); 8-VI-2003, *C. Yen y E. Estrada 15749* (CFNL). Tamaulipas: 22-VII-1970, *T Whiffin*, *J.KL: Grashoff 334* (95579LL).

**Comments:** Endemic to the north of Mexico. In northeastern Mexico, associated with Tamaulipas thornscrub and desert scrublands 350–1700 m. Outside the area, in Chihuahua, Durango, Zacatecas and San Luis Potosí.

***P. fruticosa*** (S. Watson) B.B. Simpson, Lundellia 1: 57. 1998. Basionym: *Hoffmannseggia fruticosa* S. Watson, Proc. Amer. Acad. Arts 21: 451. 1886. *Caesalpinia fruticosa* (S. Watson) Fisher, Bot. Gaz. 18: 123. 1893. *Larrea fruticosa* (S. Watson) Britton, N. Amer. Fl. 23(5): 314. 1930.

**Type:** Mexico, Coahuila, Mountains Jimulco, *C.G. Pringle 230* (Holotype: not found. Isotype: US588509!, GH62315!).

**Distinguishing features:** Dwarf shrub, angularly branched. Pinnae 1–3 pairs per leaf plus one terminal. Leaflets 2–5 pairs per pinna, with few punctate glands. Flowers in terminal racemes. Fruits 18–21 mm long, dorsally lunate, ventrally straight, pilose, and with glandular trichomes and very few tiny (0.3 mm long) palmate trichomes with few multicellular projections, each one presenting few terminal trichomes. 

**Representative examined material**: Coahuila: 21-VI-2007, *E. Estrada 20097* (CFNL; TEX); 25-VIII-1988, *J.A. Villarreal 4439*, *M.A. Carranza* (ANSM, TEX); 25-VIII-1988, *J.A. 7812 M.A. Carranza* (ANSM, TEX); 24-IX-1972, *M.C. Johnston 9504* (ANSM); 23-III-1973, *M.C. Johnston et al.*, (TEX).

**Comments:** Endemic to the north of Mexico. Recorded only in Coahuila, 1200–2100 m, in desert scrublands, rocky soils, outside the area, in Durango. Easily recognized due to its angular branching pattern.

***P. jamesii*** (Torr. & A. Gray) Walp., Repert Bot. Syst. 1: 811. 1843. Basionym: *Hoffmannseggia jamesii* Torr. & A. Gray, Fl. N. Amer. 1: 393. 1840. *Caesalpinia jamesii* (Torr. & A. Gray) Fisher, Bot. Gaz. 18: 13. 1893. *Larrea jamesii* (Torr. & A. Gray) Britton, N. Amer. Fl. 23(5): 316. 1930.

**Type:** USA, New Mexico, Sources of the Canadian, *E. James s.n.* (Holotype: NY431802!).

**Distinguishing features**: Herbaceous, perennial, hirsute, with coiled pubescence. Pinnae 2–3 pairs plus one per leaf. Leaflets 5–10 pairs per pinna, glandular punctate abaxially. Flowers in lateral and terminal racemes. Petals yellow. Fruit 2–2.5 × 1 cm, sparsely with glandular-punctate trichomes and stellate-palmate trichomes. 

**Representative examined material**: Coahuila: 27-VII-1938 E.G. *Marsh s.n.* (TEX).

**Comments:** The southern USA and north of Mexico, in northeastern Mexico, recorded only in the extreme north of the state of Coahuila, in desert scrublands.

***P. melanosticta*** S. Schauer, Linnaea 20(6): 748–749. 1847. Basionym: *Hoffmannseggia melanosticta* (S. Schauer) A. Gray, Pl. Wright. 1: 54. 1852. *Larrea melanosticta* (S. Schauer) Britton N. Amer. Fl. 25(3): 314. 1930. *Caesalpinia melanosticta* (S. Schauer) Fisher, Bot. Gaz. 18: 123. 1893. *Hoffmannseggia parryi* (Fisher) B.L. Turner. *Caesalpinia parryi* (Fisher) Eifert, in D.S. Correll & M.C. Johnston, Manual of the vascular plants of Texas 797. 1970. *Hoffmannseggia melanosticta* var. *parryi* (Fisher) Fisher, Contr. U.S. Natl. Herb. 1: 149. 1982. *Larrea parryi* (Fisher) B.L. Turner, Field & Lab. 18: 47. 1950. *Hoffmannseggia melanosticta* var. *greggii* Fisher, Contr. U.S. Natl. Herb. 1: 149. 1892. *Caesalpinia melanosticta* var. *greggii* (Fisher) Fisher, Bot. Gaz. 18: 123. 1893. 

**Type**: Mexico, Hidalgo, Zimapán, no date, *A. Aschenborn 234* (Holotype: B presumably destroyed).

**Distinguishing features:** Subshrub with a stinky aroma when fresh. Stems canescent and glandular. Pinnae 1–2 pairs plus one terminal per leaf. Leaflets 2–4 pairs per pinna. Sepals pubescent and glandular punctate of 2 types, ones flattened glandular, black (dried), the others stipitate, red or orange. Fruit 2–3.2 × 1–1.5 cm, oblique-oblong, distally curved, with scattered glandular trichomes and densely covered with red, multicellular, dendroid palmate trichomes, up to 1 mm long. 

**Representative examined material:** Nuevo León: 22-VII-1971, *H.M. Parker 532* (TEX-LL); 16-III-1973, *M.C. Johnston*, *T.L. Wendt y F. Chiang 10228b* (TEX-LL). Tamaulipas: 7-XII-1993, *A. Mora-Olivo 5041* (UAT); 10-VII-1987, *L. Hernández 2153* (UAT).

**Comments:** Southwestern Texas, Chihuahua, Coahuila and Nuevo León to Zacatecas, San Luis Potosí, Querétaro to Hidalgo. Lowlands and highlands, calcareous and gypsum soils, desert shrublands, and halophytic grasslands, 650–1800 m.

***P. wootonii*** (Britton) B.B. Simpson, Lundellia 1: 69. 1998. Basionym: *Caesalpinia wootonii* (Britton) Eifert ex Isely, Mem. New York Bot. Gard. 25(2): 51. 1975. *Larrea wootonii* Britton, N. Amer. Fl. 23(5): 315. 1930.

**Type**: Mexico, Tamaulipas, Chamal, 25-VI-1919, *E.O. Wooton s.n.* (Holotype: US2597!).

**Distinguishing features:** Subshrub up to 60 cm tall. Stems tomentose and glandular punctate. Pinnae 2–3 pairs plus one terminal per leaf. Leaflets 4–5 pairs per pinna. Sepals with long glandular-punctate trichomes, turning black when dry. Fruits with multicellular projections 0.5–2 mm long.

**Representative examined material:** Coahuila: 16-IV-1979, *J. Marroquín 3656* (ANSM), 14-VI-1987, *D. Castillo 577* (ANSM). Nuevo León: 16-V-2003, *C. Yen y E. Estrada 15653* (CFNL); 22-X-2002, *C. Yen y E. Estrada 15163* (CFNL, MEXU); 7-IX-1962, *B.L. Turner 1042 and A.M. Powell* (TEX-LL). Tamaulipas: 23-III-1999, *A. Mora-Olivo 7475* (UAT); 12-XI-1959, *J. Graham and M.C. Johnston 4655* (TEX-LL); 8-XII-1960, *M.C. Johnston 6073 and C. McMillan* (TEX-LL).

**Comments:** Endemic to northeastern Mexico. Common in calcareous and rocky soils, in Tamaulipan thornscrub, piedmont scrub, and desert scrublands, also in areas with disturbance, 250–750 m.

**Tribe Ceratonieae** Rchb., Fl. Germ. Excurs. 2(2): 544. 1832. Ceratoniinae H.S. Irwin & Barneby in R.M. Polhill & P.H. Raven, Adv. Legume System. 1: 98. 1981.

Commonly unarmed shrubs or trees. Leaves pinnate or bipinnate, imparipinnate. Flowers in racemes or panicles, bisexual or unisexual, commonly with a short hypanthium and sometimes with a basal disk. Sepals valvate to imbricate. Petals 4–6 or absent. Stamens the same, double or more than double of the petals. Fruits are linear to oblong and adaxially winged, or laterally compressed and 4-winged, with the wings paired of different sizes, 1 or several seeded, dehiscent or indehiscent. Seeds are flattened, usually separated by areas of pulpy mesocarp and pleurogram lacking [4].

**Type:** *Ceratonia siliqua* Sp. Pl.: 1026. 1753.

The tribe is composed of four genera [54], *Arcoa*, *Tetrapterocarpon*, *Acrocarpus*, and *Ceratonia*, with highly disjunct geographical distribution from Haiti and Dominican Republic, Madagascar, tropical areas in south-east Asia, and north-eastern Africa. 

Only one genus and one species recorded in northeastern Mexico, *Ceratonia siliqua*.

***Ceratonia*** L., Sp. Pl. 2: 1026. 1753.

**Type:** (Lectotype: designated by Jafri, Fl. Lybia 60: 5. 1978: Voucher: LINN-1239.1).

Shrubs or trees, dioecious, evergreen. Leaves paripinnate. Leaflets 3–6 pairs per leaf, leathery, elliptical-oval to suborbicular, adaxially shiny, abaxially opaque. Flowers are borne on old branches, in racemes, unisexual. Perianth cup-shaped, 5–7 sepals. Stamens 2–8, free, the ones of female flowers are without pollen, with long filaments and long anthers. Receptacle disc is well developed. Fruit linear, compressed, thick, indehiscent, leathery, black-reddish, pericarp fleshy-fibrous, sweet, internally pulpy. Seeds are arranged transversely.

The genus is represented by 2 species: *C. oreothauma* [61], restricted to the south of the Arabian Peninsula (Oman and Yemen) and Somalia (eastern Africa), and *C. siliqua*, widely distributed worldwide. 

***Ceratonia siliqua*** L., Sp. Pl. 2: 1026. 1753.

**Type:** (Lectotype: designated by Jafri, Fl. Lybia 60: 5. 1978: Voucher: LINN-1239.1).

**Distinguishing features:** Tree or shrub, 5–8 m high, leaflets 2–6 × 1.8–5 cm, leathery. Male flowers with 5 free stamens and a rudimentary ovary; female flowers with 5 small staminodes. Fruit 8–14 × 1.0–1.5 cm, linear, compressed, thick, indehiscent, leathery, straight or slightly curved. 

**Representative examined material:** Nuevo León: 15-IV-02, *E. Estrada 14532* (CFNL); 16-V-1982, *G. Alanís s.n.* (CFNL). Tamaulipas: 27-III-2024, *A. Mora-Olivo s.n.* (UAT).

**Comments**: Native to the Mediterranean region. In northeastern Mexico, it is used as ornamental. Known as *algarrobo*.

**Tribe Gleditsieae** Nakai, Chosakuronbun Mokuroku [Ord. Fam. Trib. Nov.]: 253. 1943.

**Type**. *Gleditsia* J. Clayton in Linnaeus, Gen. Pl. 5: 476. 1754.

Trees, deciduous or evergreen. Branches unarmed or armed with simple or branched thorns or unarmed. Leaves pinnate, bipinnate. Flowers in racemes or panicles. Flowers bisexual or unisexual, androdioecious or dioecious, small, white, greenish or violet, commonly 5-merous. Calyx gamosepalous. Petals are small. Stamens 6–10, free. Ovary sessile or stipitate. Fruit flattened to turgid, papery, leathery or woody, dehiscent, dehiscent or indehiscent, sometimes internally pulpy. Seeds are few to 40, flattened to sub-cylindric, orbicular, elliptic to ovoid.

Tribe is composed of three genera, *Gleditsia*, *Gymnocladus*, and *Amtiza*, and 20 species [61] distributed in temperate and subtropical areas of North and South America, Asia, and South Africa. Only one genus and one species recorded in northeastern Mexico, *Gleditsia triacanthos*.

***Gleditsia*** J. Clayton in Linnaeus, Gen. Pl. 5: 476. 1754. *Melilobus* Mitch., Diss. Gen. Pl.: 37. 1769. *Asacara* Raf., Neogenyton: 2. 1825. *Garugandra* Griseb., Abh. Königl. Ges. Wiss. Göttingen 24: 96. 1879. *Caesalpiniodes* Kuntze, Rev. Gen. 1: 166. 1891. *Pogocybe* Pierre, Fl. Cochinch.: t. 392B. 1899. 

Shrubs or trees, commonly dioecious, deciduous. Trunks and branches are sometimes armed with thorns, simple or branched. Branches are flexouse. Leaves alternate or apparently fasciculated, pinnate, or bipinnate. Leaflets with entire or crenate margin. Flowers axillary, in racemes or panicles, unisexual, with separate sexes or polygamous. Staminate inflorescences with abundant congested flowers. Calyx 3–5-merous. Petals as many as the teeth of the calyx, greenish or light yellow, inserted on the margin of the throat. Pistillate flowers with 4–10 staminodes and abortive anthers. Fruit elliptical, compressed, leathery, indehiscent or dehiscent, septate or continuous. 

The genus of 13 species is in northeastern North America, eastern and southeastern Asia, China, Burma, and Vietnam [5,62,63].

One species recorded in northeastern Mexico, *Gleditsia triacanthos*.

***Gleditsia triacanthos*** L., Sp. Pl. 2: 1056. 1753. Basionym: *Acacia triacantha* Hochst. ex A. Rich., Tent. Fl. Abyss. 1: 244. 1847. *Caesalpiniodes triacanthum* (L.) Kuntze, Revis. Gen. Pl. 1: 167. 1891. *Gleditsia ferox* Desf., Hist. Arb. ii. 247. Rehder. *Gleditsia triacanthos* L. var. *brachycarpos* Michx., Fl. Bor.-Amer. (Michaux) 2: 257. 1803. *Gleditsia triacanthos* var. *aquatica* (Marshall) Castigl., Viagg. Stati Uniti 2: 249. 1790. *Gleditsia triacanthos* var. *bujotii* Rehder in L.H.Bailey, Cycl. Amer. Hort. [L.H.Bailey] 2: 650. 1900. *Gleditsia triacanthos* var. *horrida* Aiton, Hort. Kew. [W. Aiton] 3: 444. 1789. *Gleditsia triacanthos* var. *inermis* (L.) Castigl., Viagg. Stati Uniti 2: 249. 1790. *Gleditsia triacanthos* var. *laevis* Sudw., Bull. Div. Forest. U.S.D.A. 14: 254. 1897. *Gleditsia triacanthos* var. *monosperma* Aiton, Hort. Kew. [W. Aiton] 3: 444.1789. *Gleditsia triacanthos* var. *macrocarpos* Michx., Fl. Bor.-Amer. (Michaux) 2: 257. 1803. *Gleditsia triacanthos* var. *polysperma* Aiton, Hort. Kew. [W. Aiton] 3: 444. 1789.

**Type***:* (Designated by Reveal, Regnum Veg. 127: 49 (1993). Voucher: Herb Clifford 489, Oidea No. 12A). 

**Distinguishing features:** Shrub or tree, up to 8 m tall, stems and branches mostly armed with branched spines. Leaves pinnate or bipinnate. Pinnae 4–8 pairs per leaf. Leaflets 11–25 per pinna, margin entire or slightly crenulate. Flowers axillary, in racemes. Calyx 5-merous, the sepals free. Petals 4–5, longer than calyx. Stamens 10, free. Fruit 10–35 cm long, flattened, linear-oblong, indehiscent, dark brown. 

**Representative examined material:** Coahuila: 20-IV-2011, *J.A. Encina 2872* (ANSM). 4-VIII-1984, *M.A. Carranza 150* (ANSM). Nuevo León: 16-IV-01, *C. Yen y E. Estrada 12378* (CFNL); 10-IV-2008, *E. Estrada 20170* (CFNL); 8-II-2009, *E. Estrada 20736* (CFNL). Tamaulipas: 6-VIII-1985, *E.M. Marsh 2006* (TEX-LL); 15-VI-1985, *O. Briones 1795* (CFNL); 27-IV-1988, *E. Estrada 1417* (CFNL). 

**Comments:** Recorded from the municipality of Villaldama [64] and the municipality of Juarez (Cerro La Silla), both in Nuevo León, associated with Quercus, *Juniperus* and *Fraxinus* forest. In the Sierra de San Carlos [49], in Tamaulipas, 950–1300 m, and from Sierra Santa Rosa (municipality of Muzquiz), in Coahuila, inhabiting *Juniperus*–*Quercus* forest.

**Tribe Schizolobieae** Nakai, Chosakuronbun Mokuroku [Ord. Fam. Tribe. Nov.]: 251. 1943. 

**Type.** *Schizolobium* Vogel, Linnaea 11: 399. 1837.

Trees or shrubs, 3–40 m tall. Stems and branches are commonly unarmed, rarely armed with thorns or modified stipules. Leaves bipinnate. Pedicels usually joined. Flowers axillary, in racemes or panicles, yellow, orange, whitish-pink or red, bisexual or unisexual. Sepals 5, unequal, free or partially fused, usually strongly reflexed. Petals 1–5, clawed, usually wrinkled, erosed marginally. Ten stamens. Fruit, linear or linear-oblong to spathulate, flat, terete or thickened, the valves papery, coriaceous or woody, sutures thickened or narrow-winged, indehiscent or inertly or elastically dehiscent. Seeds 1–many. 

Tribe is composed of 8 genera and 42–43 species [64], distributed in tropical forests and tropical savannas. Two genera are recorded in northeastern Mexico: *Parkinsonia* and *Delonix*.
1A. Plants armed with thorns; pinnae 1 pair per leaf; flowers yellow; fruit 15 cm long or sorter, oblong, linear-oblong, flattened, cylindrical or torulous***Parkinsonia***1B.Plants unarmed; pinnae 10–25 pairs per leaf; flowers red to red-orange; fruits 20–50 cm long, laterally compressed***Delonix***

***Delonix*** Raf., Fl. Tellur. 2: 92. 1836. *Aprevalia* Baill., Bull. Mens. Soc. Linn. 428. 1884.

**Type**: *Delonix regia* (Boj. ex Hook.) Raf. Fl. Tellur. 2: 92. 1837. *Delonix regia* var. *flavida* Stehlé, Bull. Mus. Natl. Hist. Nat., sér. 2, 18: 186. 1946. *Poinciana regia* Boj. ex Hook.) Bot. Mag. 56: pl. 2884. 1829.

Small trees, up to 6–7 m tall, evergreen or deciduous. Pinnae 10–25 pairs per leaf. Leaflets 10–40 pairs per pinna. Flowers in terminal racemes, large and showy, scarlet or red-orange. Calyx gamosepalous, 5-lobed, equal or half the length of the petals. Corolla with 5 petals, unguiculate, reflexed with age. Stamens 10, free, exert, filaments of different sizes, hairy. Fruit pendulous, broadly linear, sometimes greater than 50 cm, flattened, long, woody, septate between the seeds, persistent, dehiscent or indehiscent. Seeds elliptical.

Genus are represented by 11 species, nine of which are endemic to Madagascar, and also in Africa, Arabia, and India [4]. One species in northeastern Mexico, *Delonix regia*.

***Delonix regia*** (Bojer ex Hook.) Raf. Raf., Fl. Tellur. 2: 92. 1837. Basionym: *Poinciana regia* Bojer ex Hook. in Bot. Mag. 56: t. 2873. 1829.

**Type**: Martinique (Martinique), 14-VII-1939, *M. Stehlé*, *H. Stehlé 4534* (Holotype: US00376297). 

**Distinguishing features:** Trees, 4–8 m tall. Leaves bipinnate, pinnae 11–35 pairs per leaf. Leaflets 28–40 pairs per pinna. Flower in axillary racemes. Calyx 5-merous, sepals much shorter than petals. Corolla of 5 red, free, long, unguiculate petals. The banner has yellow tones and different shapes and colors of the other petals. Fruit 20–50 cm long, oblong, multi seeded. 

**Representative examined material:** Nuevo León: 21-VII-87, *E. Estrada 661* (CFNL). Tamaulipas: 2-V-1985, *C.G. Romo 457* (UAT).

**Comments:** In northeastern *Delonix regia*, known as *framboyán* or *flamboyán*, it is widely used as an ornamental species in public and private gardens; its fruits are called “sonajas” (rattles) and used as musical instruments.

***Parkinsonia*** L., Sp. Pl. 1: 375. 1753. *Cercidium* Tul., Arch. Mus. Hist. Nat. 4:133. 1844. *Cercidiopsis* Britton et Rose, N. Amer. Fl. 23: 306. 1930.

**Type**: *Parkinsonia aculeata* L., Sp. Pl. 1: 375. 1753.

Shrubs or trees up to 15 m tall. Bark green. Branches are armed with two types of thorns, ones short, due to modification of the stipules, and the other ones long, due to modification of petiole and rachis. Leaves with the rachis long and dorsoventrally flattened or cylindrical and short, fasciculate, or alternate. Pinnae 1–3 pairs per leaf. Leaflets are few or numerous, generally deciduous. Flowers are axillary and solitary, in racemes or corymbs, zygomorphic, yellow. Calyx campanulate, 5-merous, yellowish. Corolla has 5 petals, is free, and sometimes the banner has reddish tones. Stamens 10, free. Fruit oblong, linear-oblong, flattened, cylindrical, or torulous, constricted between the interseminal spaces. 

A genus of 11–12 species [4], from the central USA, through Mexico, Venezuela, Ecuador to Argentina, and also in Africa. In northeastern Mexico, two species were recorded.
1A.Leaves, 15–40 cm long, subsessile or with a tiny petiole ending in a thorn; rachis of pinnae flattened; fruit torulous***P. aculeata***1B.Leaves less than 15 cm long, petiole evident, cylindrical; fruit laterally flattened 22A.Ovary and fruits glabrous ***P. texana*** var. ***macra***2B.Ovary and fruits pubescent***P. texana*** var. ***texana***

***Parkinsonia aculeata*** L., Sp. Pl. 1: 375. 1753. Basionym: *Parkinsonia thornberi* M.E. Jones, Contr. W. Bot. 12: 12. 1908.

**Type:** (Lectotype: Designated by Polhill, Fl. Mascaraignes 80: 5. 1990. Voucher: Linnaeus Hort. Cliff.: t. 13. *Parkinsonia* (1738).

**Distinguishing features:** Branches armed with spines. Stipules are spinescent and located at both sides of the spinescent rachis. Leaves bipinnate (resembling two simple pinnate leaves). Pinnae 1 pair per leaf, each pinnae originating from a short rachis transformed into a spine, leaf rachis flattened. Leaflets are numerous and caducous. Pedicels articulated. Fruit 5–14 × 0.7–0.9 cm, cylindrical, constricted between seeds, longitudinally striated. 

**Representative examined material:** Coahuila: 3-X-2008, *J. Valdés 3129* (NSM), 12-V-2005, *J.A. Encina 3691* (ANSM). Nuevo León: 20-VII-02, *C. Yen y E. Estrada 14969* (CFNL); 19-VII-02, *C. Yen y E. Estrada 14923* (CFNL); 13-IV-02, *C. Yen y E. Estrada 14526* (CFNL); 16-IV-01, *C. Yen y E. Estrada 12489* (CFNL). Tamaulipas: 13-IV-1962, *M. Domínguez M. 8280* (UAT); 20-XI-1987, *M. Martínez 1604* (UAT); 23-III-1993, *J.L. Mora-López 348* (UAT); 27-VI-1985, *M. Martínez 695* (UAT). 

**Comments:** From the south of the USA to South America. Frequent but not abundant in Tamaulipan thornscrub, piedmont scrub, and in desert scrublands, 290–1750 m. Widely used for manufacturing poles, charcoal, firewood, and household goods. Known as *retama*.

***P. texana*** (A. Gray) S. Watson var. ***macra*** (I.M. Johnst.) Isely, Mem. New York Bot. Gard. 25(2): 175. 1975. Basionym: *Cercidium macrum* I.M. Johnst., Contr. Gray Herb. 70: 64. 1924. *Cercidium texanum* A. Gray, Smithsonian Contr. Knowl. 3(5): 58. 1852. *Parkinsonia texana* (A. Gray) S. Watson, Proc. Amer. Acad. Arts 11: 136. 1876.

**Type:** Mexico, Nuevo León, Mesas near Monterrey, *C.G. Pringle 2537* (Holotype: GH53330!).

**Distinguishing features:** Shrub or tree, up to 7 m tall, bark green, the branches flexuose, the nodes armed with small, straight, paired prickles. Leaves bipinnate, less than 15 cm long, petiole evident, cylindrical. Pinnae 1–2 pairs per leaf. Leaflets 3–4 pairs per pinna. Flowers in short racemes. Ovary glabrous. Fruit 5–8 × 1–1.5 cm, oblong, flattened, glabrous.

**Representative examined material:** Coahuila: 28-IV-1987, *A. Rodríguez 818* (ANSM), 4-V-1989, *J.A. Villarreal 4841* (ANSM). Nuevo León: 5-VII-2001, *C. Yen y E. Estrada 12887* (CFNL); 5-VII-2001, *C. Yen y E. Estrada 12855* (CFNL). 7-VII-2001, *C. Yen y E. Estrada 12988* (CFNL). 16-V-2003, *E. Estrada 15649* (CFNL). Tamaulipas: 23-III-1999, *A. Mora-Olivo 7473*.

**Comments:** In Tamaulipan thornscrub, sandy soils 125–550 m, also in southern Texas and San Luis Potosí. It is used as a tree shade for livestock. Used as ornamental, for charcoal, and for manufacturing tool handles, machetes, shovels, and talaches. Known as *palo verde*. 

***P. texana*** (A. Gray) var. ***texana***, S. Watson, Proc. Amer. Acad. Arts. 11: 136. 1876. Basionym**:** *Cercidium texanum* A. Gray, Smithsonian Contr. Knowl. 3(5): 58. 1852.

**Type:** USA, New Mexico, Western Texas, El Paso, *C. Wright 149* (Holotype: not found. Isotype P3327362!).

**Distinguishing features: S**hrub, up to 2 m tall, branched from the base, armed with small, solitary nodal spines. Pinnae 1–2 pairs per leaf. Leaflets 1–3 pairs per pinna. Ovary densely pubescent. Fruit 3–6 × 0.8–1 cm, flattened, straight, pubescent.

**Representative examined material:** Coahuila: 7-VII-2001, *J. Luna*, *M. González*, *C. Yen y E. Estrada 12988* (CFNL). Nuevo León: 5-VII-2001, *C. Yen y E. Estrada 12887* (CFNL); 5-VII-2001, *C. Yen y E. Estrada 12855* (CFNL); 16-V-2003, *C. Yen y E. Estrada 15649* (CFNL). Tamaulipas: 24-III-1985, M. Martínez 201 (UAT); 31-X-1984, B.D. Fuentes, R. Diaz, R. Aguilar 590 (UAT); 22-III-1985, M Yanez 31 (UAT).

**Comments:** Found in Tamaulipan thornscrub, sandy and stony soils. Also found in southern Texas. Easily differentiated from the var**.** *macra* by its lower size and pubescent ovary. 

**Tribe Cassieae** Bronn, Form. Pl. Legumin.: 130. 1822. 

**Type: *Cassia*** L., Sp. Pl.: 376. 1753.

Herbaceous shrubs or trees. Leaves pinnate, paripinnate. Flowers are mainly in racemes, rarely cauliflorous, and rarely 1–2. Bractlets 0–2. Flowers bisexual, zygomorphic, yellow. Sepals 5, imbricate. Petals 5, the banner is almost always interior in the bud. Stamens 2–10, when 10, all fertile or three adaxial ones staminodes. Anthers dehiscent by subterminal pores or slits, sometimes dehiscent basally. Fruit cylindrical, angulate or flattened, sometimes winged, dehiscent along one or both sutures, inertly or elastically dehiscent, or indehiscent, seeds 1 to many.

Tribe with 7 genera with pantropical distribution, extending to arid and temperate areas [65]. Three genera are recorded in northeastern Mexico: *Cassia*, *Chamaecrista*, and *Senna*.
1A.Filaments of 3 stamens sigmoid, several times larger than their anthers, the remaining 7 filaments straight and shorter; pedicels 2-bracteolate at or just above base; fruit indehiscent***Cassia***1B.Filaments of all stamens straight or curved, shorter or less than 2 times the length of their anthers; pedicels ebracteate or 2-bracteate, the bracteoles inserted in the middle portion or distally; fruit dehiscent or indehiscent 32A.Flowers without staminodes; pedicels with 2 bracteoles; fruit explosively dehiscent, valves coiling or twisting after dehiscence***Chamaecrista***2B.Flowers with staminodes; pedicels without bracteoles; fruit dehiscent or indehiscent (non-explosively) through one or both sutures***Senna***

***Cassia*** [Tourn.] L., Sp. Pl. 376. 1753. *Cathatocarpous* Pers., Syn. pl. 1: 459. 1805. *Bactyrilobium* Willd. Enum. hort. berol. 439. 1809. *Cassia* sect. *Fistula* DC ex. Colladon, L., Hist. Cass. 83.1809. *Cassia* sensu Link, Handbruch 2: 138. 1831. *Cassia* sensu Irwin & Barneby in Polhill & Raven. 1981. P. 105.

**Type:** *Cassia fistula* L. Sp. Pl. 1: 377. 1753 (CT, designated by Randall, Taxon 37: 975 (1988), Taxon 42: 689 (1993).

Trees or shrubs. Leaflets, small or large. Flowers in terminal or axillary racemes, yellow. Pedicels bi-bracteolate at or just above the base. The banner is of a different color than the rest of the petals. Ten stamens, heteromorphic, three of them with their filaments proximally sigmoid and curved distally, frequently dilated in the external or distal curvature. The remaining seven stamens are smaller, without sigmoid filaments; six of these are fertile, and the remaining one is a staminode. Ovary stipitate. Fruit indehiscent, pendulous, linear, cylindrical or laterally compressed, woody, internally septate. 

A genus of 30 species, circumtropical. Thirteen species in America, 12 of them in the Amazon region; one species endemic to Mexico [4]; and the rest in Africa, Madagascar, Thailand, India, Myanmar, and Australia.

***Cassia fistula*** L., Sp. Pl. 1: 377–378. 1753. Basionym: *Bactyrilobium fistula* (L.) Willd., Enum. Pl. [Willdenow] 1: 440. 1809. *Cassia fistuloides* Collad., Hist. Nat. Méd. Casses 87, t. 1816. *Cathartocarpus fistula* (L.) Pers., Syn. Pl. [Persoon] 1: 459. 1805.

**Type**: (Lectotype: designated by Ali, Fl. W. Pakistan 54: 12. 1973. Voucher (LINN-528.15). 

**Distinguishing features:** Tree up to 8 m tall. Leaves 15–62 cm long. Leaflets 2–7 pairs per leaf, distally accrescent, ovate, bicolored. Flowers in pendulous racemes. Sepals soon reflexed. Fruit 30–60 × 1.5–2.5 cm, cylindrical, straight, black, glabrous, internally septate, long persistent.

**Representative examined material:** Nuevo León: 20-V-2003, *C. Yen y E. Estrada 15658* (CFNL). Tamaulipas: 3-I-2024, *A. Mora-Olivo s.n.* (UAT)

**Comments:** Species native to India, widely cultivated in many countries. Used as ornamental in northeastern Mexico. Known as “lluvia de oro” (golden rain).

***Chamaecrista*** (L.) Moench, Methodus: 272. 1794. *Cassia* [infragen. unranked] *Chamaecrista* L., Sp. Pl. 1: 379. 1753. *Cassia* sect. *Chamaecrista* (L.) DC., Hist. Nat. Méd. Casses 24, 118.1816. *Sooja* Siebold, Verh. Batav. Genootsch. Kunst. 12: 56. 1830. *Disterepta* Raf., Sylva Tellur.: 126. 1838. *Hepteireca* Raf., Sylva Tellur.: 126. 1838. *Dialanthera* Raf., Sylva Tellur.: 127. 1838. *Xamacrista* Raf., Sylva Tellur.: 127. 1838. *Nictitella* Raf., Sylva Tellur.: 128. 1838. *Ophiocaulon* Raf., Sylva Tellur.: 129. 1838. *Cassia* subg. *Lasiorhegma* Vogel ex Benth., Fl. Bras. 15(2): 129. 1870. 

**Type:** *Chamaecrista nictitans* (L.) Moench, pl. hort. bot. Marburg. 272. 1794. 

Herbaceous shrubs or trees. Leaves pinnate. Leaflets opposite. Pedicels bibracteolate, near or above the middle. Flowers in axillary or subterminal racemes, 1-multiflorous, yellow. Calyx dialysepalous. Corolla dialypetalous. Petals 5. Stamens 2–10, isomorphic or heteromorphic, without staminodes. Filaments of all stamens are straight or curved, and the anthers are longer than the filament. Fruit compressed, papyraceous or leathery, explosively dehiscent, the valves coiling or twisting after dehiscence. 

A genus of 330 species in tropical and subtropical areas. Most of the species (266) are in America, with greater diversification in South America [4], 36 in Africa, and 12 native to Australia.
1A.Leaflets 1 pair per leaf***Ch. diphylla***1B.Leaflets 2-more pairs per leaf22A.Sepals obtuse; flower buds subglobose to broadly obtuse-ovoid32B.Sepals acute or acuminate; flower buds ovoid-acuminate or acute43ALeaflets 3–6 pairs, 5–11 mm largo***Ch. greggii*** var. ***greggii***3B.Leaflets 2 pairs, rarely 4 pairs, 11–17 mm long***Ch. greggii*** var. ***potosina***4A.Peduncles axillary ***Ch. flexuosa*** var. ***texana***4B.Peduncles appearing supra-axillary, attached to stem internode 0.5–1 mm or more from the point of origin 55A.Petals 3.5–6.5 mm long***Ch. nictitans*** ssp.*** disadena*** var. ***pilosa***5B.Petals 8–19 mm long66A.Stipules 7–17 mm long; leaves 3–12 cm long; leaflets 14–45 pairs***Ch. rufa*** var. ***rufa***6B.Stipules 2–7 mm long; leaves 1.5–4 cm long; leaflets 10–19 pairs77A.Perennial plants***Ch. chamaecristoides*** var. ***chamaecristoides***
7B.Annual plants***Ch. chamaecristoides*** var. ***cruziana***

***Chamaecrista chamaecristoides*** (Colladon) Greene var. ***chamaecristoides***, Basionym: *Cassia chamaecristoides* Colladon, 1816, 1.c. sense strict. Mill. Dict. 17. *Cassia chamaecrista* Mill., Gard. Dict. Ed. 8. 1768 (non Linneus, 1753). *Cassia cinerea* Cham. & Schltdl.

**Type:** Mexico, Veracruz, *Schiede 711* (Holotype: not found, Isotype: K478165!).

**Distinguishing features:** Herbaceous, perennial, psamophilous. Stems diffuse, decumbent, up to 2 m long, pubescent. Leaflets 10–19 pairs per leaf. Flowers in racemes, supra axillary, attached to stem internode almost 1 mm from the point of origin. Anthers red. Fruit 2.5–6.5 × 0.5–0.6 cm, brown to black. 

**Representative examined material:** Tamaulipas: 10-VI-1997, *A. Mora-Olivo 7287* (UAT); 23-V-1985, *B.D. Aguilar 761* (UAT); 3-X-1984, *D. Baro 489* (UAT); 29-I-1923 to 17-X-1923, *C.D. Mell s.n.* (NY 554138!); 31-I-1975 to 11-XI-1975, *A.A. Lasseigne 4986* (NY554155!); 11-I-1898 to 24-VIII-1898, *C.G. Pringle 6805* (554140!); 1/8-II-1910, *E. Palmer 254* (NY554139!).

**Comments:** Endemic to Mexico, Gulf Coast (Jalisco and Michoacán), and Pacific Coast, from Tamaulipas, Veracruz, Tabasco to Campeche.

***Ch. chamaecristoides* var*. cruziana*** (Britton & Rose) H.S. Irwin & Barneby, Mem. New York Bot. Gard. 35(2): 773. 1982. Basionym: *Chamaecrita cruziana* Britton & Rose, N. Amer. Fl. 23(5): 288. 1930.

**Type***:* Mexico, Veracruz, Pueblo Viejo, 2 km S of Tampico, *E. Palmer 534* (Holotype: NY4044!).

**Distinguishing features:** Very similar to the var. *chamaecristoides*, but this variety has an annual growth habit and narrower fruits (0.4–0.5 cm) (measurements from morphological data taken from [8]).

**Representative examined material:** Tamaulipas: 7-V-1984, *D. Baro 144* (UAT), Tamaulipas-Veracruz border: 1/2-VI-1910, *E. Palmer 534* (MO-126298!). 

**Comments:** Recorded only in the state of Tamaulipas, in coastal dunes and sandy soils, forming patches. Also found on the coasts of Texas and Veracruz, adjacent to Tamaulipas (Pánuco city).

***Ch. diphylla*** (L.) Greene, Pittonia 4: 28. 1899. Basionym: *Cassia diphylla* L., Sp. Pl. 376. 1753. 

**Type**: Habitat in India. RCN: 2957. Lectotype (designated by Irwin & Barneby in Mem. New York Bot. Gard. 35: 891. 1982): Herb. Linn. No. 528.1 (LINN).

**Distinguishing features:** Monocarpic or short-lived perennial. Stems 15–100 cm long, simple or branched, radiating from a root-crown or erect. Stipules are long or a little shorter than internodes. Leaflets 1 pair per leaf, 1–4 × 0.3–2 cm. Fruit 2.5–5.2 × 0.4–0.7 cm.

**Representative examined material:** Tamaulipas: 14-XI-1998, *A. Mora-Olivo 2023* (UAT)**.**

**Comments:** Recorded only in the state of Tamaulipas, it is the only species of *Chamaecrista* with 1 pair of leaflets in northeastern Mexico. Outside the area, reported from Veracruz to Chiapas and Oaxaca, Central America, and from Colombia to Brazil. 

***Ch. flexuosa* var. *texana*** (Buckl.) H.S. Irwin & Barneby, Mem. New York Bot. Gard. 35(2): 700. 1982. Basionym: *Cassia texana* Buckl., Proc. Acad. Nat. Sci. Philad. 1861: 452. 1861. *Chamaecrista texana* (Buckl.) Pennell. Bull. Torr. Club 44: 344. 1917.

**Type**: USA, Texas, Bastrop County, non date, *S.B. Buckley s.n.* (Holotype: not found, Isotype: GH53034!).

**Distinguishing features:** Perennial herb. Stems quadrangular in first stages, humifuse, up to 0.5 m long. Leaflets 9–25 pairs per leaf. Peduncles are axillary. Sepals acute or acuminate. Fruit 2.5–7 × 0.5 cm, oblong, ascending.

**Representative examined material:** Tamaulipas: 15-IV-1999, *A. Mora-Olivo 7526* (UAT); 1915, *Buckl. & Pennel? 2427* (NY); *R.L. Crocket 6487* (Laredo Texas, border county with Laredo, Tamaulipas Mexico (NY).

**Comments:** Recorded only in the state of Tamaulipas, in coastal plains, sometimes in disturbed places, 0–300 m. From south Texas through Tamaulipas to Yucatán in the Gulf Coast, also in the Pacific Coast, from Colima to Guerrero.

***Ch. greggii*** (A. Gray) Pollard **ex.** A. Heller var. ***greggii.*** Basionym**:** *Cassia greggii* A. Gray Smithsonian Contr. Knowl. 3(5): 59. 1852. 

**Type:** Mexico, Nuevo Leon, Monterrey, W of Rinconada and E of Corralbo [Cerralvo], 25/28-V-1847, *J. Gregg*, *s.n.* (Holotype: GH306868).

**Distinguishing features:** Shrub, erect, intricate, up to 1.9 m tall. Leaflets 3–6 pairs per leaf, 5–11 mm long. Bracts 2 on the proximal portion of the pedicel. Sepals obtuse. Fruit 2–5 × 0.5–0.7 cm, oblong, flattened, brown, elastically dehiscent.

**Representative examined material:** Coahuila: 11-IX-1997, *M.A. Carranza 2741* (ANSM), 27-XI-2001. *M.A. Carranza 3571* (ANSM). Nuevo León: 7-VII-2001, *C. Yen y E. Estrada 12989* (CFNL); *J.A. Villarreal* et al. *9140* (ANSM, MEXU). 23-VII-1999, *E. Estrada* et al. *11245* (CFNL); 15-IX-2001, *C. Yen y E. Estrada 11721* (CFNL). 24-VI-2001, *C. Yen y E. Estrada 12827* (CFNL); 5-VII-2001, *C. Yen y E. Estrada 12847* (CFNL); 9-XI-1993, *Hinton* et al. *23872* (MEXU). Tamaulipas: 23-III-1999, *A. Mora-Olivo 7468* (UAT); 27-Vi-1985, *M. Martínez 676* (UAT); 25-XI-1984, *L. Hernández 1256* (UAT).

**Comments:** Endemic to northeastern Mexico and southern Texas. In thornscrub and desert scrub, 350–1500 m.

***Ch. greggii*** (A. Gray) Pollard **ex.** A. Heller var***. potosina*** (Britton & Rose) Irwin & Barneby, Phytologia 44(7): 500. 1979. *Chamaecrista potosina* Britton & Rose, N. Amer. Fl. 23(5): 283. 1930. *Cassia potosina* (Britton & Rose) Standl.. Field Mus. Bot. 22: 78. 1940.

**Type:** Mexico, XI-1910, Minas de San Rafael [San Luis Potosí], *C.A. Purpus 4832* (Holotype: NY4119!).

**Distinguishing features:** Shrub, erect, unarmed up to 1.5 m tall, very similar to var. *greggii*, but the var. *potosina* is always only with 1–2 pairs of leaflets per leaf, rarely up to 4 pairs, if so, the leaflets longer, 1.1–1.7 cm long.

**Representative examined material:** Tamaulipas: 7-VII-1985, *P. Hiriart 910* (UAT); 29-VI-1985, *L. Hernández 1498* (UAT); 22-X-1999, *A. Mora-Olivo 7717* (UAT); 12-VIII-1941, *L.R. Stanford 871* (554253NY!).

**Comments:** Endemic to northeastern Mexico. Recorded only in the south of the state of Tamaulipas, desert scrublands, 1600–2400 m. Outside the area, in San Luis Potosí. 

***Ch. nictitans*** ssp.** *disadena*** var. ***pilosa*** Irwin & Barneby, Mem. New York Bot. Gard. 35(2). 829. 1982. Basionym: *Chamaecrista riparia* var. *pilosa* Benth. in Martis Fl. Bras. 15(2): 174. 1870. *Cassia propinqua* Kunth, Nov. Gen. & Sp. 6: 369. 1824. *Chamaecrita rekoi* Britton & Rose, N. Amer. Fl. 23(5): 289. 1930. *Chamaecrista tonduzii* Britton & Rose, N. Amer. Fl. 23(5): 290. 1930. *Chamaecrista fenixensis* Britton & Rose, N. Amer. Fl. 23(5): 291. 1930. *Chamaecrista comayaguana* Britton & Rose, N. Amer. Fl. 23(5): 291. 1930. *Chamaecrista conzatii* Britton & Rose, N. Amer. Fl. 23(5): 293. 1930. *Chamaecrista salvadorensis* Briton ex Britton & Rose, N. Amer. Fl. 23(5): 293. 1930. *Chamaecrista stenocarpoides* Britton ex Britton & Rose, N. Amer. Fl. 23(5): 293. 1930. 

**Type:** Brazil, Mina Gerais; Lagoa Santa, 26-II-1866, *Warming 207* (Presumed isotype: The American Cassinae A synoptical revision of the Leguminosae tribe Cassieae subtribe Cassinae in the New World, H.S. Irwin and R.C. Barneby, Mem. New York Bot. Gard. 35(2): 829. 1982: Vouchee: NY3893!).

**Distinguishing features:** Annual or suffrutescent, erect or decumbent and ascending, up to 2 m tall, pubescent with trichomes and setae. Leaflets 12–29 pairs per leaf. Flowers in supraciliary peduncles. Petals 3.5–6.5 mm long. Fruit 2.5–7 × 2.5–4.5 cm, pilose. 

**Representative examined material:** Tamaulipas: 22-V-1985, *B.D. Aguilar 744* (UAT); 25-VI-1984, *B.D. Aguilar 339* (UAT); 14-VI-1994, *J.L. Mora-López 529* (UAT); 23-IX-1985, *M. Yanez 462* (UAT); 23-VIII-2008, *A. Mora-Olivo 11731* (UAT); 31-VIII-1957, *H.S. Irwin 1392* (TEX97336!); 22-IX-1985, *M. Yáñes 462* (TEX973381!). 

**Comments:** Recorded only in the state of Tamaulipas. Widely distributed, from northeastern Mexico to Bolivia and Brazil.

***Ch. rufa*** (M. Martens & Galeotti) Britton & Rose var. ***rufa.*** Basionym: *Cassia rufa* M. Martens & Galeotti, Bull. Acad. Roy. Sci. Brux. 10(2): 306. 1843. *Chamaecrista jalapensis* Britton ex Britton & Rose, N. Amer. Fl. 23(5): 274. 1930. *Cassia chamaecrsita* sensu Sessé & Moc., Fl. Mex. Ed 2. 99. 1894.

**Type:** Mexico, Mirador, *H. Galeotti 3311* (Holotype: BR511981!).

**Distinguishing features:** Subshrub, up to 1.8 m tall. Stems and leaves with short incurved trichomes and setae. Stipules erect, up to 0.7–1.7 cm long. Leaves 3–12 cm long. Leaflets 18–32 pairs per leaf. Petals 8–9 mm long. Fruit 3–6 × 0.4–0.5 cm, puberulent. Measurements from morphological data taken from [8].

**Representative examined material:** Tamaulipas: 1932, *H.W. von Rozynski 599* (554440NY!).

**Comments:** Recorded only in Tamaulipas, rare, in oak to oak–pine forest, and roadside, 1500 m. Outside the area, from Veracruz to Costa Rica.

***Senna*** Mill., Gard. Dict., Abr. (ed. 4), 3. 1754. *Chamaecassia* Link, Handbuch 2: 139. 1831. *Chamaefistula* (DC. ex Collad.) G. Don, Gen. Hist. 2: 106. 1832. *Adipera* Raf., Sylva Tellur.: 129. 1838. *Diallobus* Raf., Sylva Tellur.: 128. 1838. *Ditremexa* Raf., Sylva Tellur.: 127. 1838. *Emelista* Raf., Sylva Tellur.: 127. 1838. *Herpetica* Raf., Sylva Tellur.: 123. 1838. *Isandrina* Raf., Sylva Tellur.: 126. 1838. *Panisia* Raf., Sylva Tellur.: 128. 1838. *Peiranisia* Raf., Sylva Tellur.: 127. 1838. *Scolodia* Raf., Sylva Tellur.: 128. 1838. *Cassia* subg. *Senna* (Mill.) Benth., Fl. Bras. 15(2): 83, 96. 1870. *Chamaesenna* Raf. ex Pittier, Arb. Arbus. Orden Legum.: 130. 1928. *Cowellocassia* Britton, N. Amer. Fl. 23: 251. 1930. *Earleocassia* Britton, N. Amer. Fl. 23: 247. 1930. *Echinocassia* Britton & Rose, N. Amer. Fl. 23: 251. 1930. *Desmodiocassia* Britton & Rose, N. Amer. Fl. 23: 244. 1930. *Gaumerocassia* Britton, N. Amer. Fl. 23: 252. 1930. *Leonocassia* Britton, N. Amer. Fl. 23: 268. 1930. *Palmerocassia* Britton, N. Amer. Fl. 23: 253. 1930. *Phragmocassia* Britton & Rose, N. Amer. Fl. 23: 245. 1930. *Pseudocassia* Britton & Rose, N. Amer. Fl. 23: 230. 1930. *Pterocassia* Britton & Rose, N. Amer. Fl. 23: 243. 1930. *Tharpia* Britton & Rose, N. Amer. Fl. 23: 246. 1930. *Vogelocassia* Britton, N. Amer. Fl. 23: 258. 1930. *Xerocassia* Britton & Rose, N. Amer. Fl. 23: 246. 1930. 

**Type:** *Senna alexandrina* Mill., Gard. Dict., ed. 8. n. 1. 1768.

Herbaceous or shrubs. Leaves pinnate, often paired, Leaflets 1 to numerous. Glands on the petiole and/or on the rachis of the leaf. Flowers in axillary racemes or panicles. Pedicels without bracteoles. Flowers yellow. Calyx 5-merous, the sepals free, imbricated. Corolla zygomorphic. Petals 5. Stamens 10 unequal, 7 fertile, staminodes mostly 3, anthers fertile, heteromorphic, opening by pores or longitudinal sutures. Fruit dehiscent or indehiscent, flattened, turgid, flat compressed, angled, cylindrical, sometimes with longitudinally winged ridges, pulpy or transversely arranged. 

Genus of 287 species [63]; almost all species are distributed in America, and several species are in tropical Africa, Australia, Asia, and Oceania [63].

Twenty-one species are recorded in northeastern Mexico.
1A.Flowers are strongly asymmetrical, with one petal abaxial, opposite the pistil displaced, not centric, modified in shape (obliquely dilated) and texture, its claw commonly shorter and thicker than that of the other petals. 21B.Flowers are exactly zygomorphic, or if the petals are randomly asymmetrical, then the pistil is central. 62A.Leaves 3.5 cm or more long; largest leaflets longer than 2 cm long; abaxial centric anther as long or shorter than its immediate neighbors.***S. atomaria***2B.Leaves 0.8–5 cm long, leaflets up to 15 mm long; abaxial centric anther similar to its neighbors in shape, but elevated further by a longer filament. 33A.Leaflets 1 pair per leaf.***S. monozyx***3B.Leaflets 2–4 pairs per leaf.44A.Stipules are soon caducous before their associated leaf matures; mature valves are opaque, not elevated above seeds or only in the forming one slight transverse ridge.***S. wislizeni*** var. ***painteri***4B.Stipules persistent with its associated leaf; mature valves shiny, elevated above each seed, forming two transverse ridges. 55A.All parts of the plant with appressed pubescence ***S. wislizeni*** var. ***wislizeni***5B.All parts of the plant with pilulose pubescence***S. wislizeni*** var. ***villosa***6A.Two long abaxial stamens, antepetals raised laterally to the symmetry of the floral axis, their incurved anthers opposite one another, similar to a crab pliers***S. alata***6B.Two long incurved antepetal stamens together in the opposite plane of the vexillary petal (banner), diverging from each other at a narrow angle or subparallel 77A.Subcaulescent herbs; leaflets 1 pair per leaf, these with whitish corneous margins; flowers 1–2 per raceme.***S. pumilio***7B.Caulescent, herbaceous shrubs or trees; leaflets 1 or more pairs per leaf; if only 1 pair, margins never whitish corenous. 88A.Gland present on the petiole at a variable distance below the insertion of the proximal pair of leaflets.***S. occidentalis***8B.Gland present in or slightly above the insertion of the proximal pair of leaflets.9 9A.Fertile stamens 7, equal or slightly accrescent towards the abaxial side of the flower; stamens obliquely or horizontally truncate at the orifice, sometimes attenuated at the apex (not beaked); fruit erect or erect–ascending; plants herbaceous. 109B.Fertile stamens 7, 4 of them shorter and straighter and 3 of them more strongly curved, the central one frequently sterile; fruit often (not always) widely spread or pendulous. 2110A.Monocarpic plants (die after flowering); fruit valves deeply corrugated between seeds; ovules 6–12.***S. uniflora***10B.Perennial plants, fruit valves are superficially corrugated, not deeply sulcate between seeds; ovules 14–44. 1111A.Leaflets 1 pair per leaf.1211B.Leaflets 2–9 pairs per leaf.1712A.Leaflets narrow 1.5–9 times as long as wide.1312B.Leaflets wider, 1.3–1.8 times as long as wide.1613A.Leaflets lanceolate to oblong-lanceolate, 2–9 times as long as wide, the longest ones 2–6 cm; style linear, 0.1–0.25 mm diameter.***S. roemeriana***13B.Leaflets oblong to obovate–elliptic, the largest 1.5–4 times as long as broad; style dilated, 0.4–0.6 mm diameter. 1414A.Style stout, apically dilated, 0.4–0.6 mm diameter; seeds with rough or colliculate testa.***S. bahuinioides***14B.Style linear, 0.2–0.3 mm diameter; seeds with smooth testa1515A.Leaflets of most leaves shorter than petiole; petals 8.5–9.5 mm long; style 3–3.5 mm long; seeds with pinkish-brown testa***S. ripleyana***15B.Leaflets of most leaves longer than petiole; petals 11–18 mm long; style 1.5–2 mm long; seeds with olive-brown testa***S. mensicola***16A.Sepals soon deciduous with petals; petals 7–13 mm long; anthers with conical apex, the pore obliquely intorse; style 1–1.7 mm long; ovules 38–44; seeds in two series***S. durangensis***16B.Sepals persistent with the fruit in formation; petals 8.5–10 mm long; anthers constricted immediately below terminal symmetrical pore; ovules 16–26; seeds in 1 series***S. pilosior***17A.Stipules 1 mm wide or narrower; leaflets 2–5 pairs; flowers 1–10 flowers per raceme; seeds in two series. 1817B.Stipules 1–3 mm wide; leaflets 4–9 pairs; flowers 7–45 per raceme; seeds uniseriate. 1918A.Leaflets 3–5 pairs per leaf; flowers 3–8 per raceme; fruit 1–2 cm long, inflated.***S. crotalarioides***18B.Leaflets 5–7 pairs per leaf; flowers 7–25 per raceme; fruit 3–12 cm long flattened2019A.Stems compact, 10 cm long or less; pubescence strigulous-adpressed, trichomes up to 1 mm long.***S. demissa*** var. ***demissa***19B.Stems compact, 10–35 cm long; pubescence pilose-pilulous, trichomes 0.8–2 mm long, sparse–ascending or horizontal.***S. demissa*** var. ***radicans***20A.Stems and petioles densely hairy, the trichomes fine and erect or retrorse; foliage densely subadpressed-pilulous; petals 10.5–16 mm long; fruit 3.5–6.5 × 0.6–0.9 cm.***S. lindheimeriana***20B.Stems, petioles, and foliage minutely antrorse–strigulous; leaflets glaucous; petals 8–10.5 mm long; fruit 4–12 × 0.35–0.6 cm.***S. orcutii***21A.Anthers of the 2–3 fertile abaxial stamens beaked, the beak porrectly incurved, and its orifice oblique. ***S. obtusifolia***21B.Anthers of the 2–3 abaxial fertile stamens are not beaked, the beak symmetrically truncated.2222A.Fruit flattened-compressed or sometimes turgid; Seeds areolated; ovules 28 or less.***S. guatemalensis*** var. ***calcarea***22B.Fruit cylindrical or strongly turgid; seeds without areoles; ovules 28–150.2323A.Leaflets ovate to lanceolate–acuminate, broadest below the middle***S. septemtrionalis***23B.Leaflets obovate, broadest at or above the middle***S. pendula*** var. ***ovalifolia***

***Senna alata*** (L.) Roxb. Fl. Ind. (Roxburgh) 2: 349. 1832. Basionym: *Cassia alata* L., Sp. Pl. 1: 378. 1753*. Cassia alata* L. var*. perennis* Pampanini, Nuovo Giorn. Bot. Ital. n. ser. 14: 595. 1907. *Cassia bracteata* L. f. sensu Willdenow, Sp. Pl. 2: 525. 1799. *Herpetica alata* sensu Cook & Collins, Contr. U.S. Natl. Herb. 8(2): 159. 1903.

**Type:** Habitat in America calidiore. RCN: 2976. Lectotype (designated by Isely in Mem. New York Bot. Gard. 25 (2): 197. 1975): Herb. Clifford: 158, Cassia 3 (BM-000558725!).

**Distinguishing features:** Shrub, 2–4 m tall, cultivated. Leaves, 25–60 cm long. Leaflets 7–13 pairs per leaf, 6–14 cm long. Fruit 8–16 × 1–2 cm, tetragonal, with papyraceous and crenate-winged margins along the entire length of the body.

**Representative examined material:** Nuevo León: 8-IX-1986, *E. Estrada 656* (CFNL; MEXU); 9-IX-1986, *E. Estrada 726* (CFNL; MEXU). Tamaulipas: 14-VIII-1996, *M. Freya 002* (UAT).

**Comments:** Used as ornamental, sometimes growing in abandoned areas, near human settlements, 300–400 m. Probably originally from the Orinoco and Amazon Basin [8]. 

***S. atomaria*** (L.) H.S. Irwin & Barneby, Mem. New York. Bot. Gard. 35(2): 588. 1982. Basionym: *Cassia atomaria* L. Mantissa 68. 1767. *Cassia arborescens* Mill., Gard. Dict. ed 8, *Cassia* no. 15. 1768. *Cassia longisiliqua* L., Fil. Suppl. 230. 1781. *Cassia triflora* Hahl. Ecl. Amer. 3: 11. 1807. *Cassia elliptica* Kunth, Nov. Gen. & Sp. 6: 356. 1824. *Cassia grisea* A. Richard in Sagra, Hist. Fis. Pol. y Nat. Cuba 10: 493. 1846. *Cassia chrysophylla* A. Ricahard in Sagra op. cit. 500. 1846. *Cassia michoacanensis* Sessé & Moc., Pl. Nov. Hisp. 61. 1888. *Cassia elliptica* Sessé & Moc., Fl. Mex. 101. 1893. *Cassia emarginata.* var. *subunijuga* Roob. & Bartl., Proc. Amer. Acad. Sci. 43: 53. 1907. *Isandrina maxonii* Britton & Rose, N. Amer. Fl. 23(5): 269. 1930. *Cassia planisiliqua* sensu Lamarck, Encycl. Méth. 1: 645. 1785. 

**Type:** Habitat in America. RCN: 2969. (Lectotype: designated by Irwin & Barneby, 1982. Mem. New York Bot. Gard. 35: 588).Voucher: (LINN-528.17).

**Distinguishing features:** Shrub or tree 3–15 m tall. Larger leaflets 2 cm long or longer. Petiole with branched trichomes. Flowers arise from brachyblasts. Flowers are strongly asymmetrical, with one petal abaxial, opposite the pistil displaced, not centric, modified in shape (dilated) and texture, its claw commonly shorter and thicker than that of the other petals, the longest petal folding the androecium. Abaxial centric anther as long or shorter than its immediate neighbors. Fruit 22–40 × 0.8–1.5 cm, oblong, straight, brown to black. 

**Representative examined material:** Tamaulipas: 12; IV-1986, *R. Jones. R. Treviño 91* (UAT); 18-XI-1992, *J.L. Mora-López 375* (UAT); 9-V-1982, *L. Hernández 260* (UAT); 24-VIII-1986, *L. Hernández 1987*; 19-VII-1994, *J.L. Mora-López 505* (UAT); 7-VIII-1969, *H. Puig 2428* (ASU17892!); 19-V-1983, *A. Brito 135* (BCMEX863!); 3-I-1982, *J.D. Bacon 1707* (NY554718!); 3-III-1961, *R.M. King 4060* (NY554724!); 31-III-1974, *P. Fryxell*, *R. Magill 2261* (NY554722!); 13/21-IV-1907, *E. Palmer 323* (NY554721!); 28-XII-1972, *J. Tayor*, *M. Taylor 12486* (NY554726!).

**Comments:** Recorded only in Tamaulipas, in Tamaulipan thornscrub, and deciduous woods. Distributed from Florida USA to South America. 

***S. bauhinioides*** (A. Gray) H.S. Irwin & Barneby, Phytologia 44(7): 499. 1979. Basionym: *Cassia bauhinioides* A. Gray, Boston J. Nat. Hist. 6(2): 180. 1850. *Cassia bauhinioides* var. *arizonica* Macbr., Contr. Gray Herb. n. ser. 59: 27. 1919. *Earleocassia bauhinioides* (A. Gray) Britton ex Britton & Rose, N. Amer. Fl. 23(4): 248. 1930.

**Type:** Mexico, Chihuahua, Santa Rosalía, 2-V-1847, *J. Gregg s.n.* (Original material: GH00274794!).

**Distinguishing features:** Herbaceous perennial, pubescent. Leaves bifoliolate. Leaflets 1–2.5 cm long, oblong to obovate. Style short, strong, distally dilated, less than 1.5 mm long, 0.4–0.6 mm diameter. Fruit 1.9–2.5 cm long, ovoid, densely pubescent. Seeds rough. 

**Representative examined material:** Coahuila: 8-V-1992, A. Rodríguez 1492 (ANSM), 17.VI-1992, J.A. Villarreal 6743 (ANSM). Nuevo León: 5-VII-2001, *C. Yen y E. Estrada 12853* (CFNL) 13-VII-2002, *C. Yen y E. Estrada 14812* (CFNL; MEXU); 8-IX-2001, *C. Yen y E. Estrada 13048* (CFNL); 19-VII-2002, *C. Yen y E. Estrada 14950* (MEXU). 26-VIII- 1989, *Hinton* et al. *19592* (MEXU); 15-VIII-1989, *E. Estrada 1705* (CFNL; MEXU); 27-VII-1986, *E. Estrada 535* (CFNL; MEXU). Tamaulipas: 25-VIII-1983, *L. Hernández 603* (UAT); 25-XI-2984, *L. Hernández 1254* (UAT); 4-VI-1985, *J. Jiménez 182* (UAT); 29-V-1986, *L. Hernández 1824* (UAT).

**Comments:** South of USA and north of Mexico, associated with Tamaulipan thornscrub, *Neltuma* woods, and desert scrub, 190–1900 m.

***S. crotalarioides*** (Kunth) H.S. Irwin et Barneby, Phytologia 44(7): 499. 1979. Basionym: *Cassia crotalarioides* Kunth, Mimoses 132, pl. 40. 1822. *Chamaefistula crotalarioides* (Kunth) G. Don, Gen. Hist. Dichl. Pl. 2: 452. 1832. *Earleocassia crotalarioides* (Kunth) Britton ex Britton & Rose, N. Amer. F. 23(4): 249. 1930. *Cassia vogeliana* Schltdl., Linnaea 12: 342. 1838. 

**Type:** Mexico, Crescit in collibus siccis, inter fodinam La Valenciana et urbem Guanaxuato (Regno Mexicano)], *A.J.A Bonpland*, *F.W.H.A. von Humboldt 4256* (Holotype: P00679232!)

**Distinguishing features:** Herbaceous, perennial, with appressed pubescent or ascending or spreading trichomes. Stipules 1 mm wide or narrower. Leaflets 2–5 pairs per leaf. Flowers 1–10 in distal racemes. Fruit 1.2–2 × 0.5–0.7 cm, inflated. Ovules 22–32. Seeds in two series.

**Representative examined material:** Coahuila: 10-VII-1880, *E. Palmer 281* (MICH1181877). Nuevo León: 7-VI-2003, *C. Yen y E. Estrada 15676* (CFNL); 21-VI-1989, *E. Estrada C. 1547* (CFNL; MEXU; TEX-LL); 23-VII-1993, *Hinton* et al. *23100* (MEXU; TEX-LL); 27-VII-1969, *J. and H. Meras 3282a* (TEX-LL). Tamaulipas: 22-I-1941/11-VIII-1941. *L.R. Stanford* (NY554890!).

**Comments:** Endemic to Mexico. In desert scrub and arid conifer forests 1400–2200. Outside the area, from Durango to Hidalgo and Guanajuato**.**

***S. demissa*** (Rose) H.S. Irwin & Barneby (Rose) H.S. Irwin & Barneby, Phytologia 44(7): 499. 1979. var. ***demissa***. Basionym: *Cassia demissa* Rose, Contr. U.S. Natl. Herb. 10(3): 97–98. 1906. *Earleocassia demissa* (Rose) Britton, N. mer. Fl. 23: 2148. 1930.

**Type**: Mexico, Coahuila, near Carneros Pass, 12-IX-1889, *C.G. Pringle 2783* (Holotype: NY3624!).

**Distinguishing features:** Stems 10 cm long or shorter, compact and tufted, the trichomes up to 1 mm long. Very similar to var. *radicans* but varies in stem size and type of pubescence. The var. *radicans* reach 12–35 cm long with decumbent stems, the pubescence spreading–ascending, 0.8–2 mm long.

**Representative examined material:** Coahuila: 9-VI-1993, *J.A. Villarreal 7188* (ANSM), 12-VIII-2006, *S. Gómez 229* (ANSM), 16-IX-2017, *J.A. Encina 6217* (ANSM). Nuevo León: 15-V-2003, *C. Yen y E. Estrada 15601* (CFNL); 24-VIII-1989, *J.A. Villarreal* et al. *4940* (ANSM; MEXU); 17-VI-1958, *R.M. Straw*, *M. Foreman 1379* (MEXU); 16-IX-1980, *J. Henrickson 18518* (MEXU; TEX-LL); 24-V-1992, *L. Hernández 2679* (TEX-LL). Tamaulipas: 5-XII-1984, *O. Briones 1353* (ANSM), 10-XI-2006, *J.A. Encina 1700* (ANSM); 9-XII-1976, *F. González-M. 10164* (UAT).

**Comments:** In northeastern Mexico and Zacatecas. Associated with rosetophyllous scrublands, *Pinus cembroides* forest, and halophytic vegetation, 1700–2850 m. 

***S. demissa*** (Rose) H.S. Irwin & Barneby var. ***radicans*** (H.S. Irwin & Barneby) H.S. Irwin & Barneby, Phytologia 44(7): 499. 1979. *Cassia demissa* var*. radicans* Irwin & Barneby, Sida 6(1): 9. 1975.

**Type**: Mexico, Coahuila, SW end of Sierra de la Fragua, 1–2 km N of Puerto Colorado, 2-IX-1941, *I.M. Johnston 8774* (Holotype: TEX00371197!). 

**Distinguishing features:** Herbaceous perennial. Stems 10–35 cm long, pubescent, the hairs spreading or ascending, 0.8–2 mm long. Leaflets are almost always 2 pairs, and rarely, some leaves have 3 pairs. The two ventral stamens with thecae are smaller than the 5 dorsal ones. Fruit 0.8–1.5 × 0.4–0.6 cm, inflated, oblong, appressed trichomes with bulbous base. 

**Representative examined material:** Coahuila: 12-X-1991, *M.A. Carranza 977* (ANSM); 22-VII-1969, *M.W. Bierner 86* (NY 554911!); 25-VII-1880, *E. Palmer 281* (NY 3625!); 1848, *J. Gregg 244* (NY 554917!); 3-VII-1941, *L.R. Stanford 229* (NY 554915!). Tamaulipas: 2-VIII-1985, *J. Jiménez 294* (UAT); 11-VIII-1941, *L.R. Stanford 829* (NY 554914!). 

**Comments:** Endemic to northeastern Mexico. Recorded in Coahuila and Tamaulipas. In desert scrub, izotal, *Pinus* and *Juniperus* forest, 1400–1900 m. 

***S. durangensis*** (Rose) H.S. Irwin & Barneby var. ***isleyi*** (H.S. Irwin & Barneby) H.S. Irwin & Barneby, Phytologia 44(7): 499. 1979. Basionym: *Cassia duragensis* var. *isleyi* Irwin & Barneby, Sidea 6(1): 11. 1975.

**Type:** Mexico Tamaulipas, San Fernando a Jiménez [Santander], X-1830, J.L., *J.L. Berlandier 840* (Holotype: NY3628!).

**Distinguishing features:** Stems with tiny retrorse and introrse trichomes and larger semi-setose horizontal trichomes. Leaflets 1 pair. Sepals are soon caducous with the corolla. Petals 7–13 mm long. Anthers apically conical, the pore obliquely intorse; style 1–1.7 mm long. Ovules 38–44. Fruit 2.5–4 × 0.5–0.8 cm, oblong, inflated, papyraceous. Seeds in two series. 

**Representative examined material:** Nuevo León: 20-IV-02, *C. Yen y E. Estrada 14571* (CFNL, MEXU). Tamaulipas: X-1830, *J.L. Berlandier 840* (GH53045!).

**Comments:** Endemic to southern Texas and northeastern Mexico. Recorded in Nuevo León and Tamaulipas. Desert scrublands, calcareous soils, 500–1500 m. Also in San Luis Potosí. Morphologically similar to *S. pilosior*; however, this last one has persistent sepals, shorter petals, 8.5 × 10 mm long, the anthers are constricted immediately below terminal symmetrical pore, 16–26 ovules, and the seeds in 1 series. 

***S. guatemalensis*** (Donn. Sm.) H.S. Irwin & Barneby var. ***calcarea*** Irwin & Barneby, Mem. New York Bot. Gard. 35(1): 306. 1982.

**Type:** Mexico, Tamaulipas, Municipio de Gómez Farias. Sierra de Guatemala. 22-VI-1971, *J. R. Sullivan 548* (Holotype: NY4815!).

**Distinguishing features:** Shrub 1–6 m tall, densely leafy distally. Leaflets 3–7 pairs per leaf. Racemes are shorter than leaves. Fertile stamens are 7, 4 of them shorter and straighter, and 3 of them strongly curved, the central one frequently sterile. Ovules 28 or less. Fruit flattened–compressed or sometimes turgid. Measurements from morphological data taken from [8].

**Representative examined material:** Tamaulipas: 22-VI-1971, *J.R. Sullivan 548* (NY 4815!); 21-VI-1971, *J.R. Sullivan 533* (NY 555071!).

**Comments:** Endemic of Tamaulipas (Sierra de Guatemala), found in deciduous woods and oak–pine forest, 300–1600 m. 

***S. lindheimeriana*** (Scheele) H.S. Irwin & Barneby, Phytologia 44(7): 500. 1979. Basionym: *Cassia lindheimeriana* Scheele, Linnaea 21(4): 457–458. 1848. *Earleocassia lindeheimeriana* Britton ex Britto n & Rose, N. Amer. Fl. 23(4): 249.1930.

**Type**: USA, Texas, north of New Braunfels, IX-1846, *F.J. Lindheimer 380* (Holotype: B presumably destroyed. Isotypes G370869!, GH53026! MO2289256!).

**Distinguishing features:** Herbaceous, erect, up to 1.5 m tall. Stems densely pubescent with antrorse or retrorse trichomes up to 1 mm long. Stipules are ascending, lanceolate, up to 1 cm long. Leaflets 5–8 pairs per leaf. Fruit 3.5–6.5 × 0.5–0.9 cm, oblong, compressed, inertly dehiscent. 

**Representative examined material:** Coahuila: 8-IX-1990, *R. Vázquez 101* (ANSM), 27-IX-2001, *J.A. Encina 909* (ANSM), 27-IX-2001, *M.A. Carranza 3599B* (ANSM). Nuevo León: 23-VII-2002, *C. Yen y E. Estrada 15075* (CFNL); 16-V-2003, *C. Yen y E. Estrada 15655* (CFNL). 1-X-1993, *Hinton 23787* (MEXU; TEX-LL); 9-XI-2002, *C. Yen y E. Estrada 15187* (CFNL); 30-X-2002, *M. González y E. Estrada 15210* (CFNL). Tamaulipas: 3-X-1984, *G. Malda 88* (UAT); 14-IX-1983, *McDonald 873* (UAT).

**Comments:** Southern USA and north of Mexico. Common in northeastern Mexico, 190–2800 m.

***S. mensicola*** (H.S. Irwin & Barneby) H.S. Irwin & Barneby, Phytologia 44(7): 500. 1979. Basionym: *Cassia mensicola* Irwin & Barneby, Sida 6(1): 11–13. 1975.

**Type:** Mexico, San Luis Potosí, Charcas, VII/VIII-1934, *C.L. Lundell 5345* (Holotype: CAS0001432!). 

**Distinguishing features:** Herbaceous perennial. Stems with short, retrorse trichomes mixed with ascending setae up to 1 mm long. Leaflets 2, the blade longer than the petiole. Sepals are deciduous with the corolla. Petals whitish when dry, up to 1.8 cm long. Style 1.5–2 mm long. Fruit 2.3–3.5 × 0.4–0.6 cm, hispid–setose. Seeds olive–brown. 

**Representative examined material:** Coahuila: 10-X-1989, *J.A. Villarreal 5395* (ANSM). Nuevo León: 15-VIII-1989, E. Estrada C. 1705 (CFNL, TEX-LL); 28-V-1987, *Hinton* et al. *19127* (TEX-LL); 21-VIII-1973, *J. Bacon y M. Reynolds 635* (TEX-LL); 15-VII-1998, *Hinton* et al. *27205* (TEX-LL); 26-VIII-1987, *D. Bogler*, *T. Atkins 141* (TEXLL). Tamaulipas: 24-V-1976, *F. González-Medrano 9069* (MICH1182162).

**Comments:** Endemic to Mexico. Common in desert scrub and halophytic grasslands. Also in Zacatecas, San Luis Potosí and Hidalgo, 1000–2250 m.

***S. monozyx*** (H.S. Irwin & Barneby) H.S. Irwin & Barneby, Phytologia 44(7): 500. 1979. Basionym: *Cassia monozyx* Irwin & Barneby, Sida 6(1): 16. 1975.

**Type:** Mexico, Coahuila de Zaragoza; Sierra Paila (Valle Seco), General Cepeda. Alt. 1800 m, 6-VII-1944, *G.B. Hinton 16565* (Holotype: NY3645!).

**Distinguishing features:** Shrub, up to 1.5 m tall, intricately branched, subspinescent, pubescent. Leaves up to 1.5 cm long. Leaflets 1 pair per leaf, 0.3–1 cm long. The abaxial stamen is centric with the anther, like its neighbors in shape, but elevated further by a longer filament. Fruit 8–10 × 0.7–8 cm, compressed, lustrous when mature, the valves swelling over the seeds.

**Representative examined material:** Coahuila: 24-VI-1989, *A. Rodríguez 1126* (ANSM), 30-VIII-1980, *R. Vázquez 44* (ANSM)7-VI-1986, *J.A. Villarreal 3301* (ANSM); 19-VIII-1967, *W.L. Minckley*, *D. J. Pinkava 4212*! (LL).

**Comments:** Endemic to the central region of the state of Coahuila. In desert scrub, 700–900 m. It is the only bushy *Senna* with a pair of leaflets per leaf in northeastern Mexico.

***S. obtusifolia*** (L.) H.S. Irwin & Barneby, Mem. New York Bot. Gard. 35: 252. 1982. Basionym: *Cassia obtusifolia* L., Sp. Pl. 1: 377. 1753. *Cassia tora* Persoon, Syn. Pl. 1: 456. 1805. *Cassia toroides* Raf., Med. Bot. 96. 1828. *Senna toroides* Roxb., Fl. Indica 2: 341. 1832. *Diallobus uniflorus* Raf., Sylva Tellur. 128. 1838.

**Type:** Habitat in Cuba. RCN: 2964. Lectotype (designated by Brenan in Kew Bull. 13: 251.1958): [icon] *Cassia foetida* foliis Sennae Italicae in Dillenius, Hort. Eltham. 1: 71, t. 62, f. 72. 1732- Typotype: Herb. Sherard No. 831 (OXF; iso-G).

**Distinguishing features:** Herbaceous. Stems up to 0.8 m tall. Leaflets 3 pairs per leaf, emarginate or obtuse. Flowers solitary or paired in racemes. Anthers of the 2–3 fertile abaxial stamens beaked, the beak incurved, its orifice oblique. Fruit 16–22 cm long, linear, sub-flattened, reticulated, dehiscent. 

**Representative examined material:** Nuevo León: 15-IV-87, *E. Estrada 890* (CFNL); 25-IX- 1986, *E. Estrada 682* (CFNL, MEXU). Tamaulipas: 11-X-1985, *O. Briones 702* (UAT).

**Comments:** Recorded in Nuevo León and Tamaulipas, in Tamaulipan thornscrub and subtropical woods, 200–400 m. Widely distributed in the tropics of America and Asia. In America, from Florida and Texas to Argentina. 

***S. occidentalis*** (L.) Link, Handb. 2: 140. 1829. Basionym: *Cassia occidentalis* L., Sp. Pl. 377. 1753. *Ditremexa occidentalis* (L.) Britton & Rose ex Britton & Wilson, Sci. Surv. Puerto Rico & Virgin Is. 5(3): 372. 1924. *Cassia falcata* L., Sp. Pl. 377. 1753. *Cassia planisiliqua* L., Sp. Pl. 377. 1753. *Cassia caroliniana* Walter, Fl. Carol. 135. 1788. *Cassia foetida* Persoon, Syn. 1: 457. 1805. *Cassia macradena* Colladon, Hist. Casses 132. 1816. *Cassia ciliata* Raf. Fl. Ludov. 100. 1817. *Cassia obliquifolia* Schrank, Denkschrift. Bot. Ges. Regensburg 2(1): 40. 1822.

**Type:** Habitat in Jamaica. RCN: 2966. Lectotype (designated by Reveal in Phytologia 71: 453–454.1991): Herb. Clifford: 159, Cassia 7, sheet 10 (BM-000558727!).

**Distinguishing features:** Herbaceous up to 1.5 m tall, bad smelling. Foliar gland arising near the base of the petiole. Leaflets 5–6 pairs per leaf. Petals are strongly reticulate. Two long incurved stamens together in the opposite plane of the banner, diverging from each other at a narrow-angle or subparallel. Fruit 9–10.5 × 0.6–0.8 cm, subcylindrical, slightly flattened, with margin widened and lighter in color, dark brown.

**Representative examined material:** Nuevo León: 2-VII-2002, *C. Yen y E. Estrada 11670* (CFNL); 22-X-2002, *C. Yen y E. Estrada 15162* (CFNL); 10-V-1997, *J.A. Villarreal 8603* (ANSM; MEXU); 28-VII-1993, *Hinton* et al. *22884* (TEX-LL). Tamaulipas: 11-X-1985, *O. Briones 2119* (ANSM), 25-XI- 1984, *L. Hernández 1279* (ANSM); 12-II-1993, *J.L. Mora-López 239* (UAT); 25-XI-1984, *L. Hernández 1279* (UAT); 4-XI-1985, *M. Martínez 847* (UAT); 15-IX-1986, *J. Torres 366* (UAT).

**Comments:** Recorded only in Nuevo León and Tamaulipas. In disturbed areas and abandoned crop fields, 360–1250 m. In America, from Mexico to Argentina, mainly in tropical areas. Also in Africa, India, China, Hawaii Islands, Micronesia, and Australia. 

***S. orcuttii*** (Britton & Rose) H.S. Irwin & Barneby, Phytologia 44(7): 500. 1979. Basionym: *Peiranisia orcuttii* Britton & Rose, N. Amer. Flo. 23(4): 267. 1930.

**Type:** 26-VI-1924, *C.R. Orcutt 653* (Holotype: NY4552!).

**Distinguishing features:** Herbaceous or subshrub. Stems, petioles and foliage minutely antrorse–strigulose. Leaflets 3–6 pairs per leaf, glaucous. Very similar in growing habit to *S. lindheimeriana*, differing in the type of pubescence, *S. orcuttii* has antrorse, strigulose pubescence, longer petals (8–10.5 mm long) and longer (4–12 cm long) but narrower (0.3–0.6 mm) ascending, bicarinate, brown fruit.

**Representative examined material:** Coahuila: 29-VII-1973 *M*,*C. Johnston 11905* (ANSM), 12-X-1991 *M.A. Carranza 856* (ANSM): 29-VII-1973, *M.C. Johnston 11905* (NY1584435!); 9-VII-1991, *S. Aguilar*; 29-VII-1973, *M.C. Johnston 11905* (NY1584435!); 9-VII-1991, *S. Aguilar R. 87* (SR3960!). 

**Comments:** South of Texas and New Mexico in USA, and northeastern Mexico, recorded only in the northern Coahuila, in desert scrublands.

***S. pendula*** (Humb. & Bonpl. ex Willd.) H.S. Irwin & Barneby var. ***ovalifolia*** H.S. Irwin & Barneby, Mem. New York Bot. Gard. 35(1): 391. 1982. *Cassia ovalifolia* M. Martens & Galeotti, Bull. Acad. Roy. Brux. 10(9): 305. 1843. *Adipera ovalifolia* (M. Martens & Galeotti) Britton & Rose, N. Amer. Fl. 23(4): 241. 1930. *Cassia botteriana* Benth. Trans. Lin. Soc. Lond. 27: 541. 1871. *Adipera submontana* Britton & Rose, N. Amer. Fl. 23(4): 241. 1930.

**Type:** Mexico, IV-1840/X-1840, *H.G. Galeotti 3260* (Holotype: BR5174966!).

**Distinguishing features**: Shrub or tree, 1–4 m tall. Leaflets 3–5 pairs per leaf, obovate. Anthers of the 2–3 abaxial fertile stamens are not beaked, the beak symmetrically truncated. Fruit cylindrical 10–15 × 1–1.5 cm.

**Representative examined material:** Tamaulipas: 11-XI-1975, *A.A. Lesseigne 4895* (NY1585056); 14- 3-III-1961, *R.M. King 4048* (NY1585055!). 

**Comments:** Recorded only in Tamaulipas. In deciduous woodlands, piedmont scrub and forest, 30–1200 m. Widely distributed, from southern Texas to Venezuela.

***S. pilosior*** (B.L.Rob. ex J.F. Macbr.) H.S. Irwin & Barneby, Phytologia 44(7): 500 (1979). *Cassia bauhinioides* var*. pilosior* B.L. Rob. ex MacBride, Contr. Gray Herb. 59: 27. 1919. *Cassia pilosior* (B.L.Rob. ex J.F.Macbr.) H.S. Irwin & Barneby, Sida 6(1): 10. 1975.

**Type:** USA, Texas, Bofecillos Mountains (West Texas), IX-1983, *V. Havard*, *14* (Holotype: GH53012!).

**Distinguishing features:** Herbaceous, perennial. Stems pubescent, with simple retrorse trichomes, glanduliform trichomes, and stiff setae. Leaflets 1 pair per leaf, 1.3–1.8 times as long as wide. Foliar gland arising in the insertion of the leaflets. Stamens constricted immediately below the terminal symmetrical pore. Fruit erect, 2–4 × 0.5–8 cm, with trichomes and setae, persistent after dehiscence. 

**Representative examined material**: Coahuila: 29-VII-1076, *A. Roig 113* (ANSM), 5-IX-1981, *A. Rodríguez 405* (ANSM), 28-III-2007, *J.A. Encina 5704* (ANSM; 9-VI-1960, *D.J. Pinkava 5153* (ASU0017926!); 14-VI-1968, *D.J. Pinkava 5685* (ASU0017928!); 11-VIII-1940, *I.M. Johsnton*, *C.H. Muller 127* (USF173694!).

**Comments:** Endemic to southern Texas and north of Mexico. In desert scrublands, 600–1450 m.

***S. pumilio*** (A. Gray) H.S. Irwin & Barneby, Phytologia 44(7): 500. 1979. Basionym: *Cassia pumilio* A. Gray, Boston J. Nat. Hist. 6(2): 180. 1850. *Tharpia pumilio* (A. Gray) Britton & Rose, N. Amer. Fl. 23(4): 247. 1930.

**Type:** USA, Llano and Pedernales rivers, May, Oct, 1847. Lectotype (designated by H.S. Irwin & Barneby, Mem. New York Bot. Gard. 35: 288. 1982) (GH00053032!). 

**Distinguishing features:** Tiny herbaceous, sub-acaulescent, up to 13 cm tall. Leaflets 1 pair per leaf, linear with whitish corneous margins. Flowers 1–2, originating from the base of the plant in peduncles up to 10 cm long, surpassing the leaves. Fruit 0.7–1.5 × 0.5–0.8 cm, obovoid, reflexed, inflated.

**Representative examined material**: Coahuila: 23-V-1980 *M.A. Carranza 1138* (ANSM), 13-VI-1957, *R. McVaugh 14803* (MEXU). Nuevo León: 20-VII-2002, *C. Yen y E. Estrada 14970* (CFNL)*;* 15-XI-90, *E. Estrada 1925* (CFNL); 13-VII-2002, *C. Yen y E. Estrada 14806* (CFNL, MEXU).

**Comments:** Southeastern Texas and northern Mexico, from Chihuahua and Durango to Coahuila and Nuevo León. Tamaulipan thornscrub, mezquitales (*Neltuma glandulosa* and *N. laevigata*), desert scrub, and halophytic grasslands, 250–1700 m.

***S. ripleyana*** (H.S. Irwin & Barneby) H.S. Irwin & Barneby, Phytologia 44(7): 500. 1979. Basionym: *Cassia ripleyana* H.S. Irwin et Barneby, Sida 6(1): 13–14. 1975.

**Type**: México, Chihuahua, 18 mi W of Jiménez. Alt. 4700 ft., 2-X-1965, *H.D.D. Ripley*, *R.C. Barneby 13904* (Holotype: NY3654!).

**Distinguishing features:** Herbaceous, sub-acaulescent or short caulescent. Stems 5–19 cm long. Leaflets 1 pair per leaf, shorter than petiole, 1.5–9 times as long as wide. Style linear, 0.2–0.3 mm diameter. Petals whitish-brown when dry. Fruit 1.5–2.5 × 0.5–0.8 cm, oblong, ascending, inflated, dehiscent. 

**Representative examined material**: Coahuila: 11-VIII-1974 *A. Rodríguez 1370* (ANSM); 2-IV-1982, *A. Rodríguez 1029* (NY 1585471!); 11-IX-1993, *Hinton 22997* (TEX189382!); 25-VII-1993, B.L. Turner, K. Clary, T.F. Patterson 93–115 (TEX189383!). Nuevo León: 1-VII-99, *C.Yen y E. Estrada 10969* (CFNL); 23-VII-1999, *E. Estrada 10422* (CFNL); 19-VII-1999, *E. Estrada 10342* (CFNL).

**Comments:** Southwestern Texas and north of Mexico, from Chihuahua to Nuevo León and Zacatecas. In desert scrub and halophytic grasslands. 

***S. roemeriana*** (Scheele) H.S. Irwin & Barneby, Mem. New York Bot. Gard. 35(1): 282. 1982. Basionym: *Cassia roemeriana* Scheele, Linnaea 21: 457. 1848. *Earleicassia roemeriana* (Scheele) Britton ex Britton & Rose, N. Maer. Fl. 23(4): 247. 1930. 

**Type:** Lectotype: Designated by H.S. Irwin & Barneby, 1982. The American Cassiinae A synoptical revision of Leguminosae tribe Cassieae subtribe Cassiinae in the New World. Mem. New York Bot. Gard. 35(1): 282. Voucher: USA, Texas, 1846, F.J. Lindheimer s.n. (MEL250604!).

**Distinguishing features:** Herbaceous, perennial. Stems with tiny trichomes erect or subretrorse and ascending setae up to 0.7 mm long. Leaflets 1 pair per leaf, lanceolate, 2–9 times as long as wide. Style linear, 0.1–0.25 mm diameter. Fruit ascending, 2.0–3.5 × 0.4–0.6 cm, oblong, turgid, brown, pubescent, dehiscent. 

**Representative examined material**: Coahuila: 12-X-1991 *M.A. Carranza 850* (ANSM), 25-V-2016, *J. Encina 5404* (ANSM); 18-IV-2017, *J. Encina 5741* (ANSM); 25-V-2016, *J. Encina 5404* (ANSM); 18-IV-2017, *J. Encina 5741* (ANSM); Nuevo León: 15-IV-1906, *C.G. Pringle 13751* (TEX-LL).

**Comments:** From Oklahoma, New Mexico and Texas to Coahuila, Nuevo Léon and Zacatecas. In desert scrublands, 300–1600 m. 

***S. septemtrionalis*** (Viv.) H.S. Irwin & Barneby, Mem. New York Bot. Gard. 35(1): 365. 1982. Basionym: *Cassia septemtrionalis* Viv., Elench. pl. hort. Bot. J. Car. Dinegro 14. 1802. *Cassia laevigata* Willd., Enum. pl. hort. Berol. 441. 1809. *Cassia elegans* Kunt, Nov. Gen. & Sp. 6: 342. 1824. *Senna aurata* Roxb., Fl. Indica 2: 342. 1832. *Cassia vernicosa* Clos in Gay, Hist. Chile, Flora 2: 244. 1854. *Adipera laevigata* (Willd.) Britton & Rose in Britton & Wilson, Sci. Surv. Porto Rico & Virgin Is. 5(3): 371. 1924. *Chamaefistula laevigata* (Willd.) G. Don, Gen. Hist. Dichl. Pl. 2: 452. 1832.

**Type:** No type known to survive, described from plants cultivated at Genoa and acquired ex hort. Tic. et Flor [entino] (Irwin & Barneby, 1982). 

**Distinguishing features:** Shrub 1.5–5 m tall. Leaflets 2–3 pairs per leaf, ovate–lanceolate, broadest below the middle. Foliar gland arising at the insertion of proximal pair of leaflets. Anthers of the 2–3 abaxial fertile stamens are not beaked, the beak symmetrically truncated. Fruit 8.5 cm long, linear-oblong, slightly turgid, glabrous.

**Representative examined material**: Coahuila: 16-VII-1993 *M.A. Carranza 1607* (ANSM). Nuevo León: 2-VIII-1994, *Hinton* et al. *24569* (MEXU). Tamaulipas: 6-IV-1996, *C. Ramos* 18 (CFNL); 16-III-1991, *E. Estrada 2005*, *J. Fairey*, *C. Schoenfeld* (CFNL); 6-X-2000, *E. Estrada 13146* (CFNL).

**Comments:** In oak and oak–pine forest, rain forest, and disturbed places, 800–1500 m. From northeastern Mexico to South America. Used as ornamental.

***S. uniflora*** (Mill.) H.S. Irwin & Barneby, Mem. New York Bot. Gard. 35: 258 1982. Basionym: *Cassia uniflora* Mill., Gard. Dict., ed. 8. ed. 8, *Cassia* no. 5. 1768.

**Type**: Mexico, 1-VII-1730, *Houston s.n.* (Holotype: E346665!).

**Distinguishing features:** Monocarpic, 1.5–5 m tall, malodorous. Leaflets 3–5 pairs per leaf. Foliar glands arise in almost all insertions of a pair of leaflets. Fertile stamens 7, equal or slightly accrescent towards the abaxial side of the flower, obliquely or horizontally truncate at the orifice, sometimes attenuated at the apex (not beaked). Fruit 2.5–5 × 0.3–0.5 cm, bicarinate, brown to black, setose, corrugated between seeds, appearing a false loment, dehiscent. 

**Representative examined material**: Tamaulipas: 9-Xi-1984, *S. Rodríguez 251* (UAT); 26-IX-1985, *P. Moya 72* (UAT); 21-X-1983, *L. Hernández*, *F. Gonzélez-Medrano*, *C. Cortes 800* (UAT); 8-XI-1996, *C. Ramos 150* (CFNL); 6-III-1983, *M.H. Nee 25750* (NY1585904!); 1-I-1910, *E. Palmer 44* (NY 1585877!); 

**Comments:** Recorded in the state of Tamaulipas, in tropical and subtropical areas. From north of Mexico to Brazil. 

***S. wislizeni*** (A. Gray) H.S. Irwin & Barneby var. ***painteri*** (Britton & Rose) H.S. Irwin & Barneby, Mem. N.Y. Bot. Gard. 35(2): 574. 1982. Basionym: *Palmerocassia painteri* Britton ex Britton & Rose, N. Amer. Fl. 23(4): 254. 1930. *Cassia wislizeni* var. *painteri* (Britton & Rose) H.S. Irwin & Barneby, Phytologia 44(7): 500. 1979. *Cassia wislizeni* var. *painteri* (Britton & Rose) H.S. Irwin & Barneby, Sida 6(1): 16. 1975.

**Type:** Mexico, Querétaro de Arteaga; Near Higuerillas, 23-VIII-1905, *J. N. Rose; J. H. Painter*, *Russell 9807* (Holotype: NY4527!).

**Distinguishing features:** Physiognomically very similar to *S. w.* var. *wislizeni*, but the var. *painteri* has caducous stipules and opaque or dull fruit, never shiny; the valves do not bulge above seeds, or if so, then with only one transverse crest.

**Representative examined material**: Coahuila: 25-VI-2002, J.A. Encina 1218 (ANSM). Nuevo León: 31-V-2003, *C. Yen y E. Estrada 15667* (CFNL); 21-VI-2003, *C. Yen y E. Estrada 15764* (CFNL; MEXU); 7-VI-2003, *C. Yen y E. Estrada 15682; Peña Nevada*, *8-VI-2003*, *C. Yen y E. Estrada 15745* (CFNL), 7-VI-2003, *C. Yen y E. Estrada 15678* (CFNL); 2-VIII-1980, *Hinton* et al. *17906* (TEX-LL); *R. Torres* et al. *1100* (MEXU); 29-VIII-1989, *E. Estrada 1713* (CFNL; MEXU; TEX-LL). Tamaulipas: 25-VII-1985, *D. Méndez 98* (UAT). 

**Comments:** Endemic to Mexico, from northeastern Mexico to Querétaro, in desert scrublands, 1400–2000 m.

***S. wislizeni*** (A. Gray) H.S. Irwin & Barneby var. ***villosa*** (Britton) H.S. Irwin & Barneby, Phytologia 44(7): 500. 1979. *Palmerocassia villosa* Britton ex Britton & Rose, N. Amer. Fl. 23(4): 254. 1930. *Cassia wislizeni* var. *villosa* (Britton & Rose) H.S. Irwin & Barneby, Sida 6(1): 16. 1975. 

**Type:** México, Durango; Mapimi and vicinity, 21/23-X-1898, *E. Palmer 518* (Holotype: NY4528!).

**Distinguishing features:** Physiognomically very similar to *S. w.* var. *wislizeni*, both with persistent stipules, but var. *villosa* has pilosulous pubescence, never appressed and mostly with 2–3 pairs of leaflets per leaf. 

**Representative examined material**: Coahuila: 21-X-1989, *J.A. Villarreal 5525* (ANSM), 25-VIII-1988, *J.A. Villarreal 4440* (ANSM), 4-IX-2007, *J.A. Alba 208* (ANSM),30-VI-1941, *L.R. Stanford 129* (NY1586158!).

**Comments:** Endemic to the north of Mexico, recorded in Coahuila, outside the area, in Durango. In desert scrublands, mountain slopes, and calcareous soils, 1100–1600 m.

***S. wislizeni*** (A. Gray) H.S. Irwin & Barneby var. ***wislizeni*** (Britton & Rose) H.S. Irwin & Barneby, Mem. N.Y. Bot. Gard. 35(2): 574. 1982. Basionym: *Cassia wislizeni* A. Gray, Pl. Wright. 1: 60. 1852. *Palmerocassia wislizeni* (A. Gray) Britton & Rose, N. Amer. Fl. 23(4): 254. 1930. 

**Type:** Mexico, Carizal and Ojo Caliente, S. of El Paso, *F.A. Wislizenus 107* (Holotype: GH53035!).

**Distinguishing features:** Shrub, 1.5–4.5 m tall. Stems appressed-pubescent in the whole plant. Stipules persistent. Leaflets 3–5 pairs per leaf. Fruit 9–15 × 0.5–0.8 cm, linear, flattened, subcoriaceous, shiny when ripe, reticulate, bulging over seed, like two parallel, transverse crests. Measurements from morphological data taken from [8].

**Representative examined material**: Coahuila: 19-IX-1971, *J.S. Henrickson 6877* (TEX420586!).

**Comments:** Southern USA, to the north of Mexico (Sonora to Coahuila). Recorded in Coahuila, in desert scrublands, 1400–1600 m.

**Subfamily Cercidoideae** Legume Phylogeny Working Group, Taxon 66(1): 88. 2017. Cercideae Bronn, Form. Pl. Legumin.: 134: 131. 1822.

Shrubs or trees. Stipules free. Leaves unifoliolate entire, bilobed, or bifoliolate, and the leaflets opposite. Flowers in racemes or pseudoracemes, bisexual, strongly or slightly bilaterally symmetrical, sometimes papilionated in appearance. Calyx sepals are free or united in a spathaceous structure. Petals 1–5, free, imbricate, the banner the innermost, frequently different in shape from the rest of the petals. Stamens 1–10, sometimes staminodes present. Filaments are partly connate or free. Fruit flattened, dehiscent, the valves coiling helicoidal after dehiscence. 

The subfamily is composed of 12 genera and approximately 335 species [1]. In northeastern Mexico, only two genera, *Bauhinia* and *Cercis.*
1A.Leaves simple; flowers pink or purple, papilionate in appearance, 1.5 cm long or shorter; fruit winged on adaxial margin.***Cercis***1B.Leaves bifoliolate or, if simple, always bilobed; flowers pink, purple, or white, not papilionate in appearance, greater than 1.5 cm long; fruit with entire edges, without wing on the adaxial margin***Bauhinia***

***Bauhinia*** L., Sp. Pl. 1: 374–375. 1753. *Pauletia* Cav., Icon 5: 5. 1753. *Amaria* Mutis, Sem. Nuev. Rey. Gran. 2. 25. 1810. *Casparia* Kunth, Ann. Sci. Nat. 1: 85. 1824. *Perlebia* Mart., in Spix et Mart. Reise Bras. 2: 555. 1828.

**Type:** (Lectotype: designated by Hitchcock, A. S. & M. L. Green. 1929. Standard species of Linnaean genera of Phanerogamae (1753–1754). 152. In Nom. Prop. Brit. Bot. His Majesty’s Stationery Office, London).

Shrub or trees, unarmed. Leaves entire, bilobed, or bifoliolate. Flowers are solitary, paired or in axillary or terminal racemes or panicles, and zygomorphic. Calyx gamosepalous, zygomorphic, almost entirely fused, free apically, 2–5-lobed, or forming a spathaceous structure. Petals 5 or less or only one, free, imbricate, equal or subequal, unguiculate, red, pink, greenish, yellow, or white. Stamens 10, fused basally, 3–5 or all fertile, or 9 abortive and sterile (staminodes) and only 1 fertile, free, or partially fused. The pistil is the same size as the androecium. Fruit flattened, woody, elastically dehiscent, or indehiscent. 

Represented by approximately 150–160 species [4] and pantropical distribution [66]. Mostly diversified in South America (75 species), North and Central America (35), Africa, and Madagascar (32) [4]. Twenty-seven arboreal Bauhinias are recorded for Mexico and Central America [47]. At least 30 species are recorded from Mexico [20]. Six wild species [66,67] and one cultivated in Mexico and around the world [68] have been recorded for northeastern Mexico.
1A.Fertile stamens 5; cultivated plants21B.Fertile stamen 1–3; wild plants32A.Petals pink, violet, and purple with colored spots***B. variegata*** var. ***variegata***2B.Petals white***B. variegata*** var. ***candida***3A.Fertile stamens 3***B. coulteri*** var. ***coulteri***3B.Fertile stamen 144A.Leaves entire, bilobed, but the blade never completely divided into two leaflets54B.Leaves bifoliolate, its blade completely divided into two leaflets65A.Petals pink or purple; leaf lobes wide and rounded apically***B. macranthera***5B.Petals white or whitish-cream; leaf lobes narrow and triangular or acute apically***B. divaricata***6A.Leaflets 5–9 cm long, narrow, and gradually acute apically***B. bartlettii***6B.Leaflets up to 3 cm long, wide, and rounded apically 77A.Petals white; young fruits glabrate or almost so***B. lunarioides***7B.Petals pink, dark pink to purple; young fruits densely pubescent88A.Leaflets 1.7–5 cm long, mostly 3-veined***B. ramosissima*** var. ***ramosissima***8B.Leaflets 0.4–1.5 cm long, mostly 2-veined***B. ramosissima*** var. ***uniflora***

***Bauhinia bartlettii*** B.L. Turner, Phytologia 76: 4. 1994.

**Type:** Mexico, Tamaulipas: Mpio. Hidalgo, W of Santa Engracia into the Sierra, 7.2 mi W of Guayabas, 4.0 mi W of the Guayabas-Adelaida junction. 24°1′ N, 99°34′ W, 16-IV-1988, *G. Nesom*, *L. Hernández*, *M. Martínez*, *J. Jiménez 6312* (Holotype: TEX371176!).

**Distinguishing features:** Shrub or tree up to 6 m tall. Leaflets 2 per leaf, parallel to divaricate, 5–9 x 1–4 cm, lanceolate, slightly falcate, glabrate. Petals pink to purple. Stamen 1 (fertile), staminodes 9, basally connate. Fruit 5–16 × 1.5–2 cm, oblong, subglabrous. 

**Representative examined material**: Tamaulipas: 9-XI-2001, *E. Estrada 13191* (CFNL); 7-V-1986, *García 2214* (MEXU); 22-III-1987, *García 2908* (MEXU); 19-III-1998, *A. Mora-Olivo 6808* (UAT).

**Comments:** Endemic to Tamaulipas, associated with deciduous woods, 350–700 m.

***B. coulteri*** Macbr., Contr. Gray Herb. 59: 22. 1919. var. ***coulteri*.** Basionym: *Bauhinia platypetala* Benth. ex Hemsl., Diagn. Pl. Nov. 1: 49. 1880. *Casparia coulteri* (Macbr.) Britton & Rose, N. Amer. Fl. 23: 216. 1930. *Bauhinia coulteri* forma *albiflora* Wunderlin, Rhodora 70: 286, 1968. 

**Type:** Mexico, Hidalgo, Zimapán, *T. Coulter 531* (Holype: not found. Isotype: GH 53006).

**Distinguishing features:** Small shrub, up to 1 m tall, leaves sub-glabrous with smooth margins. Flowers pink with 3 fertile stamens.

**Representative examined material**: Tamaulipas: 2-XI-2020, *E. Estrada* et al., *25115* (CFNL); 28-V-1976, *F. González-Medrano 9269* (UAT); 19-VI-1985, *L. Hernández 1483* (UAT); 20-I-1976, *F. González-Medrano 9923* (UAT); 16-VI-1987, *G. Nesom 5996* (UAT).

**Comments:** Endemic to Mexico. Outside of the study area, in San Luis Potosí, Hidalgo, and Querétaro. 

***B. divaricata*** L., Sp. Pl. 374. 1753. Basionym: *Mandarus divaricatus* (L.) Raf. Sylva Tellur. 122. 1838. *Casparia divaricata* (L.) Kunth ex Britton & Rose, N. Amer. Fl. 23: 215. 1930. *Bauhinia aurita* Ait, Hort. Kew. 2: 48. 1789. *Bauhinia porrecta* Sw. Prodr. 66. 1788. *Bauhinia latifolia* Cav. Icon. 5: 4. 1799. *Bauhinia retusa* Poir., Encyc. Suppl., 1: 599. 1811. *Bauhinia racemifera* Desv., J. Bot. 3: 74. 1814. *Bauhinia americana* Laun, Herb. Amat. 5: 315. 1821. *Bauhinia lamarckiana* DC., Prodr. 2: 512. 1825. *Bauhinia spathacea* DC., Prodr. 2: 512. 1825. *Bauhinia furcata* Desv. Ann. Sci. Nat. 9: 429. 1826. *Bauhinia dansoniana* Guill. & Per., Flor. Seneg. Tent. 1: 265. 1832. *Bauhinia versicolor* Bertol., Hort. Bonon. Pl. Nov. 1: 7. 1838. *Bauhinia mexicana* Vog., Linnaea 13: 299. 1839. *Bauhinia schlechtendaliana* M. Martens & Galeotti, Bull. Brux. 10: 308. 1843. *Bauhinia ambliopnylla* Harms in Loess., Bull. Hebr. Boissier 7: 548. 1899. *Bauhinia confusa* Rose, Contr. U.S. Nat. Herb. 10: 97. 1906. *Bauhinia goldmanii* Rose, Contr. U.S. Nat. Herb. 10: 97. 1906. *Bauhina caribaea* Jennings, Ann. Carnegie Mus. 11: 127. 1917. *Bauhinia peninsularis* Brandegee, Univ. Calif. Publ. Bot. 10: 183. 1922. *Bauhinia divaricata* var. *angustiloba* Ekman & Urban, Ark. Bot.24A: 8. 1931.

**Type:** (Lectotype: designated by Stearn, Introd. Linnaeus Sp. Pl. (Ray Soc. Ed.): 47. 1957: Voucher Herb. Clifford 156, *Bauhinia* 2 (BM)).

**Distinguishing features:** Shrub or tree up to 8 m tall. Leaves orbicular and bilobed; the lobes are narrow and triangular or acute-rounded apically, parallel or divaricate. Fertile stamen 1. Petals white or whitish-cream (sometimes turning pale pink with age). 

**Representative examined material**: Tamaulipas: 12-X-2002, *E. Estrada 15157*, *J. Pérez G.* (CFNL); 12-XII-1987, *E. Estrada 1056* (CFNL); 24-IX-1985, *M. Yanez 583* (CFNL); 19-III-1988, *E. Estrada 1372* (CFNL); 21-II-1981, *L. Hernández 1003* (CFNL); 24-V-1996, *C. Ramos 56* (CFNL); 30-III-2003, *E. Estrada 15394* (CFNL); 8-IV-1998, *A. Mora-Olivo 6995* (UAT); 26-IX-1990, *J. Sifuentes 60* (UAT); 27-VI-1992, *J.L. Mora-López 182* (UAT); 22-VIII-1986, *M. Martínez 1228* (UAT); 21-VIII-1985, *M. Yanez 429* (UAT).

**Comments:** From Baja California Sur and Tamaulipas to Central America (Costa Rica). In deciduous woods, rainforest areas, evergreen forests, and disturbed areas.

***B. lunarioides*** A. Gray ex S. Watson, Bibl. Ind. N. Amer. Bot. 205. 1878. Basionym: *Bauhinia congesta* (Britton & Rose) Lundell, Phytologia 1: 214. 1937. *Casparia congest* Britton & Rose, N. Amer. Fl. 23: 211. 1930. *Bauhinia jermyana* (Britton) Lundell, Phytologia 1: 214. 1937. *Casparia jermyana* Britton in Britton & Rose, N. Amer. Fl. 23: 211. 1930.

**Type:** Mexico, Coahuila, near Santa Rosa, I-1853, *C.C. Parry 290a* (Holotype: GH59714!). 

**Distinguishing features:** Shrub up to 4 m tall. Leaves bifoliolate or its blade bilobate for at least three-quarters or more of its length or completely divided into two leaflets. Leaflets up to 3 cm long, suborbicular slightly divergent. Fertile stamen 1, staminades 9. Petals are white (rarely light-pink). Fruit 5–8 × 1–2 cm, oblong, glabrate. 

**Representative examined material**: Coahuila: 10-VIII-1980, *R. Vázquez 14* (ANSM), 10-V-1981, *L. Rodríguez 159* (ANSM); 27-VIII-1984, *J.A. Villarreal 2674* (ANSM). Nuevo León: 9-IV-2001, *C. Yen y E. Estrada 12035* (CFNL); 7-VIII-1992, *E. Estrada 2425*, *C. Schoenfeld*, *J. Fairey* (CFNL); 24-VI-2001, *E. Estrada 12825* (CFNL); 20-VII-2002, *C. Yen y E. Estrada 14977a* (CFNL); 15-IV-2001, *C. Yen y E. Estrada 12363* (CFNL); 6-VII-2001, *C. Yen. E. Estrada 12973* (CFNL). 

**Comments:** Endemic to northeastern Mexico and the south end of Texas (USA). *B. lunaroioides* is physiognomically very similar to *B. ramosissima*, but both species can be distinguished by the color of the flowers: white in *B. lunarioides* and dark pink in *B. ramosissima*. If pink flowers are present, and it is difficult to recognize the species, just note that the immature fruits of *B. lunarioides* are always glabrous, while those of *B. ramosissima* are densely pubescent. In desert scrublands and piedmont scrub, 450–1600 m.

***B. macranthera*** Benth. ex Hemsl., Diag. Pl. Nov. 49. 1880. Basionym: *Casparia macranthera* (Bent. ex Hemsl.) Britton & Rose, N. Amer. Fl. 23: 212. 1930. *Bauhinia retifolia* Standl., Contr. U.S. Nat. Herb. 23: 416. 1922. *Casparia lunarioides* A. Gray ex Britton & Rose, N. Amer. Fl. 23: 212. 1930. *Bauhinia macranthera* var*. grayana* Wunderlin, Phytologia 15: 53. 1967.

**Type:** Mexico, *Coulter*, *s.n.* (Holotype: K264626!).

**Distinguishing features:** Shrub 1–6 m tall. Leaves simple, entire, bilobed, the lobes fused at the base by 0.6–3 cm long, lobes 3–11 cm, basally cordate, apex of lobes rounded, ovate to suborbicular. Petals pink or dark purple. Fertile stamen 1, staminodes 9, basally fused to the middle portion. Fruit 8–15 × 1–2 cm, oblong, flattened, dehiscent, brown.

Common species in piedmont scrub and oak–pine forests, 570–1800 m.

**Representative examined material**: Coahuila: 27-VI-1936, *F.L. Wynd 316* (ANSM), 6-VI-1991, *J.A. Villarreal 5965* (ANSM), 20-IV-2017 *J.A. Encina 5788* (ANSM). Nuevo León: 7-VII-2001, *C. Yen y E. Estrada 13004* (CFNL); 14-IV-2001, *C. Yen y E. Estrada 12159* (CFNL); 22-IX-2001, *C. Yen y E. Estrada 13095* (CFNL); 26-XI-1987, *E. Estrada 1030* (CFNL); 23-VII-2002, *C. Yen y E. Estrada 15034* (CFNL). Tamaulipas: 22-VI-1996, *C. Ramos 109* (CFNL); 15-III-1991, *E. Estrada 1963*, *J. Fairey*, *C. Schoenfeld* (CFNL); 16-VI-2007, *E. Estrada 20039* (CFNL); 4-VII-1994, *L. Hernéndez 3154* (UAT).

**Comments:** In piedmont scrub and oak–pine forest, 570–2200 m. Easily recognized by its only 1 fertile stamen, leaves entire but deeply divided in rounded lobes, not triangular as in *B. divaricata*. Northeastern to central Mexico. 

***B. ramosissima*** Benth. ex Hemsl. var. ***ramosissima*.** Basionym: *Bauhinia unguicularis* Benth. ex Hemsl., Diag. Pl. Nov. 49. 1880. *Casparia purpusii* Britton in Britton & Rose, N. Amer. Fl. 23: 210. 1930. *Casparia runyonii* Britton & Rose, N. Amer. Fl. 23: 120. 1930.

**Type:** Mexico, Zimapán, *Coulter 473* (Holotype: K501152!).

**Distinguishing features:** Shrub, up to 3 m tall. Leaves bifoliolate. Leaflets 1.7–5 cm long, apically rounded, mostly 3-veined. Fertile stamen 1. Petals pink, dark pink to purple. Young fruits are densely pubescent.

**Representative examined material**: Coahuila: 4-VII-1981 *A. Rodríguez 513* (ANSM). 22-VII- 1992 *J.A. Villarreal 6686* (ANSM). Nuevo León: 22-III-2003, *C. Yen y E. Estrada 15355* (CFNL); 16-V-1981, *J.M. Poole* et al. *2296* (MEXU); 16-V-1981, *Hinton 18262* (MEXU). 9-IV-1994, *T.F. Patterson 7477* (MEXU). Tamaulipas: 18-IX-1985 *R. Díaz 488* (ANSM); 9-XII-1976, *F. Guevara 10167* (UAT).

**Comments:** Endemic to Mexico. Recorded in Nuevo León and Tamaulipas. Outside the study area, in San Luis Potosí, Hidalgo, and Querétaro. Both varieties of *B. ramosissima* physiognomically (mainly in the type of leaves and leaflets) resemble *B. lunarioides*, but this last species has white flowers. In mountain slopes, desert scrublands, piedmont scrub, and oak forest, 670–1800 m.

***B. ramosissima*** var. ***uniflora*** (S. Watson) M.P. Ramírez & R. Torres, Brittonia 59(4): 364. 2007. Basionym: *Bauhinia uniflora* S. Watson, Proc. Amer. Acad. Arts 21: 451. 1886. *Caspatia uniflora* (S. Watson) Britton & Rose, N. Amer. Fl. 23: 209. 1930. *Casparia monantha* Britton & Rose, N. Amer. Fl. 23(4): 210. 1930.

**Type:** Mexico, Chihuahua [in fact, Coahuila], Jimulco, 27-IV-1985, *C.G. Pringle 174* (Holotype: not found, Isotype: NA26187!).

**Distinguishing features:** Shrub up to 2 m tall. Leaves bifoliolate. Leaflets 0.4–1.5 cm long, mostly 2-veined. Fertile stamen 1. Petals pink, dark pink to purple, young fruits densely pubescent.

**Representative examined material**: Coahuila: 24-IX-2010 *J. Alba 536* (ANSM), 17-VI-2014 *J.A. Encina 4270* (ANSM), 26-V-1982 *J. Valdés 1488* (ANSM), 14-X-1986 *J.A. Villarreal 3610* (ANSM). Nuevo León: 26-III-1994, *Hinton* et al. *24051* (TEX-LL); 9-VI-2001, *J. Medellín s.n.* (CFNL). Tamaulipas: 3-VII-1985, *P. Hiriart 752* (UAT); 12-II-1976, *F. González-Medrano 8566* (UAT); 20-IX-1993, *G. Nesom 7670* (UAT); 26-VII-1985, *M. Yanez 289* (UAT); 18-IV-1976, *P. Zavaleta 8741* (UAT); 28-V-1986, *L. Hernández 1798* (UAT).

**Comments:** Endemic of Mexico. This variety has shorter leaflets and occurs in the same habitats as var. *ramosissima*. Also in Chihuahua, Durango, San Luis Potosí, Hidalgo, and Querétaro.

***B. variegata*** var***. candida*** Voigt, Hort. Suburb. Calcutt. 253. 1845.

**Type:** not seen.

**Distinguishing features:** Morphology is similar to the var. ***variegata*** but always has white petals.

**Representative examined material**: Nuevo León, 27-III-2024, *E. Estrada 26145* (CFNL). Tamaulipas: 1-IV-2024, *A. Mora-Olivo s.n.* (UAT).

**Comments.** Native in the south of Yunnan, China. Used as ornamental in tropics and subtropics around the world. 

***B. variegata*** L., Sp. Pl. 1: 375. 1753 var. ***variegata***. Basionym: *Bauhinia chinensis* (DC.) Vogel, Nov. Actorum Acad. Caes. Leop.-Carol. Nat. Cur. 19(1): 42. 1843. *Bauhinia decora* Uribe, Fl. Antioq. 193. 1941. *Phanera variegata* (L.) Benth., Pl. Jungh. [Miquel] 2: 262. 1852.

**Type:** Colombia, non-nate, *L.U. Uribe 445* (Holotype: not found. Isotype: US00001278!).

**Distinguishing features:** Cultivated, 3–9 m tall. Leaves are simple, 5–20 cm long, bilobed, and the lobes are rounded. Flowers pink, lilac, or violet. Calyx spathaceous. Petals 5, free. Stamens 6, 5 of them fertile, subequal, or slightly smaller than petals, staminode 1. Fruit 10–22 × 1.5–2.4 cm, oblong, flattened, dehiscent. 

**Representative examined material**: Nuevo León: 6-III-2001, *E. Estrada 12696* (CFNL); Rancho Barrial, 31-VII-1971, *S. Brockwell 2* (TEX-LL). Tamaulipas: 15-III-1986, *C.G. Romo 416* (UAT); 21-II-1985, *R. Ddíaz 296* (UAT); 17-III-1993, *J.L. Mora-López 250* (UAT).

**Comments:** Used as an ornamental. Native of the Yunnan province (China).

***Cercis*** L., Sp. Pl. 1: 374. 1753.

**Type: (**Lectotype: *Cercis siliquastrum* L. designated by N. L. Britton and A. Brown, Ill. Fl. N. U.S. ed. 2. 2: 334. 1913.

Trees or shrubs, deciduous. Leaves simple, alternate, palmately veined. Pedicels articulated. Flowers in short racemes, appearing before the leaves, purple–pink, bisexual, 5-merous, pink or light purple. Calyx gamosepalous, 5-dentate, magenta–red. Corolla papilionoid in appearance, dialypetalous, petals 5. Stamens 10, free, filaments basally pubescent. Fruit flattened, narrowly winged at one side, veined, reddish-brown, tardly dehiscent along both sutures.

The genus is represented by 10–11 species [4,67]: four species in North America, one in Europe, one in Asia, and four in China [67]. 

***Cercis canadensis*** L., Sp. Pl. 374. 1753. Basionym: *Cercis occidentalis* var. *texensis* S. Watson, Bibliogr. Index N. Amer. Bot. 209. 1878. *Cercis reniformis* Engelm. ex A. Gray, Boston J. Nat. Hist. 6: 177. 1850. *Cercis canadensis* var. *texensis* S. Watson, Bibliogr. Index N. Amer. Bot. 209. 1878. *Cercis mexicana* Rose, N. Amer. Fl. 23(4): 202. 1930. *Cercis canadensis* L. var. *mexicana* (Rose) M. Hopk., Rhodora 44: 208. 1942.

**Type:** (Lectotype: Designated by Reveal, Taxon 46(3): 466. 1997. Voucher: (LINN-524.2).

**Distinguishing features:** Tree or shrub, 3–8 m tall, young branches reddish, densely pubescent. Leaves orbicular. Calyx reddish brown. Fruit 5–7.5 × 0.8–1.2 cm, oblong, acute at both ends, flexible, with a small wing longitudinally along one side, dark-brown or dark-reddish, transversely reticulated, glabrous. 

**Representative examined material**: Coahuila: 30-V-1993 *J.A. Villarreal 5906* (ANSM); 23-VIII-2014, *J. Encina 4013* (CFNL); 17-V-2008, *S.G. Gómez 507*, *E.F. García* (CFNL). Nuevo León: 21-IV-2000, *C. Yen y E. Estrada 11406* (CFNL); 3-III- 1988, *E. Estrada C. 1349* (CFNL; MEXU); *22-IX-2001*, *C. Yen y E. Estrada 13075* (MEXU); 13-IV-2002, *C. Yen y E. Estrada 14548* (CFNL; MEXU); 2-III-2003, *C. Yen y E. Estrada 15219* (CFNL). Tamaulipas: 2-VI-1986, *M. Martínez 1176* (CFNL); 20-II-1990, *I. Mata 31* (CFNL); 17-II-1996, *C Ramos 67* (CFNL). Tamaulipas: 5-V-1988, *C.G. Romo 650* (UAT); 28-II-1986, *L. Hernández 1675* (UAT); 16-V-1994, *M. Martínez 2291* (UAT).

**Comments:** In northeastern Mexico, the plants show morphological features such as the margin, shape, and texture of the leaves and the pubescence on branches, leaves, and inflorescences that correspond to both of the recognized varieties for northeastern Mexico (var. *texensis* and var. *mexicana*) [48]; however, in the collected (and herbarium) samples, the morphological characteristics that distinguish them from a cline or overlap with each other, so they cannot be differentiated from each other concerning their infraspecific distinction. In oak, oak–pine forests, piedmont scrub, 850–2400 m. Known as *duraznillo* and used as ornamental. 

**Subfamily Detarioideae** Burmeist., Handb.Naturgesch: 319. 1837.

Trees or shrubs. Leaves paripinnate, bifoliolate, or unifoliolate. Leaflets sometimes have translucent glands. Flowers in racemes or panicles, bisexual or unisexual, actinomorphic or zygomorphic but not papilionate. Sepals 4–5 (rarely to 7), two of them fused, rarely absent, or less than four. Petals present or absent, when present 5–7, imbricate, the adaxial one, the outermost, similar in size or the adaxial one the largest, the other four smaller or only the abaxial one smaller or rudimentary. Stamens commonly 10, sometimes 2-numerous, the filaments connate or free, the anthers opening by longitudinal slits. Fruit dehiscent or indehiscent, woody or the valves thin and samaroid. The subfamily is composed of 84 genera and approximately 760 species [1]. In Mexico, five genera are present (*Cynometra*, *Dicymbe*, *Hymenaea*, *Peltogyne*, and *Tamarindus*). In northeastern Mexico, only one species is recorded, *Tamarindus indicus*.

***Tamarindus*** Gen. Pl., ed. 5. 1754. (Monotypic genus).

**Type:** *Tamarindus indica* L., Sp. Pl. 1: 34. 1753.

***Tamarindus indica*** L., Sp. Pl. 1: 34. 1753. Basionym: *Tamarindus occidentalis* Gaertn., Fruct. Sem. Pl. ii. 310. t. 146. *Tamarindus officinalis* Hook., Bot. Mag. 77: t. 4563 (1851).

**Type**: Habitat in India, America, Aegypto, Arabia. RCN: 271. Lectotype (designated by Jansen, Spices, Condiments Med. Pl. Ethiopia: 249. 1981): Herb. Linn. No. 49.2 (LINN).

**Distinguishing features**: Tree up to 15 m tall. Branchlets in zigzag. Leaves pinnate. Leaflets 9–16 pairs per leaf, glabrous. Flowers pendulous in axillary or terminal racemes, yellow, red-veined, bisexual. Calyx 4-lobed, the lobes with reddish tones. Petals 5, the lower 2 small, resembling scales. Fertile stamens 3, longer than petals, fused and forming a tube with small staminodes. Ovary stipitate, style apically swollen. Fruit pendulous, 7–15 × 2–3.5 cm, compressed, with a thin, brittle, and dry outer layer; a pulpy median layer with fibers; and an internal leathery, septate layer between the seeds. Seeds square to rhomboid, laterally compressed, brown, and shiny.

**Representative examined material**: Nuevo León: 23-IV-2006, *E. Estrada 19874* (CFNL); 6-VII-2003, *C. Yen y E. Estrada 15893* (CFNL).

**Comments**: Used as ornamental in Mexico. It is common to prepare refreshing drinks. Probably native to Africa, but it is cultivated in many parts of the world today.

## 3. Discussion

The family Fabaceae is present throughout the entire surface of Mexico; however, its diversity is more abundant in southern Mexico, especially in the tropical zones [69]. However, the diversity of legumes with an affinity for arid climates is more abundant in northeastern Mexico; this is undoubtedly due to the climate of the region, which is predominantly semiarid. Several genera, such as *Senna* and *Chamaecrista*, present a rich diversity in northeastern Mexico since almost 30% of all Mexican species of both genera are distributed in this region. Twenty-five percent of *Bauhinia* species are distributed in northeastern Mexico. Twenty-five percent of the *Bauhinia* species in Mexico are distributed in the northeast region of the country. Half of the *Erythrostemon* species recorded for Mexico have affinities for semi-arid climates, but there are also genera such as *Hoffmansseggia* and *Pomaria*, where more than 50% of the species present in Mexico are distributed in the northeast region and can be considered genera whose species have arid–thermophilous affinities. Of the three subfamilies of Fabaceae studied in northeastern Mexico, Caesalpinioideae is the most diversified, followed by Cercidoideae and Detarioideae. The subfamily Caesalpinioideae is abundant in tropical biomes, and most species are distributed below 2500 m elevation [4,7,52]. This same distribution and ecological behavior pattern, concerning altitude, is repeated in the northeast of Mexico; the diversity of species of Caesalpinioideae is distributed in practically almost all the ecosystems of this region, mainly in semiarid and subtropical areas, while Cercioideae and Detarioideae are most frequent in subtropical climates in low plains. The tribe Caesalpinieae incorporates new segregated genera of *Caesalpinia* for northeastern Mexico, such as *Coulteria*, *Denisophytum*, *Erythrostemon*, and *Guilandina* [5]. The new taxonomic modifications in Caesalpinioideae [4,5] also altered the number of species in the old and the new segregated genera. This is the case of *Caesalpinia*, with approximately 20 species in northern Mexico, where 19 of them have now been transferred to the new genera or to others such as *Conzattia* [5], *Coulteria* [5], *Denisophytum* [5], *Erythrostemon* [4,5], *Guilandina* [5], *Hoffmannseggia* [5,59], and *Pomaria* [5,60]. Of the total species recorded for the three subfamilies, 88% are native, and 12% are exotic in northeastern Mexico; all of these exotic species are shrubs or trees. The predominant biological forms of the 71 taxa of Fabaceae recorded in northeastern Mexico are almost equally distributed; the herbaceous total 31 taxa, while the shrubs and trees add up to 40 taxa. However, ecologically, shrubs and trees play a predominant role in the vegetation of northeastern Mexico, where those are often the dominant elements of the landscape [69,70]. Of the 63 species of Fabaceae recorded in northeastern Mexico, 22 of them are endemic to Mexico and the state of Texas, USA. Only 11% of the recorded Fabaceae species are endemic to northeastern Mexico. The genera with the highest number of endemism in northeastern Mexico are *Bauhinia* and *Pomaria*. Only *Bauhinia bartlettii* and *Senna guatemalensis* var. *calcarea* are endemic to one state, Tamaulipas. In northeastern Mexico, *Haematoxylum* has only been recorded with pinnate leaves. *Conzattia arborea* has been reported for the state of Tamaulipas, but no records of this species were found by the authors, nor in SEINet databases or any national or foreign herbaria mentioned above, so this species is not included in this study.

## 4. Materials and Methods

### Study Area

Northeast Mexico is represented by the states of Tamaulipas (80,249 Km^2^), Nuevo Léon (64,156 Km^2^), and Coahuila (151,595 Km^2^) (Figure 4). Within this surface are three large physiographic provinces: the Eastern Sierra Madre, the Northern Gulf Coastal Plain, and the Great North American Plain. In northeastern, contrasting changes in vegetation are the result of the variability in soil, relief, and climate [70].

The variability in soils is due to their physical and chemical properties [70]; likewise, the climate is varied along the altitudinal gradient [71]. Among the most common types are tropical, dry, temperate, and cold [71,72] The altitudinal gradient in northeastern Mexico consists of low plains (0–150 m altitude), low hills (200–600 m), steep mountains (900–1600 m), high plains (1400–1800 m), and high peaks (3600 m) [72]. This rich physiographic, climatic, and edaphic variability has evident effects on the structure, composition, and diversity of vegetation, and six of the nine main vegetation types of Mexico are present in this region: xeric scrub, low deciduous forest, evergreen tropical forest, mountain cloud forest, grasslands, oak forest, oak–pine forest, coniferous forest, and subalpine meadow [24,71]. The Fabaceae is one of the three groups of plants best represented in Mexico [73]. It is distributed in all plant communities in northeastern Mexico, being the most ecologically important element in the landscape [19,72]. The aim of this study is to establish the diversity of legumes of the subfamilies Caesalpinioideae (excluding tribe Mimoseae), Cercidoideae, and Detarioideae distributed in the different environments of northeastern Mexico.

Similar to our previous work on this family of plants in northeastern Mexico [11], this study consisted of two phases; The first one was to capture in a database all the genera and species of Fabaceae, subfamilies Caesalpinioideae, Cercidoideae, and Detarioideae recorded in the scientific bibliography for northeastern Mexico. The database was expanded with the personal species records of each of the authors in the study area. In order to complement the entire diversity of this family, botanical specimens from the national herbaria were included: ANSM, CFNL, MEXU, and UAT. The second phase consisted of finding records of legume species from northeastern Mexico in foreign herbaria (CAS, MICH, NY, TX-LL, US). The databases and high-resolution digital photographs of these herbaria were used as well. The databases of digital images of type specimens from the JSTOR Global Plants database were consulted and recorded. The Tropicos [74] platform database, the book Order out of Chaos [75], and the study of *Hoffmansseggia* [76] were consulted to query the designation of lectotypes. In the representative material examined, the symbol “!” indicates that the type specimen for the species was observed by the authors. The accepted scientific names follow WFO (World Flora Online) [77].

To differentiate the subfamilies, tribes, genera, and species of the Fabaceae, different dichotomous keys were created: (1) to differentiate the three studied subfamilies; (2) to separate the tribes within the subfamily (when necessary); (3) to differentiate the genera within each subfamily or tribe; and (4) to separate species within each genus. These keys include the main morphological characters useful for differentiating groups. Measurements of morphological structures were carried out by the authors; when the specimen was not available, measurements were obtained from the literature.

We include the currently accepted name for the species [76]. In addition, the type species for each of the genera and the type specimen for each of the species are incorporated. The synonymy of each of the genera and the species (basionyms, homonyms, and heteronyms) is also contained. As a fundamental part of the species, a morphological description of each subfamily, tribe, genera, species, and infraspecific category was added; it contains the distinctive characters and some of the most characteristic morphology for their recognition. At the end of each description, a comments section is added in order to provide information regarding its global, regional, or endemic distribution, ecology, and uses. The subfamilies, tribes, genera, and species are arranged alphabetically.

## Figures and Tables

**Figure 1 plants-13-02477-f001:**
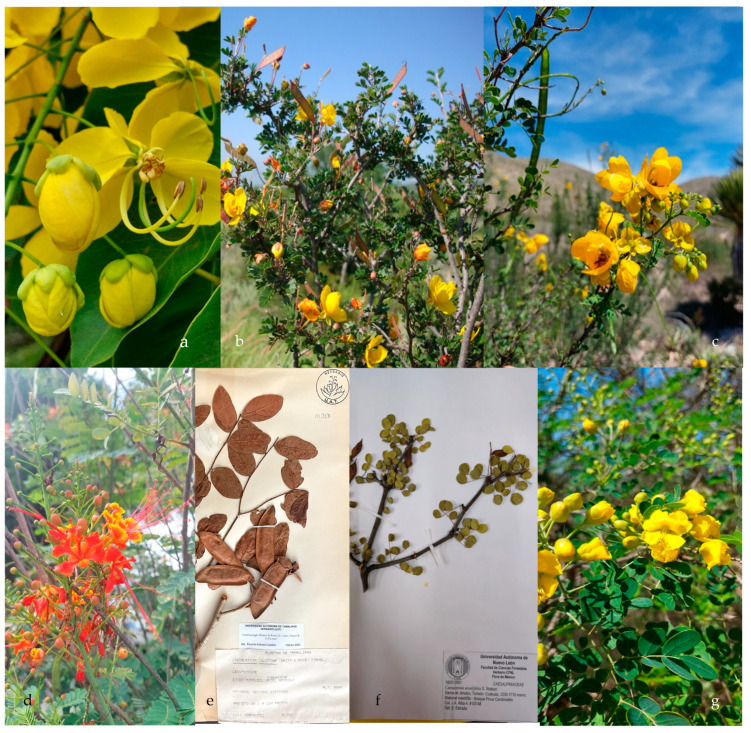
Species representative of the genera of subfamily Caesalpinioideae in northeastern Mexico. Tribe Cassieae: *Cassia fistula* (**a**), *Chamaecrista greggii* var. *greggii* (**b**), and *Senna wislizeni* var. *wislizeni* (**c**). Tribe Caesalpinieae: *Caesalpinia pulcherrima* (**d**), *Coulteria pringlei* (**e**) (Herbarium specimen, deposited in the CFNL scientific collection), *Denisophytum sessilifolium* (**f**) (Herbarium specimen, deposited in the CFNL scientific collection), *Erythrostemon mexicanus* (flowers) (**g**).

**Figure 2 plants-13-02477-f002:**
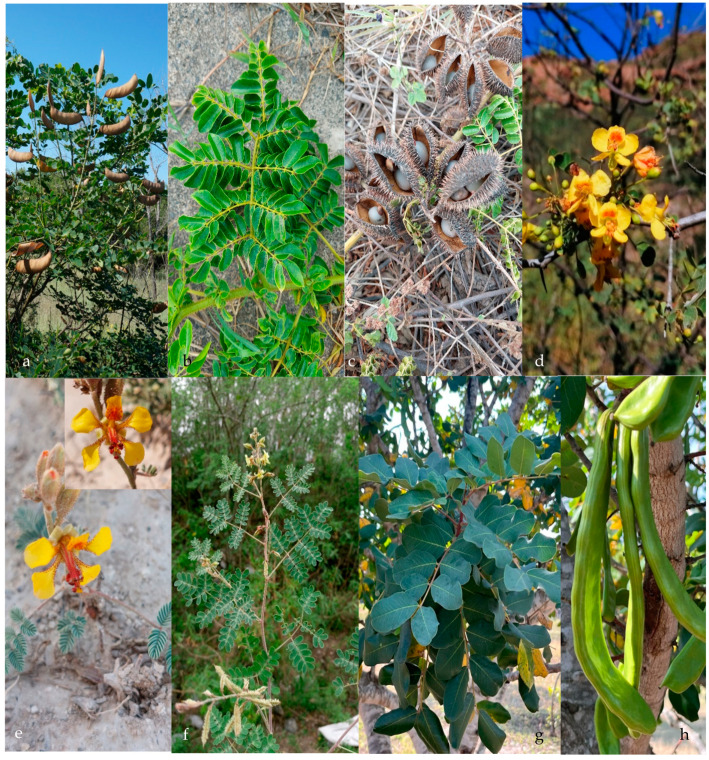
Species representative of the genera of subfamily Caesalpinioideae northeastern Mexico, tribe Caesalpineae: *Erythrostemon mexicanus* (fruits) (**a**), *Guilandina bonduc* leaves (**b**), *Guilandina bonduc* fruit and seeds (**c**), *Haematoxylum brasiletto* (**d**), *Hoffmannseggia glauca* (**e**), *Pomaria wootonii* (**f**). Species representative of the genera of tribe Ceratonieae in northeastern Mexico: *Ceratonia siliqua* leaves (**g**) and *C. siliqua* fruits (**h**).

**Figure 3 plants-13-02477-f003:**
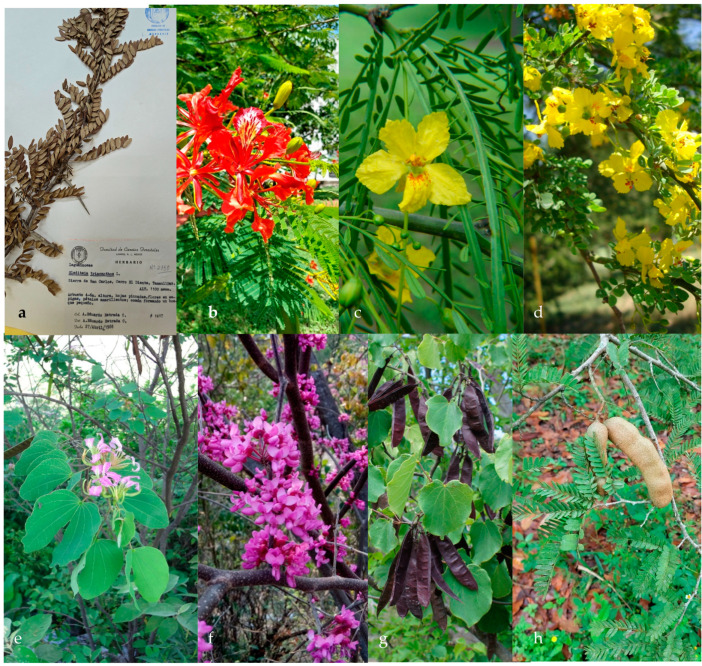
Species representative of the genera of tribe Gleditsieae: *Gleditsia triacanthos* (**a**) (Herbarium specimen, deposited in the CFNL scientific collection). Species representative of the genera of tribe Schizolobieae in northeastern Mexico: *Delonix regia* (**b**), *Parkinsonia aculeata* (**c**), *Parkinsonia texana* var. *macra* (**d**), Species representative of the genera of tribe Cercidoideae: *Bauhinia macranthera* (**e**), *Cercis canadensis* flowers (**f**), *Cercis canadensis* leaves and fruits (**g**). Species representative of the genera of tribe Detarioideae: *Tamarindus indicus* (**h**).

**Figure 4 plants-13-02477-f004:**
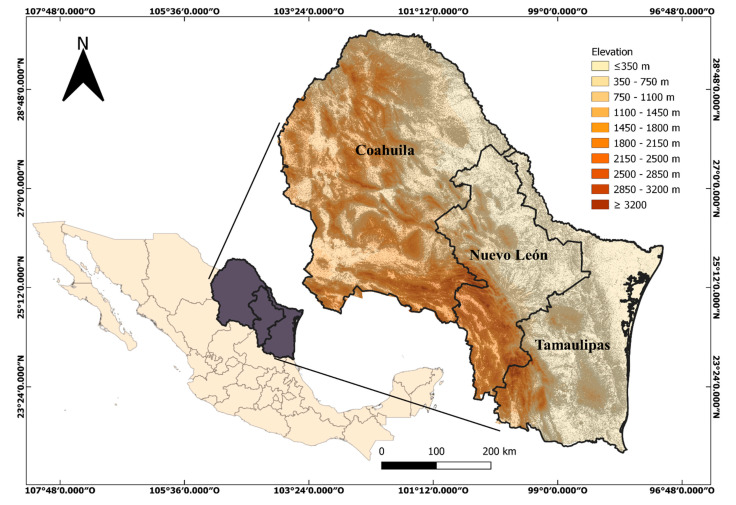
Northeastern Mexico comprises three states: Coahuila, Nuevo León, and Tamaulipas.

**Table 1 plants-13-02477-t001:** Number of species per genus of the three subfamilies in northeastern Mexico compared to Mexico and the world.

	Species Number		
Taxa	World	Mexico	NE Mexico
CAESALPINIOIDEAE			
*Caesalpinia*	10 [4]	1 [4,5]	1
*Coulteria*	11 [4]	7–8 [56,57]	1
*Denisophytum*	8 [4]	1 [5]	1
*Erythrostemon*	31 [4]	11 [58]	5
*Guilandina*	20? [4]	1 [58]	1
*Hoffmannseggia*	23 [4]	9 [58]	5
*Haematoxylum*	5 [4]	3 [52,59]	1
*Pomaria*	16 [4]	8 [51]	6
*Cassia*	39 [4]	3 [58]	1
*Chamaecrista*	368 [4]	20 [58]	6
*Senna*	287 [4]	60 [58]	21
*Ceratonia*	2 [54]	1 [60]	1
*Gleditsia*	13 [1]	1 [61]	1
CERCIDOIDEAE			
*Bauhinia*	160 [52]	30 [57]	7
*Cercis*	10 [52]	1 [52]	1
DETARIOIDEAE			
*Tamarindus*	1 [52]	1 [52]	1

**Table 2 plants-13-02477-t002:** Endemism of the subfamilies Caesalpinioideae and Cercidoideaeeae in northeastern Mexico.

Region	Subfamily Caesalpinioideae	Subfamily Cercidoideae
Endemic to Mexico	*Coulteria pringlei*	*Bauhinia coulteri*
	*Denisophytum sessilifolium*	*Bauhinia ramosissima* var. *ramosissima*
	*Hoffmannseggia humilis*	*B. ramosissima* var. *uniflora*
	*Hoffmannseggia watsonii*	
	*Chamaecrista chamaecristoides* var. *chamaecristoides*	
	*Senna crotalarioides*	
	*S. wislizeni* var. *painteri*	
Endemic to the north of Mexico	*Pomaria canescens*	
	*P. fruticosa*	
	*Senna wislizeni* var. *villosa*	
Endemic to NE Mexico	*Erythrostemon caudatus*	
	*E. phyllanthoides*	
	*Hoffmannseggia drummondii*	
	*H. oxycarpa*	
	*Pomaria wootonii*	
	*Chamecrista greggii* var. *potosina*	
	*Senna demissa* var. *demissa*	
	*S. demissa* var. *radicans*	
Endemic to one state in Mexico	*Senna guatemalensis* var. *calcarea* (Tamaulipas)	*Bauhinia bartlettii* (Tamaulipas)
	*S. monozyx* (Coahuila)	

## Data Availability

The data presented in this study are available on request from the corresponding author.

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
