# Peer review of "The Fabaceae in Northeastern Mexico (Subfamilies Caesalpinioideae (Excluding Tribe Mimoseae), Cercidoideae, and Detarioideae)"

_plants, 2024, doi:10.3390/plants13172477_

Round 1

Reviewer 1 Report (Previous Reviewer 3)

Comments and Suggestions for Authors

After the elaboration, this paper has been significantly improved. I have minor concerns about this manuscript.
Some earlier versions seem to have left their traces in the present text. As a result, some phrases like 'Staminodes present or absent present' (p. 8) appear.
Monosymmetric (zygomorphic) flowers cannot be truly terminal, so the statement that flowers are axillary or terminal in Caesalpinia pulcherrima is most likely erroneous.
Please double-check all Latin names. I do understand there are hundreds of them in this paper but there are some spelling mistakes which are very undesirable in a paper on taxonomy. For example (p. 30), there are specific epithets durangensis and duragensis in the same paragraph.
For me, statements like 'Anthers of the 2-3 abaxial fertile stamens not beaked, the beak symmetrically truncated' sound obscure, as I cannot understand how the absent beak might be truncated.
There are some spelling mistakes, like 'staminades' instead of 'staminodes' and alike.
After a final revision by the authors, this text can be recommended for publication. 

Comments on the Quality of English Language

There are some points to be corrected in this text but these are easy to be make in the course of the editorial process. Much more dangerous are spelling mistakes in terms and Latin names.

Author Response

Comments and responses appear in attached file.

Reviewer 2 Report (New Reviewer)

Comments and Suggestions for Authors

·       Overall, the study is interesting, as the Fabaceae is the most taxonomically complex family and recently updated as per APG IV.

·       In the abstract section “The subfamily Cercidoideae include two genera (Bauhinia and Cercis), and 8 species and the subfamily Detarioideae, includes only 1 genus and 1 species (Tamarindus indicus)” the authors mentioned te term subfamily, but why the species is very few in these two subfamilies?

·       In the introduction section would be better if the authors provide a short overview of the APG to APG-IV for the classification of Fabaceae? The authors only mentioned previously 3 and now 6 subfamilies of Fabaceae. A short paragraph is okay, so one should know clearly about the Fabaceae classification.  

·       In the results section, the authors mentioned that the subfamily Caesalpinioideae is composed of six tribes. The six tribes as a whole? Or only in northeastern Mexico? Please clarify.

·       In the discussion section, “Twenty-five percent of Bauhinia species are distributed in northeastern Mexico. Twenty-five percent of the Bauhinia species in Mexico are distributed in the northeast region of the country” ? Seems repetition?

·       Although these macro-morphological data are available in Flora databases but due to recent changes the author reanalyzed the subfamilies, to clarify and convey a clear message for the readers about the subfamilies of Fabaceae which is worth reading.

·       These Ceasalpinioideae are adapted to which environmental condition? I noted that the authors talking about different environmental conditions in different places, such as arid, semi-arid and tropical regions but it is suggested to analyze the environmental data for the study as well. Such as mean annual temperature, precipitation, humidity, etc illustrated on maps? So, mean annual or monthly (Jan-Dec) bioclimatic is required.  

·       By doing this, the readers will know that these species were adapted to this type of environment and will provide bases for future research such as climatic change.

·       Would be better if the authors provided some figures for the visualization of family genera and species to make it easy for the readers.

·       Please add limitations of the study at the end of the discussion and provide future recommendations as well.

   Note: The MS is interesting and will add a significant contribution to the field.

Comments on the Quality of English Language

Minor editing of English language required

Author Response

Comments and responses appear in attached file.

Round 2

Reviewer 2 Report (New Reviewer)

Comments and Suggestions for Authors

Line 49-53: What about the number of genera and species of the subfamily Duparquetioideae?

The authors did not respond to the previous comment. Please add study limitations at the end of the discussion section and provide future recommendations.

Comments on the Quality of English Language

Minor editing of English language required

Author Response

The changes made are shown in the attached file.

This manuscript is a resubmission of an earlier submission. The following is a list of the peer review reports and author responses from that submission.

Round 1

Reviewer 1 Report

Comments and Suggestions for Authors

The authors have submitted to the Fabaceae of northeastern Mexico by providing various dichotomous keys and morphological descriptions. When describing the distinguishing features, it should be clarified whether they are the personal measurements of the authors or literature data, citing the relevant sources. Complete binomials should be used when referring to species and families for the first time in the text. Although the captions report clear letters, they are missing the corresponding letters in the images, making it difficult for non-experts to connect the name to the image.

However, the nomenclatural section is the most problematic, with numerous gaps and inaccuracies. The authors claim to provide new nomenclatural changes and retypification, but this is not clearly explained. The "Type" section also lacks clear data. If lectotyping is carried out, the research procedure should be explained in more detail, along with which herbaria were checked. They must show more detailed nomenclatural investigations. The authors mention some lectotypes, but it's unclear whether they were created by them or other scholars. The authors consulted the herbarium specimens via JSTOR, and it's important to specify whether all samples were viewed digitally or not. There appear to be new unclear combinations e.g. Chamaecrista flexuosa ssp. texana which is currently considered as a variety while the authors report it as a subspecies. Also I suggest using subsp. rather ssp. Additionally, the international code must be cited, and the articles must be correctly applied for the nomenclatural part.

Comments on the Quality of English Language

The text must be profoundly revised from a linguistic point of view; there are difficult-to-understand sentences, especially in the introduction.

Reviewer 2 Report

Comments and Suggestions for Authors

The article under review provides a critical assessment of the diversity of three subfamilies of Fabaceae in north-eastern Mexico. The work is very thorough and detailed. The work presented for evaluation is the second part of the results of an extensive survey. 

In my opinion, the paper is well prepared and has only minor shortcomings. Of course, if the manuscript under review were not the second part, I would suggest numbering the genera and species within genera to make it easier for the reader to find the information they are looking for. But as this manuscript is the second part, it is irrational to change the presentation style. 

It is very irrational that the most important illustration of the paper, with a map of the study area, is at the end of the paper, when it should be at the beginning, together with the whole Methods section (after the Introduction). I understand the journal's suggestion to put the methodology at the end of the article, but it is not imperative. 

The question is, why are the figures of the subfamily Caesalpinioideae (or tribe?) divided into three figures? It would be more rational to put the photos of plants of one subfamily in one figure, or in the worst case in two figures, but not to spread them over three figures. 

In Table 1, the punctuation needs to be checked and tidied up. The caption of the table and the abbreviations in the table itself are not written in the same way.

The numbering of the tables needs to be corrected. There are now two tables with identical numbers (Table 1).

In the table (now labelled Table 1, but should be Table 2), I suggest clarifying the exact states in which the endemic species, now classified as 'Endemic to one state in Mexico', are located.

The manuscript contains punctuation and technical errors (missing full stops, commas, spaces between words). It is therefore necessary to read it carefully and to correct the omissions.

Comments on the Quality of English Language

Minor corrections required

Reviewer 3 Report

Comments and Suggestions for Authors

The reviewed manuscript is important for florists and taxonomists as it aims to describe the diversity of three leguminous subfamilies in northern Mexico. However, in its present form this paper is not ready for publication nor review. First of all, not only would the language of this text benefit from special editing but the very review process is impossible without a thorough language correction, preferentially by a native speaker. I made some changes and suggestions (see attached file) but these are not enough.
There are numerous typos which should be corrected. Of course, some language and style editing can be done in the course of the paper preparation for publishing. However, very many things can be improved only by a specialist. These are so abundant that I would recommend that this paper should be rejected with encouragement to resubmit in updated form.
Some terminology is used erroneously. It is of utmost importance to distinguish between flowers and inflorescence; the latter cannot be in racemes and panicles as racemes and panicles are inflorescences themselves.
The introductory and concluding sections should be reviewed and shortened. There is no need to describe all changes made in the leguminous phylogeny so many times.
With this in mind, I would suggest to reject this paper in its present form. It is not because of its scientific quality; on the contrary, I guess this paper is timely. However, the very text with its language and style need to be updated to make the paper readable and reviewable. I wish the authors good luck.

Comments on the Quality of English Language

Very serious review of language and style is needed focusing on punctuation (which is sometimes excessive) and spelling (there are typos even in terms).

Round 2

Reviewer 1 Report

Comments and Suggestions for Authors

The manuscript's nomenclatural aspect requires further clarification. It is not evident from the text whether the lectotypes were designated by the authors or by previous authors. Additionally, if the lectotypes were proposed by other authors, this must be explicitly stated in the manuscript.

Many type indications are incorrect, for example: for Bauhinia divaricata L. is reported "Type: Mexico, State of San Luis Potosi. Tamasopo Canyon, 25-VI-1890, C.G. Pringle 3104 (Isotype: M217258!; UC101719!; GH59707!). However, this species was lectotypified by Stearn in Introd. Linnaeus' Sp. Pl. (Ray Soc. ed.) : 47 (1957) with a voucher in the Herb. Clifford: 156, Bauhinia 2 (BM).  Therefore, more detailed research by the authors needs to be carried out.

Furthermore, for the next revision step I suggest sending a clean version of the text for better reading and understanding.

Reviewer 3 Report

Comments and Suggestions for Authors

As I may see, the authors performed significant work on their manuscript. The description of changes were is very helpful. I am glad to see unnecessary repetition removed.
However, there are still many typos and flaws which should be removed. The most undesired are typos in terms and taxonomical names, such as Mimosoeae instead of Mimoseae.  Some of generic and specific names are not italicized. Please double-check all Latin names as it is the thing which can only be corrected by the authors.
Some language correction is still in need especially in punctuation.
Again, the descriptions of inflorescences should be reformatted. It was not enough to remove 'arranged'. Let me illustrate this by one example. You wrote, 'Inflorescences in terminal or axillary racemes or panicles.' It is not correct because racemes and panicles are inflorescences themselves, they are not parts of some other inflorescences. Of course, the overall inflorescence morphology is one of the most complicated areas of plant morphology, full of terminological controversies. However, the descriptions you gave should be corrected. In the sentence cited, it could be rephrased like, 'Flowers in terminal or axillary racemes or panicles.' However, it is not always so easy. I would recommend probably the best review on the leguminous inflorescences to reference:
https://www.cabidigitallibrary.org/doi/full/10.5555/19931640374
(I am not sure however if it downloadable)
Some information can be also found here:
https://academic.oup.com/aob/article/112/8/1567/148515
Although this paper is devoted to the inflorescences of Papilionoideae, some terms can be found there.
I hope these suggestions would not discourage the authors and help make this important text more readable and comprehensive.

Comments on the Quality of English Language

There are some flaws, especially connected with excessive punctuation. At best, these changes can be made by the editorial team in the course of this paper's preparation for publication. The correction of typos in terms is the thing which only the authors can do.